# The chiral SYK model in three-dimensional holography

Alexander Altland[1] [*], Dmitry Bagrets[1,2] [†], Nele Callebaut[1] [‡],
and Konstantin Weisenberger[1] [∘]

**1** Institute for Theoretical Physics, University of Cologne, Zülpicher Str. 77a,
50937 Cologne, Germany
**2** Peter Grünberg Institute, Quantum Computing Analytics (PGI-12),
Forschungszentrum Jülich, 52425 Jülich, Germany

[*] alexal@thp.uni-koeln.de
[†] d.bagrets@fz-juelich.de
[‡] nele.callebaut@thp.uni-koeln.de
[∘] kweisenb@uni-koeln.de

## Abstract

A celebrated realization of the holographic principle posits an approximate duality between the $(0+1)$-dimensional quantum mechanical SYK model and two-dimensional Jackiw-Teitelboim gravity, mediated by the Schwarzian action as an effective low energy theory common to both systems. We here propose a generalization of this correspondence to one dimension higher. Starting from different microscopic realizations of effectively chiral $(1+1)$-dimensional generalizations of the SYK model, we derive a reduction to the Alekseev-Shatashvili (AS)-action, a minimal extension of the Schwarzian action which has been proposed as the effective boundary action of three-dimensional gravity. In the bulk, we show how the same action describes fluctuations around the Euclidean BTZ black hole configuration, the dominant stationary solution of three-dimensional gravity. These two constructions allow us to match bulk and boundary coupling constants, and to compute observables. Specifically, we apply semiclassical techniques inspired by condensed matter physics to the computation of out-of-time-order correlation functions (OTOCs), demonstrating maximal chaos in the chiral SYK chain and its gravity dual.

# 1 Introduction and summary of results

Holography posits a duality between gravitational bulk theories and quantum boundary theories in one dimension lower [1–3]. While this principle is formulated at a great level of

generality, only relatively few concretely worked out realizations exist. Alongside Maldacena's duality between four-dimensional super Yang-Mills theory and five-dimensional AdS quantum gravity [3–6], the holographic correspondence linking two-dimensional Jackiw-Teitelboim gravity [7, 8] to the quantum mechanics of the SYK model describing $N$ randomly interacting Majorana fermions [9, 10] has been another case study attracting a great deal of attention [10–17]. This low-dimensional manifestation of holography is not only simpler than the classic example but also conceptually different: While the latter involves two precisely defined theories, the two-dimensional duality relates an *ensemble* of random boundary theories to a gravitational bulk describing ensemble correlations [18]. This statistical correspondence is formulated at a high level of concreteness. In fact, there exist *two* bridges between bulk and boundary, describing parametrically distinct regimes. The first addresses fine-grained structures of microscopic spectra or, equivalently, late times of the order of the inverse of the many-body level spacing, exponential in the number of microscopic constituents of the boundary theory, $N$ [18–22]. The second focuses on the complementary regime of early times, polynomial in $N$. It is based on a reduction of the SYK model and the JT partition sum to Liouville quantum mechanics or the Schwarzian action as a common effective theory in the relevant time window [12, 14, 17, 23].

What both correspondences just outlined have in common is that they rely on a high level of microscopic control individually for bulk and boundary theory. On its basis, we understand that the correspondence between the SYK ensemble and of the JT partition sum is, in fact, not perfect but limited to effective theories with parametrically defined scopes, as indicated above. For example, at finite temperature $T = \beta^{-1}$, Liouville quantum mechanics is described by the imaginary time action

$$S[f] = -M \int_0^\beta d\tau \{f, \tau\}, \qquad \{f, \tau\} = -\frac{1}{2} \left(\frac{f''}{f'}\right)^2 + \left(\frac{f''}{f'}\right)', \qquad (1)$$

where $f' = d_\tau f(\tau) \geq 0$, and $M \sim N$ is a coupling constant of dimensionality 'time'. In the context of the SYK model, this action describes the invariance of the theory under reparametrizations of time, where $f : S^1 \to S^1, \tau \mapsto f(\tau)$, is a diffeomorphism of the imaginary time circle onto itself, $f \in \mathrm{Diff}_+(S^1)$ [10]. In JT gravity, on the other hand, $S$ quantifies the action cost associated to fluctuations of a one-dimensional boundary of two-dimensional space, as described by a geometric deformation $f(\tau)$ [12]. The holographic correspondence unfolds via the approximate reduction of boundary and bulk theory to this common effective action in the 'semiclassical' regime of time scales $\tau \sim N$.

The success of such concepts has sparked a surge of activity aiming for generalizations to higher dimensions. An important step in this direction was achieved by Cotler and Jensen, who proposed the Alekseev-Shatashvili (AS) action to assume the role of Liouville quantum mechanics as a boundary theory for AdS$_3$ gravity. This theory, a $(1 + 1)d$ extension of the Schwarzian theory, is defined by the action [24, 25]

$$S_\pm[f] = \frac{C}{24\pi} \int_0^\beta d\tau \int_0^L dx \left(\frac{f'' \partial_\pm f'}{f'^2} - \frac{4\pi^2}{\beta^2} f' \partial_\pm f\right), \qquad \partial_\pm = \frac{1}{2}(u^{-1}\partial_\tau \mp i\partial_x), \qquad (2)$$

where $f(x, \tau)$ now is a reparametrization field depending on space and time, subject to periodic boundary conditions in both directions, and $f' \equiv \partial_\tau f > 0$. The action depends on two coupling constants, the dimensionless $C \sim N \gg 1$, and the velocity scale $u$. Finally, the sign '$\pm$' indicates a sense of chirality to be discussed below.

In the limit of highly anisotropic boundaries, $\beta \gg L/u$, nonvanishing spatial derivatives get frozen out, $\partial_x f(x, \tau) = 0$, and $S_\pm[f]$ reduces to the Schwarzian action. Much as the

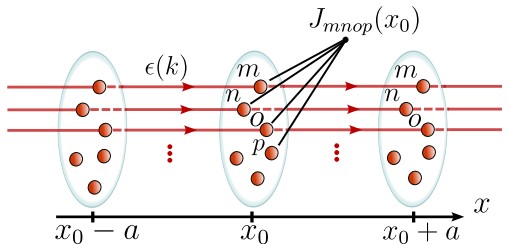

**Figure 1:** Sketch of the chiral SYK model: A chain of SYK grains, each containing $N$ Majorana fermions with the well-known grain-local Majorana interaction $J_{mnop}(x_0)$. Across grains, each species of Majoranas is coupled independently by a kinetic term with chiral dispersion $\epsilon(k)$.

latter describes black hole states in JT-gravity, the spatiotemporal fluctuations of the AS theory describe black hole states in AdS$_3$ [26, 27], the so-called BTZ solutions [28, 29]. The latter dominate the gravitational path integral at high temperatures, $\beta \ll L/u$, indicating that the AS action is the relevant semiclassical theory for this[1] regime.

The purpose of the present paper is to add concreteness to the holographic principle revolving around the AS action. Our first contribution is the definition of a microscopic boundary theory, i.e. a theory assuming the role of the SYK model in one-dimension lower. The second is an explicit derivation of the AS theory as effective theory of fluctuations around BTZ states in the three-dimensional bulk. These derivations will get us in a position to relate bulk and boundary on a microscopic level, matching coupling constants. Building on this correspondence, we will introduce streamlined approaches to the computation of correlation functions, specifically out-of-time-order correlation functions (OTOCs). On this basis, we will discuss the concept of 'maximal chaos' on both sides of the holographic correspondence.

## 1.1 Boundary perspective

What is missing so far in the discussion of the three-dimensional holographic correspondence is a microscopic boundary theory assuming the role of the SYK model in one dimension lower. In this paper we propose such a model system, building on a combination of four principles, exhibited by various 'realistic' classes of condensed matter systems: a chiral, approximately linear single particle dispersion, strong local interactions, static randomness, and the violation of particle number conservation. The first is realized at edge modes of topological insulators, such as two-dimensional quantum Hall insulators [30]. In these systems, a nonvanishing bulk topological invariant requires the presence of left- or right-propagating gapless boundary modes governed by an approximately linear dispersion. In the presence of Coulomb interactions — the second principle — these modes define a universality class known as the *helical liquid* [31]. We consider a helical liquid with $N/2 > 1$ co-propagating edge modes, corresponding to a bulk with invariant $N/2$. We also consider the system coupled to a nearby superconductor 'proximitizing' the system via particle-number non-conserving Andreev scattering operators. Finally, we assume the presence of impurities rendering the systems single particle states effectively random (cf. Fig. 1).

All these principles can be individually realized at topological insulator surfaces. However, we here make the stronger assumption that for an appropriate parameter configuration they combine to define a system in the universality class of the chiral SYK model. To see how this may come about in principle, consider the system coarse grained into spatial units

---

[1]By analogy with the two-dimensional correspondence, the description of long imaginary time scales $\sim \exp(N)$, calls for a different boundary theory.

whose extension is defined by the range of the two-body interaction. If we now consider the creation operators of helical edge fermions in a Majorana representation, symbolically $c = \gamma + i\gamma'$, and represent the interaction operator in a basis of the random single particle eigenstates defined by the combination of impurity and Andreev scattering, we obtain the model Hamiltonian

$$H = \sum_{ik} \epsilon(k)\gamma_k^{i\dagger}\gamma_k^i + \sum_{ijkl} \int_0^L dx\, J_{ijkl}(x)\gamma^i(x)\gamma^j(x)\gamma^k(x)\gamma^l(x), \qquad (3)$$

for $N = (N/2) \times 2$ Majorana modes $\gamma_i$. Here the coefficients $J_{ijkl}$ inherit the randomness of the single particle states and hence are random themselves, while the first term accounts for a dispersion with low energy asymptotics $\epsilon_k \sim k$. (Note that a chiral system in which quantum states propagate only in one direction retains its chirality in the presence of randomness, there is no 'backscattering' in this case.)

In Eq. (3), $\gamma^i(x) = \frac{1}{\sqrt{L}} \sum_k \gamma_k^i e^{ikx}$, are Majorana fermion operators, $\gamma^i(x)^\dagger = \gamma^i(x)$, with anti-commutation relations $\{\gamma^i(x), \gamma^j(x')\} = \delta(x - x')\delta^{ij}$. Adopting a rationale previously applied in the construction of 'effective Hamiltonians' for quantum dots subject to random scattering [32], we assume the interaction coefficients to be Gaussian distributed as

$$\langle J_{ijki}(x)J_{ijkl}(x')\rangle = \frac{3!J^2}{(k_0 N)^3}\delta(x - x'), \qquad (4)$$

where $J$ is the characteristic single particle interaction energy, and $k_0$ an effective momentum cut-off related to the correlation length $a$ of the random interaction, $k_0 = \pi/a$. We make the non-trivial assumption that the interaction energy is dominant, in the sense that

$$J \gg \Lambda \equiv |\epsilon(k_0)| \qquad (5)$$

for all relevant momentum scales below $k_0$, where the scale $\Lambda$ defines a maximal kinetic energy.

Before turning to the further discussion of this model, let us mention a few other spatial extensions of the SYK model. Ref. [33] introduced a model with linear dispersion $\epsilon(k) = vk$ and random but spatially uniform interaction coefficients. While this system is chaotic and chiral like Eq. (3), its long range correlations lead to symmetries different from local reparametrization invariance. A model with long-ranged disorder and linearly dispersive non-chiral fermions, ($\epsilon(k) = \pm vk$ for right/left movers) was discussed and approximately solved in [34]. The existence of connections to AdS$_3$ was left as an open question by the authors. Finally, Refs. [35] and [36] proposed two-dimensional generalizations of the SYK model (conceptually, one-dimensional models subject to time dependent interactions). Compared to the standard '0-dimensional' SYK Hamiltonian, this is a stepup by two dimensions, with a less direct connection the holographic principle in one dimension lower. The criteria motivating the present model ansatz Eq. (3) include spatial locality of interactions and statistical correlations, a symmetry contents identical to that of the AS action, and principal realizability as a many-body system (The Hamiltonian Eq. (3) may be conceptualized as an effective Hamiltonian describing electrostatically correlated multi-channel quantum Hall edge modes).

## 1.2   Bulk perspective

In section 3, we will derive the AS action from bulk gravity. In essence, this amounts to the lifting of an analogous construction for JT gravity, where the Schwarzian action emerged as a boundary-fluctuation action of two-dimensional black hole solutions. In one

dimension higher, alongside global AdS$_3$, BTZ black holes provide stationary solutions, and they reduce to JT black holes upon dimensional reduction [37]. One expects the same to happen at the level of the boundary actions, suggesting that fluctuations around the BTZ saddle are described by the AS action. Indeed, a number of works revealed intimate connections between the BTZ geometry and the AS action (2). To categorize these previous contributions, we temporarily denote an AS action with Schwarzian derivatives acting in the spatial, imaginary, or a real time direction by AS$_{x,\tau,t}$, respectively.

Working within the Chern-Simons representation of three-dimensional gravity (for a review, see section 3.3), the seminal paper [26] identified two chiral copies of the AS$_x$-action as the theory of fluctuations around global AdS$_3$, the vacuum state of AdS$_3$-gravity. Modular invariance[2] then implied that two chiral copies of the AS$_\tau$-action provide the theory of fluctuations around Euclidean BTZ black holes. Given that the BTZ solution has one boundary in Euclidean signature and two in Lorentzian signature (see e.g. [38]), Refs. [26] and [39] proposed a quadruple AS$_t$ description for the two-sided Lorentzian BTZ. Ref. [27] went one step further to suggest AS$_\tau$ as a universal effective quantum theory of AdS$_3$-gravity in the high-temperature regime. Other investigations of AdS$_3$ gravity employing the AS action include [40], and works that focus on the phase space of one- [26, 41, 42] and two-boundary solutions [39, 43], or the spectral form factor [44].

In this paper we will add a missing element to this web of constructions, namely a direct derivation of the AS$_\tau$ action as the theory describing fluctuations around the BTZ black hole in Euclidean signature. This derivation, as opposed to a more indirect one based on modular invariance arguments, will be instrumental for our explicit comparison of bulk and boundary. Starting from Chern-Simons theory defined on a manifold topologically equivalent to a solid torus, we will derive the theory of fluctuations in a framework exchanging the role of space and time relative to the expansion around the AdS$_3$ vacuum [26]. This reassignment plays a crucial role in our construction. It leads to a high level of parallelism in the reduction of the boundary theory (the chiral SYK chain) and the bulk theory to their respective low energy fluctuation actions, and in particular allows us to match coupling constants. On this basis, we will then move on to the computation of observables.

The rest of the paper is organized as follows. In the next section, we process the model Eq. (3) by an extended version of the $G\Sigma$-approach to the SYK model [9], and derive the AS action. In section 3, we obtain the same action from Euclidean gravity. Finally, in section 4 we consider signatures of chaos described by the Liouville field theory reformulation of the AS functional integral. We conclude in section 5, and various technical details are relegated to Appendices.

## 2   Derivation of the AS action from the boundary theory

In this section, we derive the semiclassical boundary theory from the one-dimensional SYK model. The construction proceeds along a sequence of steps, which are extended versions of those used in one dimension lower en route from the SYK model to the Schwarzian action: a representation of the disorder averaged theory in terms of a $G\Sigma$-functional, a mean field treatment of the latter, an identification of a weakly broken reparametrization symmetry and a derivation of the corresponding Goldstone mode action, which happens to be of the AS$_\tau$ form. In course of the mean-field analysis, one realizes that the behavior of

---

[2]The AdS/CFT correspondence combined with modular invariance of CFT on a torus imply a dual relation between the Euclidean global AdS and the BTZ solution of AdS$_3$ gravity. Both solutions are defined on solid tori, and they map onto each other by an exchange of the boundary space and Euclidean time coordinate, cf. section 3.3.

the one-dimensional SYK model based on *helical liquids* crosses over to the 'Fermi-liquid' at temperatures below $T_\Lambda \sim \Lambda^2/J$, where $\Lambda$ is a measure for the spatial coupling strength. As a consequence, the holographic correspondence of this particular model is limited to a (parametrically wide) temperature interval, $T_\Lambda < T < J$. This limitation motivates us to formulate another variant of a chiral SYK model, whose dispersion remains approximately flat over an extended window around $k = 0$. Such an exotic dispersion relation can be physically realized in so-called soft quantum Hall edges. We demonstrate that the SYK-like phase for this second model can be stabilized including the deep infrared limit, leading to a holographic duality also for the range of temperatures $T_* < T < J$, where the scale $T_* \ll T_\Lambda$ is identified in Appendix D.3.

## 2.1   Generalized $G\Sigma$-action and mean-field analysis

We begin with a recapitulation of the construction of the Luttinger-Ward functional [45] for a single SYK cell and the emergent reparametrization invariance of the corresponding stationary phase equations. In the context of the SYK model, this functional is known as $G\Sigma$-action and assumes the following form:

$$S[\Sigma, G] = -\frac{N}{2}\left[\mathrm{Tr}\ln(\partial_\tau + \Sigma) + \int d\tau_{1,2}\left(\frac{J^2}{4}\left[G_{\tau_1\tau_2}\right]^4 + \Sigma_{\tau_2\tau_1}G_{\tau_1\tau_2}\right)\right]. \tag{6}$$

Here, $\tau$ is imaginary time, which in the zero temperature limit considered presently becomes an unbounded real variable, and the pair of field variables $(G, \Sigma)$ is bilocal in time (and in a replica index which we suppress for simplicity, as it will not play a role throughout).

The factor $N$ upfront invites a stationary phase approach, where variation of the action in $G$ and $\Sigma$ leads to the stationary phase equations $-(\partial_\tau + \Sigma)G = \mathbb{1}$ and $\Sigma_{\tau\tau'} = J^3(G_{\tau\tau'})^3$. In the low energy limit, where $\partial_\tau$ is negligibly small compared to the high energy scale $J$, these possess a famously degenerate [10, 13] set of solutions. To formulate it, consider a *reparametrization of time*, i.e. a smooth and invertible function $\tau \mapsto t = f(\tau)$, with inverse $t \mapsto \tau = F(t)$. One can then show [10, 13], that the configurations

$$G_{\tau_1\tau_2} \equiv f_1'^{1/4} G^0_{t_1t_2} f_2'^{1/4}, \qquad G^0_{tt'} = -\frac{1}{(4\pi)^{1/4}J^{1/2}}\frac{\mathrm{sgn}(t-t')}{|t-t'|^{1/2}},$$

$$\Sigma_{\tau_1\tau_2} \equiv f_1'^{3/4} \Sigma^0_{t_1t_2} f_2'^{3/4}, \qquad \Sigma^0_{tt'} = -\frac{J^{1/2}}{(4\pi)^{3/4}}\frac{\mathrm{sgn}(t-t')}{|t-t'|^{3/2}}, \tag{7}$$

with $t_i = f(\tau_i)$ and $f_i' = f'(\tau_i)$ solve the mean-field equations. Thinking of imaginary time compactified on a circle, and $f(\tau)$ as a diffeomorphism from the circle onto itself, this identifies the diffeomorphism group $\mathrm{Diff}(S^1)$ as the symmetry group of the theory. Included in this group is the three-dimensional subgroup $\mathrm{SL}(2,\mathbb{R})$, represented as

$$f(\tau) = \frac{a\tau + b}{c\tau + d}, \qquad ad - bc = 1. \tag{8}$$

This subset of transformations leaves the stationary phase configurations invariant, i.e. $X_{\tau\tau'} = X^0_{\tau\tau'}$ with $X = G, \Sigma$. In this way, we have identified the coset space $\mathrm{Diff}(S^1)/\mathrm{SL}(2,\mathbb{R})$ as the Goldstone mode manifold of the single SYK grain. (We note that the innocently looking $\mathrm{SL}(2,\mathbb{R})$ subgroup will play an extremely important role as a consistency checker in the construction of the full theory below.)

We now generalize the $G\Sigma$–functional (6) to the (1+1)-dimensional case. Defining the Green's function

$$G^{xx'}_{\tau\tau'} = -\frac{1}{N}\sum_i \langle \gamma^i(\tau, x)\gamma^i(\tau', x')\rangle, \tag{9}$$

which is now non-local both in time and space, the functional becomes

$$S[\Sigma, G] = -\frac{N}{2}\left[\mathrm{Tr}\ln\left(\partial_\tau + \epsilon_{\hat{k}} + \Sigma\right) + \frac{J^2}{4k_0^3}\int dx\, d\tau_{1,2}\left[G_{\tau_1\tau_2}^{xx}\right]^4 + \mathrm{Tr}(G\Sigma)\right], \qquad (10)$$

Here, the trace extends over temporal and spatial degrees of freedom with a self-energy $\Sigma_{\tau\tau'}^{xx'}$ whose matrix structure is defined by that of $G$. The caret notation, $k \to \hat{k}$, indicates that space and position operators are subject to canonical commutation relations, $[\hat{k}, \hat{x}] = -i$. Variation of the action (10) leads to the extended set of equations,

$$(i\epsilon - \epsilon_k - \Sigma_\epsilon)G_{\epsilon,k} = 1, \qquad \Sigma_{\tau_1\tau_2} = (J^2/k_0^3)\left[G_{\tau_1\tau_2}^{xx}\right]^3, \qquad (11)$$

with a saddle point self-energy $\Sigma_{\tau_1\tau_2}^{x_1x_2} = \delta_{x_1x_2}\Sigma_{\tau_1\tau_2}$, local in space[3]. Similarly, the Green's function at coinciding spatial points $G_\epsilon \equiv G_\epsilon^{xx}$ is given by the momentum integral,

$$G_\epsilon = \frac{1}{2\pi}\int\limits_{-k_0}^{k_0} dk\, G_{\epsilon,k} = \frac{1}{2\pi}\int\limits_{-k_0}^{k_0} \frac{dk}{i\epsilon - \epsilon_k - \Sigma_\epsilon}, \qquad \epsilon_k = v_0 k, \qquad (12)$$

and $G_{\tau-\tau'}$ is its temporal Fourier transform from the energy domain.

Before turning to the self-consistent solution of Eqs. (11-12), we need to address the regularization of our model and its associated many-body Hilbert space. Recalling that the parameter $a$ defined below Eq. (4) sets a correlation radius of the random interaction, we introduce coarse-grained Majorana operators $\gamma_n^j$ in position space at discrete location $x_n = na$ as

$$\gamma_n^j = \left(\frac{a}{L}\right)^{1/2}\sum_{|k|<k_0}\gamma_k^j e^{ikx_n}, \qquad \frac{1}{2}\{\gamma_n^i, \gamma_{n'}^j\} = \delta_{nn'}\delta^{ij}, \qquad (13)$$

with $L$ being the system size. The Hamiltonian of the SYK-like interaction then becomes a sum over lattice sites,

$$H_{\mathrm{SYK}} = \frac{1}{4!}\sum_n\sum_{ijkl}J_{ijkl}^n\gamma_n^i\gamma_n^j\gamma_n^k\gamma_n^l, \qquad \langle J_{ijkl}^n J_{ijkl}^{n'}\rangle = \frac{3! J^2 \delta^{nn'}}{(\pi N)^3}, \qquad (14)$$

with lattice two-body matrix elements $J_{ijkl}^n = a^{-1}J_{ijkl}(x_n)$. The kinetic energy, as before, is best expressed as a sum over discrete momenta, quantized in units of $2\pi/L$, see Eq. (3). With $N_x = L/a$ representing the number of sites, the dimension of the Hilbert space is then given by $\mathcal{D} = 2^{N_x N/2}$. Fig. 1 illustrates this representation of a model reflecting its coarse-grained regularization. Within such regularization scheme the Majorana Green's function at coinciding spatial points, $G_\epsilon$, is expressed via a momentum integral bounded by the UV cut-off $k_0$, see Eq. (12).

Turning to the solution of the mean-field equations, we introduce the important energy scale,

$$T_\Lambda = \frac{(\pi\Lambda)^2}{J}, \qquad T_\Lambda \ll \Lambda \ll J, \qquad (15)$$

delineating two distinct regimes: 'weakly dispersive' ($\epsilon \gg T_\Lambda$) and 'strongly dispersive' ($\epsilon \ll T_\Lambda$) (see Ref. [46]). In the weakly dispersive limit, the kinetic energy is negligible compared to the self-energy, i.e. $\epsilon_k \ll \Sigma(\epsilon)$, and both the Green's function and self-energy

---

[3]To shorten formulae we will often abbreviate expressions for the $\delta$-function as $\delta_{x_1x_2} = \delta(x_1 - x_2)$.

can, with good accuracy, be approximated by the conventional SYK form (cf. Eq. (7)), shown here in the energy and time domain for later reference:

$$G_\epsilon^0 = -\frac{ik_0\mathrm{sgn}(\epsilon)}{\sqrt{J|\epsilon|}}, \qquad \Sigma_\epsilon^0 = -\frac{i\mathrm{sgn}(\epsilon)}{\pi}\sqrt{J|\epsilon|}, \qquad \epsilon \gg T_\Lambda, \tag{16}$$

and

$$G_{\tau\tau'}^0 = -\frac{k_0\mathrm{sgn}(\tau-\tau')}{\sqrt{2\pi J|\tau-\tau'|}}, \qquad \Sigma_{\tau\tau'}^0 = -\frac{J^{1/2}\mathrm{sgn}(\tau-\tau')}{(2\pi|\tau-\tau'|)^{3/2}}, \qquad \tau \ll 1/T_\Lambda. \tag{17}$$

In the strongly dispersive limit, the integration range in Eq. (12) can be extended to infinity and the momentum integral is defined by a pole at $k_* = -\Xi_\epsilon/v_0$, where we have introduced $\Xi_\epsilon = \Sigma_\epsilon - i\epsilon$. This results in a simple, 'Fermi-liquid'-like Green's function,

$$G_\epsilon^{\mathrm{FL}} = -\frac{i\mathrm{sgn}(\epsilon)}{2v_0}, \qquad G_{\tau-\tau'}^{\mathrm{FL}} = -\frac{1}{2\pi v_0} \times \frac{1}{\tau-\tau'}, \tag{18}$$

which is independent of the details of $\Xi_\epsilon$. Consequently, the self-energy evaluates to $\Sigma_\tau = (J^2/k_0^3)G_\tau^3 \propto \tau^{-3}$, which results in a quadratic energy dependence, $\Sigma_\epsilon \propto i\epsilon^2$. Taking into account all numerical constants, limiting results for the self-energy read,

$$\Sigma_\epsilon = -\frac{i\mathrm{sgn}(\epsilon)}{\pi}\sqrt{J|\epsilon|}, \qquad T_\Lambda \ll \epsilon \ll J,$$
$$\Sigma_\epsilon = \frac{i\sqrt{J}}{16\,T_\Lambda^{3/2}}\epsilon^2\mathrm{sgn}(\epsilon), \qquad \epsilon \ll T_\Lambda. \tag{19}$$

The two asymptotics match at the scale $\epsilon \sim T_\Lambda$, indicating the self-consistency of the above analysis. In the deep infrared limit, the self-energy when continued to real energies, defines a damping rate of Majoranas, $\gamma(\epsilon) \propto (\epsilon/T_\Lambda)^2$, reminiscent of the Fermi-liquid physics. It is also worth noting here that a crossover from SYK to the 'Fermi-like' saddle-point described by Eq. (19) is fully analogous to that occurring in granular SYK arrays [47, 48].

The above considerations imply that stationary solutions for $G$ and $\Sigma$ exhibit an approximate reparametrization invariance only within the time range $J^{-1} < t < T_\Lambda^{-1}$. At longer times, the 'Fermi-liquid' saddle-point takes over and the invariance is lost. Since the holographic duality for the SYK model relies heavily on the reparametrization symmetry, we will restrict our analysis of the linearly dispersive model (3) to temperatures $T_\Lambda < T < J$, and refer to it as Model I throughout. In the next subsection we will introduce the complementary flat-band Model II, for which the SYK phase remains stable down to the deep infrared limit.

## 2.2 Flat-band model

In this section we discuss a variant of the chiral SYK model whose dispersion relation $\epsilon_k$ is shown in Fig. 2, where $k_0 < \pi/a$ defines the width of an (approximate) flat region and $\gamma > 1$ is the dispersive exponent for large momenta $> k_0$. We assume the parameter $k_1/k_0 \ll 1$ to be small. In the following, we will demonstrate that the SYK phase remains stable for this model. Although this spectrum of chiral boundary fermions may seem exotic, it emerges, for example, at the 'soft' edges of gate-confined IQHE samples. Specifically, within the self-consistent electrostatic framework developed by Chklovskii et al. [49, 50], one finds the exponent $\gamma = 3/2$ (see Appendix A).

The set of self-consistent mean-field equations (11-12) for Model II can be solved neglecting the $i\epsilon$–term. Referring for details of this procedure to Appendix B, the scale $T_\Lambda$ plays the same role as in Model I. Specifically, in the weakly dispersive limit ($\epsilon \gg T_\Lambda$), one

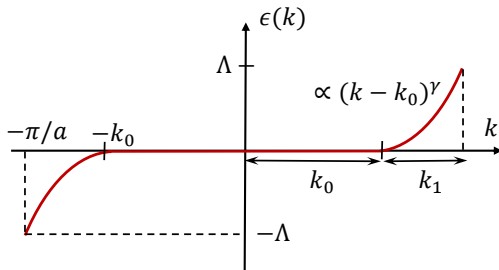

**Figure 2:** Dispersion relation of Majorana edge modes in the chiral SYK model (3) with flat bands (Model II). For the dispersive part one defines $\epsilon_k = \Lambda(\delta k/k_1)^\gamma$, where $\delta k = k - k_0$, so that $\epsilon(\pi/a) = \Lambda \ll J$ in accordance with Eq. (5).

arrives at the SYK-like solutions Eqs. (16) and (17). (One may make these solutions more accurate by substitution $k_0 \to k_0 + k_1$ and $J^2 \to J^2(1+k_1/k_0)^3$, however this refinement will be irrelevant throughout.)

In the strongly dispersive regime (deep IR limit), the Green's function and self-energy receive perturbative corrections of order $O(k_1/k_0)$ to the SYK solutions. Specifically, in the energy domain (at $\epsilon \ll T_\Lambda$), they read

$$\frac{\delta\Sigma_\epsilon}{\Sigma_\epsilon^0} = \sigma_1(\gamma)\left(\frac{k_1}{k_0}\right)\left|\frac{\epsilon J}{\Lambda^2}\right|^{1/2\gamma} + \mathcal{O}(k_1^2/k_0^2), \qquad \frac{\delta G_\epsilon}{G_\epsilon^0} = g_1(\gamma)\left(\frac{k_1}{k_0}\right)\left|\frac{\epsilon J}{\Lambda^2}\right|^{1/2\gamma} + \ldots, \qquad (20)$$

where for details of the derivation and the explicit definitions of functions $\sigma_1(\gamma)$ and $g_1(\gamma)$ we refer the reader to Appendix B. The main point here is that the kinetic energy term $\epsilon_k$ is an *irrelevant* perturbation, including in the deep IR limit. It produces a correction to the leading SYK self-energy $\Sigma_\epsilon^0$ which is parametrically small in $\epsilon$. Physically, the flat-band dispersion of the kinetic energy enhances electron correlations, thereby stabilizing the SYK phase. This is different from Model I, whose linear dispersion $\epsilon_k$ qualitatively changes the nature of the saddle-point at low energies.

In the Fourier conjugate time domain, at large times $\tau \gg 1/T_\Lambda$, we obtain to the same accuracy

$$\frac{\delta\Sigma_\tau}{\Sigma_\tau^0} = \tilde{\sigma}_1(\gamma)\left(\frac{k_1}{k_0}\right)\left|\frac{J}{\Lambda^2\tau}\right|^{1/2\gamma} + \ldots, \qquad \frac{\delta G_\tau^{xx}}{G_\tau^0} = \tilde{g}_1(\gamma)\left(\frac{k_1}{k_0}\right)\left|\frac{J}{\Lambda^2\tau}\right|^{1/2\gamma} + \ldots, \qquad (21)$$

where the numerical coefficients $\tilde{g}_1(\gamma)$ and $\sigma_1(\gamma)$ are discussed in Appendix B.2. The result for the Green's function affords a clear physical interpretation: The scaling dimension of Majoranas in the SYK model is 1/4. The first-order correction to the bare SYK result in Eq. (21) arises from the small fraction of mobile Majoranas, whose scaling dimension crosses over to $\frac{1}{2}\Delta_1 = 1/4 + 1/(4\gamma)$ at times longer than $T_\Lambda^{-1}$.

To better illustrate the last point, one can introduce a subset of fermion-like operators that describe only mobile Majoranas,

$$\lambda_l^j = \left(\frac{a_1}{L}\right)^{1/2}\sum_{|k|=k_0}^{\pi/a}\gamma_k^j e^{ikx_l}, \qquad \frac{1}{2}\{\lambda_l^i, \lambda_{l'}^j\} = \delta_{ll'}\delta^{ij}, \qquad a_1 = \pi/k_1, \qquad (22)$$

and are defined at the coarse-grained positions $x_l = la_1$. Note, that the summation above goes only over a narrow strip of momenta where the kinetic energy $\epsilon_k$ is nonvanishing. By introducing a two-point function, $g_{\tau\tau'}^{xx'}$, for these fields in analogy to Eq. (9), its value at

coinciding spatial points on a level of the mean-field approximation is given by

$$g_\epsilon^{xx} = -\frac{1}{2\pi}\int_{\epsilon_k \neq 0}\frac{dk}{\epsilon_k + \Sigma_\epsilon}. \tag{23}$$

In the weakly dispersive limit, $\epsilon \gg T_\Lambda$, the kinetic energy is negligible compared to the self-energy. Therefore, for short times, $\tau \ll T_\Lambda^{-1}$, a temporal behavior of the correlator $g_\tau^{xx}$ remains of the conventional SYK form. For longer times, $\tau \gg T_\Lambda^{-1}$, corresponding to the strongly dispersive limit, the correlator $g_\tau^{xx}$ crosses over to $\delta G_\tau^{xx}$. This time dependence is summarized as

$$g_\tau^{xx} \sim -\frac{k_1 \mathrm{sgn}(\tau)}{\sqrt{2\pi J}} \times \begin{cases} |\tau|^{-1/2}, & \tau \ll T_\Lambda^{-1}, \\ |\tau|^{-\Delta_1}|T_\Lambda|^{-1/2\gamma}, & \tau \gg T_\Lambda^{-1}, \end{cases} \tag{24}$$

where few numerical factors are omitted for brevity. The two asymptotics here match at the intermediate time scale $\tau \sim T_\Lambda^{-1}$, where a transmutation of the scaling exponent from $1/2$ to $\Delta_1 = 1/2 + 1/2\gamma$ governing the temporal dependence of the correlator occurs. In Sec. 4 we will analyze how quantum fluctuations beyond the mean-field analysis affect the above result.

## 2.3 Reparametrization invariance and soft modes

The concept of reparametrization invariance has been essential in the context of holographic duality between the SYK model and JT gravity. At the level of the $G\Sigma$–functional it manifests itself through a degenerate coset manifold of saddle-points (7) when the symmetry breaking time derivative operator $\partial_\tau$ is neglected. In the chiral SYK model, the kinetic energy $\epsilon_k$ acts as an additional source of symmetry breaking. In this subsection, we argue how global (i.e. independent of $x$) and *adiabatic* reparametrizations of time, $\tau \to t = f(\tau)$, can be employed to construct a coset manifold of approximate saddle-point solutions to the (1+1)-dimensional $G\Sigma$–action, despite the presence of this term. In this way, we show that reparametrizations remain valid soft modes in the problem. We use the latter in the next subsection to derive the AS action from the chiral SYK model.

The reparameterization invariance of the SYK model can be considered as a generalization of invariance under scaling $t = \lambda t'$, to diffeomorphism invariance $t = f(\tau)$. We therefore start our analysis of the higher dimensional model by considering its behavior under scaling. Temporarily focusing on the case of zero temperature, let $G(t; \Lambda)$ and $\Sigma(t; \Lambda)$ be approximate solutions of the zero temperature saddle-point equations ignoring the time derivative operator $\partial_\tau$ and evaluated at coinciding spatial points ($x' = x$). As shown above, this approximation is always valid for the weakly dispersive limit and it also holds in the strongly dispersive limit of the 'flat-band' model. Schematically, the corresponding equations read

$$G(t - t'; \Lambda) = -\left[\left(\epsilon_{\hat{k}} + \Sigma\right)^{-1}\right]^{xx}_{tt'}, \qquad \Sigma(t; \Lambda) = (J^2/k_0^3)\, G^3(t; \Lambda), \tag{25}$$

where the self-energy operator is defined by $\Sigma_{tt'}^{xx'} \equiv \delta_{xx'}\Sigma(t - t'; \Lambda)$, and we explicitly indicate the dependence of the solutions on the kinetic energy scale $\Lambda$. The solutions of equations (25) satisfy the scaling relations

$$G(t/\lambda; \Lambda') = \lambda^{1/2}G(t; \Lambda'/\lambda^{1/2}), \qquad \Sigma(t/\lambda; \Lambda') = \lambda^{3/2}\Sigma(t; \Lambda'/\lambda^{1/2}), \tag{26}$$

with $\lambda \in \mathbb{R}^+$. To see that this condition must hold, consider the action $\mathcal{S} \approx -\int dt\, H$ defined by the Hamiltonian (3), where we substitute Majorana operators by Grassmann variables, $\hat{\gamma} \to \chi$ (all quantum indices are omitted for brevity), and we have dropped the

term $\int dt \chi \dot{\chi}$, in accordance with the rationale of equations (25). The action $\mathcal{S}$ is invariant under the (global) rescaling,

$$t' = t/\lambda, \qquad \chi' = \lambda^{1/4}\chi, \qquad \Lambda' = \lambda^{1/2}\Lambda, \qquad J' = J. \tag{27}$$

In particular, the engineering scaling dimension of the kinetic energy is $[\Lambda] = 1/2$. Considering that the dimension of the Green's function is $[G] = 2 \times [\chi] = 1/2$, we arrive at the scaling symmetries of the approximate mean-field solutions (26). As a sanity check, it is straightforward to verify that the long-time asymptotic of the Green's function and self-energy (21) in the 'flat-band' model, which are valid in the strongly dispersive regime, are consistent with this scaling invariance.

On this basis, we wish to generalize from scaling to reparametrizations $f(\tau)$ of time. Staying close to the rationale previously applied to the local SYK model, we require strict invariance of $(G, \Sigma)$ under the subgroup of $\mathrm{SL}(2, \mathbb{R})$ transformations Eq. (8). We begin by introducing the $\mathrm{SL}(2, \mathbb{R})$-invariant ratio

$$r_{\tau_1 \tau_2} = \frac{f_1 - f_2}{\sqrt{f_1' f_2'}}, \qquad f_i = f(\tau_i), \qquad f_i' = f'(\tau_i), \tag{28}$$

and the rescaled kinetic energy cutoff,

$$\Lambda_{\tau_1 \tau_2} = \frac{\Lambda}{(f_1' f_2')^{1/4}}, \tag{29}$$

where $f' \equiv \partial_\tau f$. Next we define the reparameterized fields

$$\begin{aligned}
\widetilde{\Sigma}_{\tau_1 \tau_2} &= \Sigma(r_{\tau_1 \tau_2}; \Lambda) \equiv f_1'^{3/4} \Sigma(f_1 - f_2; \Lambda_{\tau_1 \tau_2}) f_2'^{3/4}, \tag{30} \\
\widetilde{G}_{\tau_1 \tau_2} &= G(r_{\tau_1 \tau_2}; \Lambda) \equiv f_1'^{1/4} G(f_1 - f_2; \Lambda_{\tau_1 \tau_2}) f_2'^{1/4}, \tag{31}
\end{aligned}$$

as generalizations of the scaling transformations Eqs. (26). The correspondence between these definitions follows from the identifications $t \to f_1 - f_2$ and $\lambda \to \sqrt{f_1' f_2'}$ together with $\Lambda' \to \Lambda$. We note that the fields $\widetilde{G}$ and $\widetilde{\Sigma}$ remain invariant under the restricted class of $\mathrm{SL}(2, \mathbb{R})$ transformations Eqs. (8).

It remains to be shown that the reparameterized fields are approximate solutions of the stationary phase equations Eq. (11), provided $(G, \Sigma)$ are. With Eqs. (26) and (30), the second of these, $\widetilde{\Sigma} = (J^2/k_0^3)\widetilde{G}^3$, is manifestly satisfied. However, the first requires substantially more discussion. In the following, we analyze this equation to which the pair $(\widetilde{G}, \widetilde{\Sigma})$ is an approximate solution. Our discussion will also introduce various concepts and definitions which will play a key role in our subsequent derivation of the fluctuation action.

We begin by noting that the self energy $\widetilde{\Sigma}^{xx} \propto (\widetilde{G}^{xx})^3$ is spatially local, and translationally invariant on average. It therefore does not exhibit coordinate dependence, and we denote it by $\widetilde{\Sigma}$ throughout. Turning to the more involved first equation in (25), consider a formal solution $\widetilde{\mathcal{G}}_{\tau_1 \tau_2}^{x_1 x_2}$ evaluated for the self energy $\widetilde{\Sigma}$. Turning to Fourier space, it reads

$$-\int d\tau_2 \left( \epsilon_k \delta_{\tau_1 \tau_2} + \widetilde{\Sigma}_{\tau_1 \tau_2} \right) \widetilde{\mathcal{G}}_{\tau_2 \tau_3}(k) = \delta_{\tau_1 \tau_3}. \tag{32}$$

### 2.3.1 Linear transformation in time space

In the following, it will be helpful to think of Eq. (32) as a 'matrix equation', and to consider the transformation $t \to \tau = F(t)$ as a change of basis. Referring to Appendix C for the detailed formulation of this picture in the language of linear algebra, we here note that the

transformation between the two representations acts on general bilocal operators $O_{\tau_1 \tau_2}$ with scaling dimensions $\Delta$ via a (non-unitary) linear map $\mathcal{M}_\Delta$ as

$$\mathcal{M}_\Delta : O_{\tau_1 \tau_2} \;\mapsto\; \overline{O}_{t_1 t_2} = F_1'^{\Delta/2} O_{\tau_1 \tau_2} F_2'^{\Delta/2}, \quad \tau_i = F(t_i), \tag{33}$$

where $F$ is the inverse of the map $f$, $F(f(\tau)) = \tau$, and we have abbreviated $F_i' = F'(t_i)$. We aim to apply $\mathcal{M}_\Delta$ in Eq. (32), keeping in mind the scaling dimensions $\Delta_\Sigma = 3/2$ and $\Delta_G = 1/2$. As seen from (30-31), this transformation effectively eliminates the conformal factors $f'^{\Delta/2}$ and then changes the time frame from $\tau$ to $t$. To conveniently describe its action, we define the function,

$$b_t \equiv 1/\sqrt{F_t'} \equiv \left.\sqrt{f_\tau'}\right|_{\tau = F(t)}, \tag{34}$$

which will be used extensively throughout. Note that $F_t' > 0$, making the square root well defined. It is straightforward to verify that the transformed representation of the Dyson equation (32) reads

$$-\int dt_2 (\epsilon_k \, b_{t_1}^{-1} \delta_{t_1 t_2} + \overline{\Sigma}_{t_1 t_2}) \overline{\mathcal{G}}_{t_2 t_3}(k) = \delta_{t_1 t_3}, \tag{35}$$

where the combination of Eqs. (30) and (33) implies

$$\overline{\Sigma}_{t_1 t_2} = \Sigma(t_1 - t_2; \Lambda_{t_1 t_2}), \qquad \Lambda_{t_1 t_2} = \frac{\Lambda}{(b_{t_1} b_{t_2})^{1/2}}. \tag{36}$$

At this point, the rationale behind the $\mathcal{M}$ - transformation becomes evident: to leading order $\overline{\Sigma}_{t_1 t_2}$ coincides with the SYK self-energy (17), while the dependence on $\Lambda_{t_1 t_2}$ is a next-order effect due to the presence of a kinetic energy term $\epsilon_k$.

### 2.3.2 Wigner-Moyal expansion

One can solve Eq. (35) in an adiabatic approximation, assuming the field $b_t$ exhibits 'slow' dependence on the relative time scale $t = t_1 - t_2$. This idea can be formalized using Wigner symbols and the Moyal expansion, which are widely employed to represent similarly structured equations in many-body physics [51, 52]. We define the Wigner transform $O_\epsilon(s)$ of the operator $\overline{O}_{t_1 t_2}$ as

$$\overline{O}_\epsilon(s) = \int dt \, e^{i\epsilon t} \overline{O}_{s+\frac{t}{2}, s-\frac{t}{2}}, \tag{37}$$

where $s = (t_1 + t_2)/2$ is the 'center of mass' time, while the energy $\epsilon$ is conjugate to relative time $t$. For the self-energy (36), we use the local approximation by substituting $\Lambda_{t_1 t_2} \to \Lambda/b_s$. This translates into its Wigner symbol $\overline{\Sigma}_\epsilon(s) \simeq \Sigma(\epsilon; \Lambda/b_s)$, which is a conventional Fourier transform of the mean-field self-energy $\Sigma(t; \Lambda)$ over time. Relying on the above approximation, we then introduce the Wigner symbol of an effective Hamiltonian,

$$h_{\epsilon,k}(s) \equiv \epsilon_k/b_s + \Sigma(\epsilon; \Lambda/b_s), \tag{38}$$

such that the Dyson equation reads: $-h_{\epsilon,k}(s) \star \overline{\mathcal{G}}_{\epsilon,k}(s) = 1$, where '$\star$' denotes the Moyal product[4]. This latter equation can be solved to leading Moyal order, yielding

$$\overline{\mathcal{G}}_{\epsilon,k}(s) = -1/h_{\epsilon,k}(s) + \mathcal{O}(\hbar^2), \tag{39}$$

---

[4]If $A_\epsilon(s)$ and $B_\epsilon(s)$ are Wigner symbols of $\hat{A}_{t_1 t_2}$ and $\hat{B}_{t_1 t_2}$, the Moyal product $A_\epsilon(s) \star B_\epsilon(s)$ is the Wigner symbol of $(AB)_{t_1 t_2}$. It affords the semiclassical expansion $(A \star B)_\epsilon(s) = A_\epsilon(s) e^{-\frac{i\hbar}{2}(\overleftarrow{\partial_s}\overrightarrow{\partial_\epsilon} - \overleftarrow{\partial_\epsilon}\overrightarrow{\partial_s})} B_\epsilon(s) = A_\epsilon(s)B_\epsilon(s) - \frac{i\hbar}{2}A_\epsilon(s)(\overleftarrow{\partial_s}\overrightarrow{\partial_\epsilon} - \overleftarrow{\partial_\epsilon}\overrightarrow{\partial_s})B_\epsilon(s) + \mathcal{O}(\hbar^2)$.

where the $\mathcal{O}(\hbar^2)$–term symbolically denotes possible 2nd order gradient corrections[5] of order $b'^2$ and $bb''$, see subsection D.2 below for the detailed discussion of this point. We also note that the rescaled kinetic energy in the effective Hamiltonian can be written as

$$\epsilon_k/b_s = (\Lambda/b_s)(k/k_1)^\gamma = \epsilon_k\big|_{\Lambda \to \Lambda/b_s}. \tag{40}$$

As a result, the propagator at coinciding spatial points affords the approximate representation

$$\overline{\mathcal{G}}^{xx}_{t_1 t_2} = \int_{\epsilon,k} \overline{\mathcal{G}}_{\epsilon,k}(s) e^{-i\epsilon(t_1-t_2)} \simeq -\int_{\epsilon,k} \frac{e^{-i\epsilon(t_1-t_2)}}{\epsilon_k/b_s + \Sigma(\epsilon;\Lambda/b_s)} = G(t_1 - t_2; \Lambda/b_s), \tag{41}$$

where $s = \frac{t_1+t_2}{2}$ as before, and we have abbreviated $\int_{\epsilon,k} \equiv \int dk\, d\epsilon/(2\pi)^2$ for all subsequent formulae. We finally transform the rescaled kinetic energy $\Lambda/b_s$ back to its exact value $\Lambda_{t_1 t_2}$ (the accuracy of this substitution is again of order $b'^2$), then apply the inverse mapping $\mathcal{M}^{-1}_{1/2}$, cf. definition (33), to both sides of Eq. (41), and use the scaling identities (31) for the mean-field Green's function $G(t;\Lambda)$ to arrive at the result

$$\mathcal{G}^{xx}_{\tau_1 \tau_2} = \widetilde{G}_{\tau_1 \tau_2} + \mathcal{O}(\hbar^2), \tag{42}$$

i.e. an equation stating an identification of the solution to the mean field equations with the reparameterized Green function $\widetilde{G}$ in the semiclassical limit of slowly varying reparameterization transformations. Corrections, symbolically indicated as $\mathcal{O}(\hbar^2)$, involve higher order derivative acting on $f$, which are ignored in the present analysis (but carefully analyzed in Subsection D.2).

To summarize, we have defined the reparameterized saddle-point $(\widetilde{G}, \widetilde{\Sigma})$ based on the approximate stationary solution of the $G\Sigma$-functional, excluding the $\partial_\tau$ operator. The $f$-dependent fields solve the second equation in (25) exactly, and the first approximately. These fields are solutions on the full coset manifold $\mathrm{Diff}(S^1)/\mathrm{SL}(2,\mathbb{R})$ of reparameterizations provided the latter fluctuate slowly. In this way, we have identified reparametrizations $f(\tau,x)$ as the relevant soft modes of the chiral SYK model.

## 2.4   Gradient expansion

In this section, we will expand the $G\Sigma$-action in slow spatial and temporal fluctuations of the reparametrizations $f(\tau,x)$ to derive the Alekseev-Shatashvili action (2). In the process, we will determine the coupling constants $C$ and $u$ of the theory to leading logarithmic accuracy. Before embarking on the actual derivation, let us summarize our main results.

For Model I with dispersion $\epsilon_k = v_0 k$, and at intermediate temperatures, the two coupling constants evaluate to

$$C = \left(\frac{3\pi}{4}\right) \frac{N\Lambda}{J} \ln \frac{J}{T_\Lambda}, \qquad u = 2\Lambda/k_0, \qquad T_\Lambda \ll T \ll \sqrt{J\Lambda}, \tag{43}$$

where our expansion in derivatives requires $C \gg 1$, which in turn requires $N \gg 1$. For lower temperatures Model I lacks a holographic correspondence to AdS$_3$. For Model II, the coupling constants in the temperature regime above are the same, Eq. (43). For lower temperatures, $C$ does not change, while $u$ becomes temperature dependent as

$$u(T) = \frac{u_0}{\ln(J/T)}, \qquad u_0 = \frac{2\Lambda}{k_0} \ln \frac{J}{T_\Lambda}, \qquad T \ll T_\Lambda. \tag{44}$$

---

[5]Indeed, if $G_\epsilon(s) = -1/h_\epsilon(s)$, then $(G \star h)_\epsilon(s) = -1 - \frac{i\hbar}{2}(\partial_s G\, \partial_\epsilon h - \partial_\epsilon G\, \partial_s h) + \mathcal{O}(\hbar^2) = -1 + \mathcal{O}(\hbar^2)$.

In the logarithmic $T$-dependence manifests a temperature scaling, as described by the flow equation

$$\frac{d\ln u}{dl} = -\frac{u}{u_0}, \qquad l = \ln(J/T), \tag{45}$$

subject to the initial condition $u(T_\Lambda) = 2\Lambda/k_0$ at the boundary to the intermediate temperature regime. The diminishing of the velocity $u$ at low temperatures is a consequence of the vanishing group velocity, $v_k = \partial_k \epsilon_k$, at low momenta.

To set the stage for our subsequent derivations, let us consider the AS action at zero temperature,

$$S_\pm^\infty[f] = \frac{C}{24\pi} \int\limits_{-\infty}^{+\infty} d\tau \int\limits_0^L dx \frac{f'' \partial_\pm f'}{f'^2}, \qquad \partial_\pm = \frac{1}{2}(u^{-1}\partial_\tau \pm i\partial_x), \tag{46}$$

as the $\beta \to \infty$ limit of the finite temperature action $S_\pm \equiv S_\pm^\beta$, Eq. (2), where imaginary time is compactified to a circle. The finite temperature generalization is obtained from Eq. (46) by a substitution $f(\tau, x) \to \tan(\pi f(\tau, x)/\beta)$, i.e.

$$S_\pm^\beta[f] = S_\pm^\infty[\tan(\pi f/\beta)], \tag{47}$$

where a restriction $\tau \in [0, \beta]$ is implied on both sides of the relation. In the following, we will focus on the derivation of (46), keeping in mind that its finite temperature generalization may be realized in a secondary step, by application of Eq. (47). Our starting point is the $G\Sigma$-functional,

$$S[f] = -\frac{N}{2}\left[ \text{tr}\ln\left(\partial_\tau + \epsilon_{\hat{k}} + \widetilde{\Sigma}\right) + \frac{J^2}{4k_0^3}\int d^2\tau dx\, (\widetilde{G}_{\tau_1\tau_2}^{xx})^4 + \text{Tr}(\widetilde{G}\widetilde{\Sigma}) \right], \tag{48}$$

where the self-energy and Green's function are reparametrized according to (30) and (31), and we have promoted $f(\tau, x)$ to include space dependence. Importantly, the starting action (48) is locally $\text{SL}(2,\mathbb{R})$–invariant. This is a direct consequence of the invariance of the fields $\widetilde{G}$ and $\widetilde{\Sigma}$ under the transformations (8), where the coefficients $a, \ldots, d$ may depend on position, $a = a_x$, etc. (As discussed in the foregoing section, the invariance of the fields follows from the definitions (30-31) indicating that $\widetilde{G}$ and $\widetilde{\Sigma}$ are expressed through the $\text{SL}(2,\mathbb{R})$–invariant ratio (28).) Consequently, the gradient expansion of action (48) must also preserve this symmetry, which will turn out to be a highly nontrivial consistency check. In fact, the AS action will emerge as the minimal local functional respecting this coset symmetry.

We also note that the AS action (46) contains real and imaginary parts: $S_\pm^\infty[f] = S_R[f] \pm iS_I[f]$, where the real part is the position dependent Schwarzian (up to a boundary term), while the imaginary part is the kinetic term containing mixed spatial and time derivatives of $f(\tau, x)$. In the following, it will be useful to consider these two terms employing the field $b$ (34) as an independent variable. More precisely, we generalize the definition (34) to include spatial dependence [6],

$$b_{t,x} \overset{\text{def.}}{=} 1/\sqrt{F_{t,x}'} \equiv \sqrt{f_{\tau,x}'}\Big|_{\tau=F(t,x)}, \tag{49}$$

where $f' = \partial_\tau f$, $F' = \partial_t F$ and the relation $t = f_x(F_x(t))$ is defined locally. On changing in Eq. (46) the time integration variable as $t = f(\tau, x)$ at given $x$, it is straightforward to

---

[6]In what follows an abridged notation is used: $F_{t,x} \equiv F(t,x)$, and the same for all other functions of time and space, e.g. $b_{t,x} \equiv b(t,x)$ and so on.

verify that the real part becomes Gaussian (the integration measure, however, is not flat in terms of the field $b$, i.e. the complexity of the theory is now hiding there), while the imaginary part assumes the form of a 'cubic' vertex:

$$S_\pm^{AS}[b] = \frac{C}{12\pi u} \int_{t,x} b'^2 \pm \frac{iC}{24\pi} \int_{t,x} \partial_x F \left( b'' b - b'^2 \right), \quad b' \equiv \partial_t b. \tag{50}$$

The equivalence between this representation and Eq. (46) follows from a few integrations by parts in time, assuming that the boundaries $\pm\infty$ of the zero temperature time domain $t \in \mathbb{R}$ are identified. In the following, we will use the pair of fields $(b, \partial_x F)$ to carry out the gradient expansion, and to derive the AS action as in Eq. (50). Below, we outline the main steps of this procedure, relegating the majority of technical details to Appendices.

The starting reparametrized $G\Sigma$–action (48) requires some care, as its first term is expressed via a functional determinant of the operator $\mathcal{D} = \partial_\tau + \epsilon_{\hat{k}} + \widetilde{\Sigma}$. We have chosen to regularize it as

$$S_{\text{reg}}[f] = -\frac{N}{2} \ln\det\mathcal{D} + \frac{N}{2} \ln\det\widetilde{\Sigma}^0 - \frac{N}{2} I[\widetilde{G}, \widetilde{\Sigma}], \tag{51}$$

where $\widetilde{\Sigma}^0$ is the reparametrized self-energy of the original SYK solution (7) and $I[\widetilde{G}, \widetilde{\Sigma}]$ adds the remaining two pieces of the $G\Sigma$–action:

$$I[G, \Sigma] = \frac{J^2}{4k_0^3} \int d^2\tau\, dx\, [G_{\tau_1\tau_2}^{xx}]^4 + \text{Tr}\,(G\Sigma). \tag{52}$$

A closer inspection reveals that a regulator above contributes only an inessential $f$-independent constant[7], namely $\text{tr}\ln\widetilde{\Sigma}^0 = \text{tr}\ln\Sigma^0$. To keep the notation slim, we will suppress its presence in much of our discussion below.

To prepare the gradient expansion, we will first shuffle the time reparametrizations from the 'large' self-energy to the 'small' symmetry-breaking terms, $\partial_\tau$ and $\epsilon_{\hat{k}}$, in the operator $\mathcal{D}$. To do so, we introduce a generalized variant of the transformation (33),

$$\mathcal{M}_\Delta : O_{\tau_1\tau_2}^{x_1x_2} \quad \mapsto \quad \overline{O}_{t_1t_2}^{x_1x_2} = F_1'^{\Delta/2} O_{\tau_1\tau_2}^{x_1x_2} F_2'^{\Delta/2}, \quad \tau_i = F(t_i, x_i), \tag{53}$$

defined to be applicable to spatially non-local operators $O_{\tau_1\tau_2}^{x_1x_2}$ with scaling dimension $\Delta$ and diffeomorphisms $F$ carrying space dimension (in Eq. (53), $F_i' \equiv F'(t_i, x_i)$, as usual). We apply this transformation to each term in the action (51) individually, with account for the respective scaling dimensions $\Delta_\Sigma = 3/2$ and $\Delta_G = 1/2$, to obtain

$$S_{\text{reg}}[f] = -\frac{N}{2} \ln\det\overline{\mathcal{D}} - \frac{N}{2} I[\overline{G}, \overline{\Sigma}]. \tag{54}$$

In particular, the action $I$ remains invariant, as follows from the change of differentials, $dt_i = f_i'\, d\tau_i$ in Eq. (52). Here, the self-energy $\overline{\Sigma}$ is given by Eq. (36), with the field $b(t, x)$ being position dependent, and an analogous expression applies to the Green's function $\overline{G}$. At the same time, the differential operator $\mathcal{D}$ transforms with a scaling dimension $\Delta = 3/2$, and after the mapping (53) it becomes

$$\mathcal{M}_{3/2} : \mathcal{D} \mapsto \overline{\mathcal{D}} = \rho + j + \overline{\Sigma}, \tag{55}$$

---

[7]Indeed, the variation of the action $S_0[f] := -(N/2)\ln\det\widetilde{\Sigma}^0 - (N/2)I[\widetilde{G}^0, \widetilde{\Sigma}^0]$ over $f$ vanishes, if $(\widetilde{G}^0, \widetilde{\Sigma}^0)$ is the reparametrized SYK saddle-point. Therefore, one finds that $S_0[f] = -(N/2)\ln\det\Sigma^0 - (N/2)I[G^0, \Sigma^0]$ is some constant. The last piece here, $I[\widetilde{G}^0, \widetilde{\Sigma}^0] \equiv I[G^0, \Sigma^0]$, is explicitly reparametrization invariant. Hence the determinant is also invariant just by itself.

where $\rho$ and $j$ denote the transformed time derivative operator $\partial_\tau$ and the kinetic energy $\epsilon_{\hat{k}}$, respectively. It will become clear shortly, that the former two afford an interpretation as energy density and heat current, respectively.

Referring to Appendix D.1 for details, the operator $\rho$ is diagonal in space, i.e. $\rho^{xx'} = \delta^{xx'}\rho_x$, where $\rho_x \equiv \rho$ is a first order differential operator in time,

$$\rho = \frac{1}{2}\left(b\overrightarrow{\partial_t} - \overleftarrow{\partial_t}b\right), \tag{56}$$

with parametric space dependence from $b = b(x, t)$. Its action on smooth functions $g_t$ and $h_t$ is defined by the matrix elements

$$\langle g|\rho|h\rangle = \int dt\, b(g(\partial_t h) - (\partial_t g)h). \tag{57}$$

The second operator, $j$, requires a bit more work. Unlike $\rho$, it has a non-trivial structure in both position and time. To define it, we introduce its Wigner symbol in position-momentum space as $j_k(x) = \int dy\, e^{-iky} j^{x+\frac{y}{2}, x-\frac{y}{2}}$, cf. Eq. (37). In Appendix D.1 we show that the lowest order Moyal expansion, of this Wigner symbol assumes the form

$$j_k(x) = \epsilon_k/b + \frac{i}{2}\partial_k\epsilon_k \times \left(b\partial_x F\overrightarrow{\partial_t} - \overleftarrow{\partial_t}b\partial_x F\right) \equiv j_k^0(x) + j_k^1(x), \tag{58}$$

where $b = b(t, x)$ as above. We have also split $j_k(x)$ in two parts, containing zero and one derivative operator, respectively. Similarly to Eq. (57) above, the matrix element of the latter is defined as

$$\langle g|j_k^1(x)|h\rangle = \frac{i}{2}\partial_k\epsilon_k \int dt\, \partial_x F\, b(g(\partial_t h) - (\partial_t g)h). \tag{59}$$

Collecting terms, to lowest non-trivial order in gradients, the position-momentum Wigner symbol of the operator (55) is given by the expansion:

$$\overline{\mathcal{D}}_k(x) = \rho + j_k^1(x) + h_k(x), \qquad h_k(x) = \epsilon_k/b + \overline{\Sigma}, \tag{60}$$

where $h_k(x)$ is the effective Hamiltonian introduced previously in Eq. (38).

With the result (60) at hand, we are now in a position to organize the gradient expansion of the action (54) around the approximate saddle-point solution. For that purpose, we define the propagator $\overline{\mathcal{G}}$ with a corresponding position-momentum Wigner symbol $\overline{\mathcal{G}}_k(x)$, which obeys the Dyson equation $h_k(x) \star \overline{\mathcal{G}}_k(x) = -1$. This is a generalization of our previous definition (35) to the case of spatially inhomogeneous reparametrizations $f = f(\tau, x)$. One then rewrites the regularized action (54) in an equivalent form by splitting it into the fluctuation contribution,

$$S_{\mathrm{fl}}[f] = -\frac{N}{2}\mathrm{tr}\ln\left(1 - (\rho + j^1)\overline{\mathcal{G}}\right) \tag{61}$$

and one originating from the approximate saddle-point,

$$S_*[\overline{G}, \overline{\Sigma}] = -\frac{N}{2}\mathrm{tr}\ln\left(j^0 + \overline{\Sigma}\right) - \frac{N}{2}I[\overline{G}, \overline{\Sigma}], \tag{62}$$

so that the sum of these two actions gives back $S_{\mathrm{reg}}[f]$. In Appendix D.3 we verify that the second piece, $S_*[\overline{G}, \overline{\Sigma}]$, does not contribute to the AS action (50). This follows from the fact that $(\overline{G}, \overline{\Sigma})$ defines a manifold of approximate saddle-point solutions of the functional $S_*[G, \Sigma]$, parametrized by $f$. On the other hand, the gradient expansion of the first piece, Eq. (61), describes the effects of fluctuations beyond the mean-field approximation.

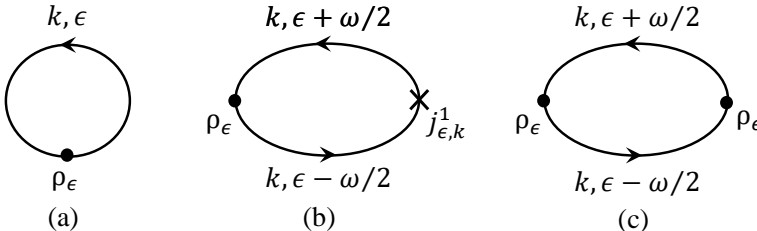

$k, \epsilon$          $k, \epsilon + \omega/2$          $k, \epsilon + \omega/2$

$\rho_\epsilon$    $j^1_{\epsilon,k}$    $\rho_\epsilon$    $\rho_\epsilon$

$\rho_\epsilon$        $k, \epsilon - \omega/2$        $k, \epsilon - \omega/2$

(a)            (b)            (c)

**Figure 3:** Diagrams of the 2nd order gradient expansion contributing to the AS action.

The second-order expansion in $\rho$ and $j^1$ generates several terms, where those contributing to the AS action (50) are given by

$$S_{\mathrm{fl}}[f] \to \frac{N}{2}\mathrm{tr}(\rho\overline{\mathcal{G}}) + \frac{N}{2}\mathrm{tr}(j^1\overline{\mathcal{G}}\rho\overline{\mathcal{G}}) + \frac{N}{4}\mathrm{tr}(\rho\overline{\mathcal{G}}\rho\overline{\mathcal{G}}) \equiv S_\rho[f] + S_{j\rho}[f] + S_{\rho\rho}[f]. \qquad (63)$$

Referring to Appendix D for details, we note that the sum of the two terms, $S_\rho[f] + S_{j\rho}[f]$, defines the imaginary part of the AS action $S^{\mathrm{AS}}_\pm$, expressed as a functional of the pair of fields $(b, \partial_x F)$. In particular, these two terms determine the central charge (43). The remaining term, $S_{\rho\rho}[f]$, provides the real part, and it assumes the form of the Schwarzian action when expressed in terms of $f(\tau, x)$. It sets a ratio $C/u$ and gives the scale of velocity (44).

The physical meaning of the terms contributing to the gradient expansion (63) can be interpreted as follows. To leading order, the operators $\rho$ and $j^1$ have the Wigner symbols

$$\rho_{\epsilon,k} = -ib\epsilon + \dots, \qquad j^1_{\epsilon,k} = (b\partial_x F) \cdot \epsilon\partial_k\epsilon_k + \dots, \qquad (64)$$

identifying them as energy density and heat current operators, the latter being proportional to the group velocity $v_k = \partial_k\epsilon_k$. Accordingly, the individual terms in the expansion (63) describe correlation functions of $\rho_{\epsilon,k}$ and $j^1_{\epsilon,k}$, whose Feynman diagrams are shown in Fig. 3. A low frequency expansion of these correlation functions in $\omega \ll \epsilon$ up to $\mathcal{O}(\omega^2)$ yields terms of the type $b'^2$ and $bb''$ in the action $S^{\mathrm{AS}}_\pm[b]$, see Eq. (50). At the same time, the integration region over the energy $\epsilon$ is wide, $T_\Lambda < |\epsilon| < J$, leading to the appearance of log-prefactors in the coupling constants $C$ and $u$. In this reading, the Schwarzian action is a density-density correlator, see Fig. 3(c). Likewise, the central charge and the kinematic part of the AS action are defined by a current-density correlator, Fig. 3(b).

For completeness, we mention that the gradient expansion contains two extra terms, not contributing to the AS action. These are a first-order term, $S_j[f]$, which reduces to a constant when subjected to our regularization scheme with a reparametrization dependent cut-off. There also is a contribution $S_{jj}[f]$ containing two spatial derivatives $\propto (\partial_x F)^2$. The latter does not comply with $\mathrm{SL}(2, \mathbb{R})$ invariance, and we attribute it to the Jacobian of the transformation from initial to rotated Majoranas under the mapping (53). Finally, for Model II the quantum fluctuations of $f(x, \tau)$ may compromise the SYK saddle point at temperature scale $T_* \ll T_\Lambda$, which is defined by the competition of the small parameter $k_1/k_0 \ll 1$ and the large ratio $J/T_\Lambda$, see Eq. (D.34) for the explicit definition. The latter limits a holographic duality of Model II to the range of temperatures $T_* < T < J$. For further details we again refer to Appendix D.

To conclude this section, we note that the chiral SYK Hamiltonian (3), which features right-moving Majorana fermions with $\partial_k\epsilon_k \geq 0$, gives rise to a single copy of the AS action, namely to $S_-[f]$. However, if the underlying topological insulator has either a Hall bar or Corbino geometry, the opposite boundaries will host Majorana edge states with opposite chiralities. In this setting, two independent copies of the AS action, $S_\pm[f]$, emerge, which is a scenario that takes place in a holographic dual of this model discussed in the next section.

# 3 The AS action from gravity

In this section, we consider the theory describing fluctuations around the Euclidean BTZ black hole and provide the direct reduction of this theory to the AS action. This computation gets us in a position to match coupling constants as well as time reparametrizations emerging in the boundary and bulk framework, respectively. We begin with quick reviews of the phase space of AdS$_3$ gravity in subsection 3.1 and the Chern-Simons formulation of three-dimensional gravity in 3.2. Expert readers can skip these subsections and directly turn to 3.3. There, we discuss the Euclidean BTZ black hole solution in the Chern-Simons formulation for a particular choice of analytic continuation. Finally, subsection 3.4 derives the above-mentioned link to AS theory by quantizing gravity on top of the Euclidean BTZ background.

## 3.1 The phase space of asymptotically AdS$_3$ gravity

To understand the role of the AS action in the context of asymptotically AdS$_3$ spacetimes, we start by reviewing some properties of three-dimensional pure gravity in the metric formalism. The most important difference to gravity in higher dimensions is the topological nature of the three-dimensional theory, meaning that it has no local degrees of freedom [53]. For example, mass sources spacetime curvature locally, but deformations of the metric away from the source are pure gauge. This, however, does not mean that the theory without matter is trivial; its solutions can have non-trivial global properties like horizons or mass and angular momentum, as we will briefly discuss.

The Einstein-Hilbert action for three-dimensional pure gravity with negative cosmological constant $\Lambda < 0$ is given by

$$S = \frac{1}{16\pi G} \int_{\mathcal{M}} \mathrm{d}^3 x \sqrt{-g} (R - 2\Lambda). \tag{65}$$

It describes what we will refer to as "AdS$_3$ gravity". Here, $G$ is Newton's constant, and $g_{\mu\nu}$ is the metric on the three-manifold $\mathcal{M}$ with determinant $g$ and associated curvature scalar $R$. The variation of this action yields field equations that express constant negative curvature, $R_{\mu\nu} = 2\Lambda g_{\mu\nu}$ or $R = -6/l^2$, where in the latter equation we introduced the AdS curvature radius $l$ by writing the cosmological constant of dimension mass squared as $\Lambda = -1/l^2$. As a consequence, the spacetime 'looks the same everywhere', a high level of symmetry manifesting itself in a SO(2, 2)-isometry: all solutions are locally AdS$_3$. This fact in turn implies that all solutions with an inequivalent causal structure must be obtainable from a 'basic solution' via identifications under the action of a discrete group $\Gamma$ [29, 53]. This basic solution is given by global AdS$_3$ with the metric

$$\mathrm{d}s^2 = -\left(1 + \frac{r^2}{l^2}\right) \mathrm{d}t^2 + \frac{1}{(1 + \frac{r^2}{l^2})} \mathrm{d}r^2 + r^2 \mathrm{d}\phi^2, \tag{66}$$

where the coordinate ranges are $t \in \mathbb{R}$, $\phi \in (0, 2\pi)$ and $r > 0$.

Three-dimensional holography is generically considered subject to Dirichlet boundary conditions. These are specified in terms of an induced metric on the asymptotic boundary $\mathcal{M}_\infty$ at spacelike infinity $r \to \infty$. In this limit, the metric should reduce to

$$\mathrm{d}s^2_\infty = \frac{l^2}{r^2} \mathrm{d}r^2 + \frac{r^2}{l^2} dx^+ dx^-, \tag{67}$$

where $dx^+ dx^-$ describes the flat two-dimensional metric, in light-cone coordinates $x^\pm = l\phi \pm t$, of the conformal boundary.

The problem famously allows for a black hole solution, which is called the BTZ black hole [28]. It is labeled by two charges, namely its mass $M$ and angular momentum $J$ (with $|J| < Ml$ to avoid naked singularities). The non-rotating BTZ solution is

$$\mathrm{d}s^2 = -\frac{\xi(r)^2}{l^2}\mathrm{d}t^2 + \frac{l^2}{\xi(r)^2}\mathrm{d}r^2 + r^2\mathrm{d}\phi^2, \qquad \xi(r) \equiv \sqrt{r^2 - r_+^2}, \tag{68}$$

with mass $M = r_+^2/(8Gl^2)$ a function of the location $r_+$ of the horizon, and coordinate ranges $t \in \mathbb{R}$, $\phi \in (0, 2\pi)$ and $r > r_+$. This geometry will play an important role throughout. The surface area of the horizon, $2\pi r_+$, and the periodicity $\beta = 2\pi l^2/r_+$ of Euclidean time $\tau = it$ determine thermodynamic properties of the black hole: the Bekenstein-Hawking entropy $S_{BH} = 2\pi r_+/(4G)$ and the Hawking temperature $T = 1/\beta$.

The most general solution satisfying the Dirichlet boundary conditions is the Banados solution [54]

$$ds^2 = \frac{l^2}{r^2}dr^2 + \frac{r^2}{l^2}\left(dx^+ + \frac{l^2}{r^2}L(x^-)dx^-\right)\left(dx^- + \frac{l^2}{r^2}\bar{L}(x^+)dx^+\right). \tag{69}$$

It is characterized by functions $L$ and $\bar{L}$, which quantify the gravitational energy of the solution (e.g. the BTZ black hole mass $M = (L+\bar{L})/(4G)$ and angular momentum $J = (L-\bar{L})/(4G)$ for constant horizon-dependent values of $L$ and $\bar{L}$ in that case). These geometries are referred to as 'asymptotically AdS'.

There is a class of diffeomorphisms $\xi$ which reach the boundary and change the near-boundary $L$ and $\bar{L}$, but only in such a way that they obey the prescribed 'asymptotically AdS' fall-off behavior of the metric components $g_{\mu\nu}(r)$ as $r \to \infty$ (namely $L \to L'$ and $\bar{L} \to \bar{L}'$ but the solution remains of the Banados form, see e.g. [55]). Such diffeomorphisms become physical and obey the algebra of an asymptotic symmetry group. In this case, the Virasoro algebra with Brown-Henneaux central charge $C = 3l/2G$ [56]. Indeed the Brown-Henneaux diffeomorphisms $\xi$ reduce on the boundary to conformal transformations of the light-cone coordinates $x^\pm \to f_\pm(x^\pm)$. This is the famous Brown-Henneaux result that identifies the role of two-dimensional conformal symmetry in (classical) AdS$_3$ gravity, interpreted today as a precursor of AdS/CFT.

Once a global structure of spacetime (e.g. global AdS or BTZ) has been fixed, the perturbations around it are asymptotically realized degrees of freedom referred to as 'boundary gravitons'.

The situation parallels that realized in topological field theory: While topological field theory on a space without boundary is gauge invariant, this feature is lost in the presence of a boundary $\mathcal{M}_\infty$. Gauge transformations which do not reduce to the identity at the boundary are governed by an effective boundary action on this surface, which upon quantization defines an infinite-dimensional Hilbert space. In the gravitational context, the role of the latter is taken by the AS action [24, 25] [26, 27, 39, 57], whose derivation we proceed to discuss in the following sections. For this, we make use of the CS gauge theory description of AdS$_3$ gravity, which makes the above analogy quite explicit.

## 3.2 Three-dimensional gravity as Chern-Simons theory

The Einstein-Hilbert action has long been known to have an equivalent Chern-Simons description [58, 59]. We present a short review of this equivalence for (65) in order to set notation, and refer the interested reader to the standard review [60] for more details.

First, one rewrites the Einstein-Hilbert action (65) in a first order form, meaning in terms of degrees of freedom that appear in the action with first order derivatives rather than second order ones (as for the metric in the metric formulation). These degrees of

freedom are the frame field or vielbein, in this case dreibein, $e^a$ and spin connection $\omega^a{}_b$, with Latin letters for the frame index or 'internal index' $a = 0, 1, 2$. The vielbein, sometimes referred to as the 'square root of the metric', is the object that expresses the existence of a local inertial frame $\eta_{ab}$ in each point of the manifold, $ds^2 = \eta_{ab}e^a e^b$. It is a one-form $e^a = e^a_\mu dx^\mu$, whose coefficients $e^a_\mu$ allow transformation between spacetime indices $\mu$ and frame indices $a$,

$$g_{\mu\nu} = e^a_\mu e^b_\nu \eta_{ab}, \qquad \eta_{ab} = g_{\mu\nu}e^\mu_a e^\nu_b = \mathrm{diag}(-1, 1, 1). \tag{70}$$

The spin connection $\omega^a{}_b$ is a one-form $\omega^a{}_b = \omega^a{}_{b\mu}dx^\mu$, with $\omega^a{}_{b\mu}$ taking over the role of the Christoffel symbols in the metric formulation, such as in the definition of a covariant derivative. In terms of these variables, the Einstein-Hilbert action (65) becomes

$$S = -\frac{1}{16\pi G} \int \epsilon_{abc} e^a \wedge \left( d\omega^{bc} + \omega^b{}_d \wedge \omega^{dc} + \frac{1}{3l^2}e^b \wedge e^c \right). \tag{71}$$

Variation with respect to $e$ and $\omega$ impose respectively constant curvature and vanishing torsion, as an equivalent formulation of Einstein's field equations.

The first order action (71) can be further rewritten as a CS action for the group $SO(2,2)$, by further combining $e^a$ and $\omega^a{}_b$ into a CS gauge field valued in $SO(2,2)$ (and identifying $l/4G$ with the CS level $k$). Its invariance, up to boundary terms, under $SO(2,2)$ gauge transformations reflects the local isometry of the locally $AdS_3$ solutions of pure $AdS_3$ gravity discussed in the last section. The isomorphism $SO(2,2) \simeq (SL(2,\mathbb{R})_L \times SL(2,\mathbb{R})_R)/\mathbb{Z}_2$ finally allows to introduce instead two $\mathfrak{sl}(2,\mathbb{R})$ valued CS gauge fields $A = A^a j_a$ and $\bar{A} = \bar{A}^a \bar{j}_a$, with the coefficients $A^a$ and $\bar{A}^a$ related to the gravitational degrees of freedom by

$$A^a = \frac{1}{2}\epsilon^{abc}\omega_{bc} + \frac{e^a}{l}, \qquad \bar{A}^a = \frac{1}{2}\epsilon^{abc}\omega_{bc} - \frac{e^a}{l}. \tag{72}$$

They are real and independent. The $\mathfrak{sl}(2,\mathbb{R})$ generators $j_a$ and $\bar{j}_a$ satisfy

$$\mathrm{tr}(j_a j_b) = \frac{1}{2}\eta_{ab}, \qquad [j_a, j_b] = \epsilon_{abc}\eta^{cd}j_d \tag{73}$$

with $\epsilon_{012} = 1$, and similarly for the barred sector. In terms of these variables, the action (71) takes the form of the difference of Chern-Simons actions for $A$ and $\bar{A}$:

$$S = S_{CS}[A] - S_{CS}[\bar{A}], \qquad S_{CS}[A] = -\frac{k}{4\pi}\int_{\mathcal{M}} \mathrm{tr}\left(A \wedge dA + \frac{2}{3}A \wedge A \wedge A\right) \tag{74}$$

with the level given by $k = l/(4G)$. The equations of motion now read $F = dA + A \wedge A = 0$ (and similarly for $\bar{F}$), which together impose constant curvature and vanishing torsion. Essential to the gauge theory interpretation of three-dimensional gravity is the identification of the frame indices with CS Lie algebra indices in (72). The CS gauge fields as such are one-forms $A^a = A^a_\mu dx^\mu$ and $\bar{A}^a = \bar{A}^a_\mu dx^\mu$ (with $\mu = t, \phi, r$).

At the quantum level, the considered CS theory is $Z = \int \mathcal{D}(A, \bar{A})e^{i(S_{CS}[A] - S_{CS}[\bar{A}])}$ (where the path integration is over the appropriate quotiented CS gauge group). Of course, the above argument at the level of the action provides us with an equivalence of the classical theories, and not of the full quantum theories. For the purposes in this paper, it will be sufficient that the equivalence holds for the path integral in the neighborhood of classical saddles.

## 3.3 The BTZ black hole saddle of Chern-Simons theory

The reviewed CS description (74) of three-dimensional gravity will allow us to establish the connection to the AS action. Since the boundary theory is developed in a finite temperature framework, we first analytically continue to Euclidean signature. In this section we discuss the analytic continuation, and the Euclidean BTZ solution for the resulting action.

We analytically continue to Euclidean time $\tau$ by setting $t = -i\tau$ as well as $A_t = iA_\tau$ and $\bar{A}_t = i\bar{A}_\tau$. Since

$$A_t dt = A_\tau d\tau, \tag{75}$$

the action $S$ in form notation (74) remains the same under this procedure. That is with the understanding that $A^a = A^a_\mu dx^\mu$ and $\bar{A}^a = \bar{A}^a_\mu dx^\mu$ are now one-forms on a Euclidean manifold $\mathcal{M}$, with the spacetime index taking values $\mu = \tau, \phi, r$.

From the gravity perspective, the fact that $S$ does not pick up a factor of $i$ means that under this continuation we continue to integrate over Lorentzian 'target space' metrics $g_{\mu\nu} = \eta_{ab}e^a_\mu e^b_\nu$ (with an $iS$ in the exponent) rather than over Euclidean 'target space' metrics $g_{\mu\nu} = \delta_{ab}e^a_\mu e^b_\nu$ (with typically $(-S_E)$ in the exponent). As such, the above analytic continuation is different from the one discussed more traditionally in the context of Euclidean quantum gravity [61]. What is crucial here is that our CS gauge field $A^a_\mu$ has two types of indices: a 'worldsheet' spacetime index $\mu$ and a 'target space' index $a$ valued in the Lie algebra. In [61], and as detailed nicely in the appendix of [62], one not only analytically continues $\mu$ from $t$ to $\tau$ in the same way as discussed above, but also continues the index $a$ from being $\mathfrak{sl}(2,\mathbb{R})$ to $\mathfrak{sl}(2,\mathbb{C})$ valued. The latter symmetry reflects the local isometry of locally AdS$_3$ solutions of pure AdS$_3$ gravity in Euclidean signature. However, from a condensed matter theory perspective, it is very unnatural to change the group upon analytically continuing the theory. So we proceed with the continuation as outlined in the previous paragraph, with $j_a$ and $\bar{j}_a$ being still $\mathfrak{sl}(2,\mathbb{R})$ generators. Note that this is the same analytic continuation as the one used in [57] based on similarity to JT/Schwarzian discussions. What we require is that the Euclidean BTZ metric can be obtained as a saddle of our continued CS theory. As we will shortly see this is indeed the case, by allowing a complex $e^a$ and thus complex $(A^a, \bar{A}^a)$ solution.

The Euclidean CS theory we employ is thus (74) with $\mathcal{M}$ a Euclidean manifold spanned by coordinates $(\tau, \phi, r)$. We take the three-manifold to be of the form $\Sigma \times S^1$ with $\Sigma$ a disk,

$$\mathcal{M} = D \times S^1. \tag{76}$$

This is in analogy to the seminal discussion of CS theory on $\Sigma \times \mathbb{R}^1$ (with different choices of $\Sigma$) in [63], where the $\mathbb{R}^1$ direction naturally provides a time for a canonical formulation of the theory. In our case, $S^1$ similarly provides a Euclidean timelike direction for a canonical perspective. We take this direction to be labeled by $\phi$ (not $\tau$, crucially different from [26]). The remaining coordinates $(r, \tau)$ span the disk $D$. That is, the manifold on which the CS gauge field lives is a solid torus with $\tau \sim \tau + \beta$ labeling the contractible cycle with period $\beta$, and $\phi \sim \phi + 2\pi$ the uncontractible cycle with period $2\pi$. This worldsheet torus is then characterized by the modular parameter $\boldsymbol{\tau} = \frac{i\beta}{2\pi l}$ [8].

As in the seminal work [63], the CS action can be brought in a form where the gauge field coordinate $A_\phi$ in the direction away from $\Sigma$ acts as a Lagrange multiplier. To achieve this, we proceed to split the gauge field

$$A = A_\phi\, d\phi + \tilde{A}_i\, dx^i \tag{77}$$

---

[8] Throughout, we use boldface notation $\boldsymbol{\tau}$ for the modular parameter and leave $\tau$ for imaginary time.

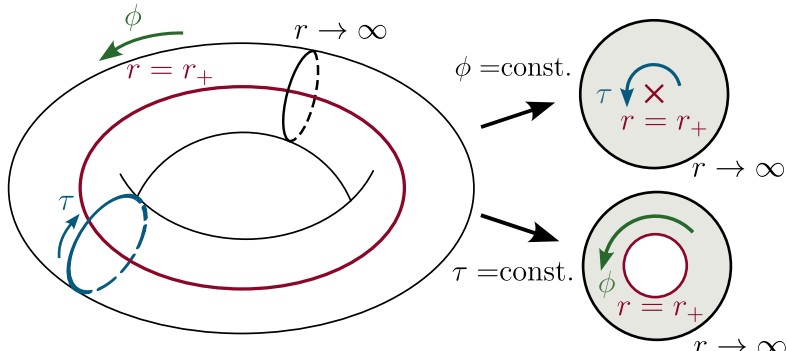

**Figure 4:** The topology of the Euclidean BTZ black hole (84) with horizon at $r = r_+$ is that of a solid torus with contractible $\tau$-cycle and uncontractible $\phi$-cycle.

with $x^i = \{r, \tau\}$ and define $\tilde{F} = F_{r\tau}\, \mathrm{d}r \wedge \mathrm{d}\tau$[9]. The action $S_{CS}[A]$ becomes

$$I_{CS}[A] = -\frac{k}{2\pi} \int_{\mathcal{M}} \mathrm{d}\phi \wedge \mathrm{tr}\left(-\frac{1}{2}\tilde{A} \wedge \partial_\phi \tilde{A} + A_\phi \tilde{F}\right). \tag{78}$$

Because a boundary term has been dropped to obtain this expression, we employ a new letter $I_{CS}$ for this action. It is equal to $S_{CS}$ up to a boundary term of the form $iA_\phi A_\tau$. The new form will allow a rewriting of the theory as effective WZW theory at the boundary.

The action $I_{CS}[A] - I_{CS}[\bar{A}]$ gives rise to a well-defined variational problem subject to chiral boundary conditions when it is supplemented by the boundary action

$$I_{\mathcal{M}_\infty}[A, \bar{A}] = \frac{ik}{4\pi} \int_{\mathcal{M}_\infty} \mathrm{d}^2x \left(\mathrm{tr}(A_\tau^2) + \mathrm{tr}(\bar{A}_\tau^2)\right). \tag{79}$$

Indeed, the variation of the total action

$$I = I_{CS}[A] - I_{CS}[\bar{A}] + I_{\mathcal{M}_\infty}[A, \bar{A}] \tag{80}$$

is given by

$$\delta I = (\text{E.O.M}) + \frac{k}{2\pi} \int_{\mathcal{M}_\infty} \mathrm{d}^2x \left(\delta A_\tau(-A_\phi + iA_\tau) + \delta\bar{A}_\tau(\bar{A}_\phi + i\bar{A}_\tau)\right), \tag{81}$$

the boundary contribution of which vanishes for the boundary conditions $A_\phi = iA_\tau$ and $\bar{A}_\phi = -i\bar{A}_\tau$ at $\mathcal{M}_\infty$. These are the Euclidean versions of the chiral boundary conditions $A_\phi = A_t$ and $\bar{A}_\phi = -\bar{A}_t$ typically used in the CS treatment of gravity [60, 64, 65].

A particular solution of the variational problem with chiral boundary conditions is

$$A_* = \begin{pmatrix} \frac{\mathrm{d}r}{2\xi(r)} & \frac{\mathrm{d}\overline{w}}{2l^2}(r - \xi(r)) \\ \frac{\mathrm{d}\overline{w}}{2l^2}(r + \xi(r)) & -\frac{\mathrm{d}r}{2\xi(r)} \end{pmatrix}, \quad \bar{A}_* = \begin{pmatrix} -\frac{\mathrm{d}r}{2\xi(r)} & -\frac{\mathrm{d}w}{2l^2}(r + \xi(r)) \\ -\frac{\mathrm{d}w}{2l^2}(r - \xi(r)) & \frac{\mathrm{d}r}{2\xi(r)} \end{pmatrix}, \tag{82}$$

where $\xi(r)$ is defined in Eq. (68) and we have defined light-cone coordinates and their Wick rotations

$$x^+ = t + l\phi \mapsto -i\tau + l\phi = \overline{w} \qquad x^- = -t + l\phi \mapsto i\tau + l\phi = w. \tag{83}$$

---

[9]We define $\epsilon^{\tau\phi r} = 1$.

By (72), the solution (82) can be straightforwardly checked to describe the (non-rotating) Euclidean BTZ metric

$$\mathrm{d}s^2 = \frac{\xi(r)^2}{l^2}\,\mathrm{d}\tau^2 + \frac{l^2}{\xi(r)^2}\,\mathrm{d}r^2 + r^2\,\mathrm{d}\phi^2\,, \qquad \xi(r) \equiv \sqrt{r^2 - r_+^2} \tag{84}$$

with $\tau \in (0,\beta)$, $\phi \in (0,2\pi)$ and $r > 0$. It is characterized by a contractible $\tau$-circle, as visualized in Fig. 4. The other way around, to construct the CS gauge field form of this metric, one reads off the dreibeins from the metric, then constructs the spin connections using the zero torsion condition, and finally combines those into chiral and anti-chiral gauge fields through (72).

Solutions to the equations of motion $F = 0$ are pure gauge fields $A = g^{-1}dg$ with $g \in \mathrm{SL}(2,\mathbb{R})$. For the Euclidean BTZ solution, we have

$$g_* = \begin{pmatrix} \rho_* \cosh\frac{\pi\overline{w}}{\beta} & \rho_*^{-1}\sinh\frac{\pi\overline{w}}{\beta} \\ \rho_*\sinh\frac{\pi\overline{w}}{\beta} & \rho_*^{-1}\cosh\frac{\pi\overline{w}}{\beta} \end{pmatrix}, \qquad \overline{g}_* = \begin{pmatrix} \rho_*^{-1}\cosh\frac{\pi w}{\beta} & \rho_*\sinh\frac{\pi w}{\beta} \\ \rho_*^{-1}\sinh\frac{\pi\overline{w}}{\beta} & \rho_*\cosh\frac{\pi w}{\beta} \end{pmatrix}, \tag{85}$$

with $\rho_* = (r/r_+ + \xi(r)/r_+)^{1/2}$. Here we note that for $\tau \to \tau + \beta$, $g_*$ and $\overline{g}_*$ are multi-valued as elements of $\mathrm{SL}(2,\mathbb{R}) \times \mathrm{SL}(2,\mathbb{R})$ with $(g_*,\overline{g}_*) \to (-g_*,-\overline{g}_*)$, but this is cured via the diagonal $\mathbb{Z}_2$ quotient appearing in the gauge group discussed above (72). The single-valuedness is equivalent to a trivial holonomy around the $\tau$-cycle and we will be careful to preserve it in our fluctuation analysis. For $\phi \to \phi + 2\pi$, $g_*$ and $\overline{g}_*$ are multi-valued, indicating a non-trivial $\phi$-holonomy which encodes mass and angular momentum of the black hole [64].

As can be read off from (82), the Euclidean BTZ saddle of our CS theory has complex components, in particular $A_\tau = A_\tau^a j_a$ and $\overline{A}_\tau = \overline{A}_\tau^a \overline{j}_a$ are pure imaginary (with $j_a$ and $\overline{j}_a$ as discussed being real $\mathfrak{sl}(2,\mathbb{R})$ generators). Later we will consider real fluctuations around this complex saddle. As a final consistency check, we calculate the on-shell action of the BTZ solution (82), to which only the boundary term of (80) contributes. Substitution of Eq. (82) leads to

$$-iI_{\mathcal{M}_\infty}[A,\overline{A}] = \frac{k}{4\pi}\int_{\mathcal{M}_\infty} \mathrm{d}^2x\big(\mathrm{tr}(A_\tau^2) + \mathrm{tr}(\overline{A}_\tau^2)\big) = \beta M - S_{BH}, \tag{86}$$

reproducing, as it should, the result of the Euclidean, on-shell Einstein-Hilbert action for the BTZ black hole. This evaluates to the free energy with the familiar Bekenstein-Hawking entropy $S_{BH} = 2\pi r_+/(4G)$.

Interpreting the BTZ solution (84) as being asymptotically indistinguishable from the global AdS solution suggests a set of fall-off conditions on the general metric that gives a precise meaning to the notion of an "asymptotically AdS spacetime". These are the Brown-Henneaux boundary conditions [56]. In gauge field language they are given by the condition that a solution displays the following behavior at $r \to \infty$:

$$A = \begin{pmatrix} \frac{\mathrm{d}r}{2r} + O(r^{-2}) & O(r^{-1}) \\ \frac{r\mathrm{d}\overline{w}}{l^2} + O(r^{-1}) & -\frac{\mathrm{d}r}{2r} + O(r^{-2}) \end{pmatrix}, \qquad \overline{A} = \begin{pmatrix} -\frac{\mathrm{d}r}{2r} + O(r^{-2}) & -\frac{r\mathrm{d}w}{l^2} + O(r^{-1}) \\ O(r^{-1}) & \frac{\mathrm{d}r}{2r} + O(r^{-2}) \end{pmatrix}. \tag{87}$$

With these boundary conditions, the asymptotic symmetry group (of residual diffeomorphisms in metric formulation, or gauge transformations in the CS formulation) famously is that of the Virasoro symmetry of 2D CFT, with the combination of gravitational parameters $3l/(2G)$ taking the role of a central charge $C$ [56]. It is these Brown-Henneaux conditions that we will impose on fields describing real fluctuations around the BTZ saddle.

## 3.4 AS action for fluctuations around the BTZ black hole

With the Chern-Simons action (80) at hand, we now proceed to the AS representation of the theory. We employ the same strategy as that of [26], but applied to the Euclidean BTZ rather than global AdS$_3$. Compared to the AS action for fluctuations around global AdS$_3$ derived in [26], ours will have interchanged roles of $\tau$ and $\phi$. This is important for the match to the AS action (47) we obtain from the boundary perspective. The AS action (47) or (2) appeared before in gravitational discussions, e.g. in [27, 39, 66], without an explicit derivation employing the techniques of [26]. It is this derivation we provide now.

The gauge field we discuss lives on a manifold with the topology of a solid torus $D \times S^1$ with $D$ the disk ($\{r, \tau\} \times \{\phi\}$). The AS construction amounts to the application of a standard protocol [63] reducing the theory to one defined on the torus boundary with coordinates $(\phi, \tau)$. In a first step, the integration over $A_\phi$ imposes the constraint $\tilde{F} = 0$, i.e. $\tilde{A} \equiv g^{-1}\tilde{d}g$ where $g \in \mathrm{SL}(2, \mathbb{R})$ is again single-valued in $\tau$, and $\tilde{d}$ is the exterior derivative with respect to $\{r, \tau\}$. We note that this representation implies a local redundancy $g \to v(\phi)g$ with $v \in \mathrm{PSL}(2, \mathbb{R})$, which we will comment on at the end of the section.

The substitution of this representation into the action (80) leads to the sum of two chiral Wess-Zumino-Witten models [67] (see, e.g., Ref. [60] for details)

$$I = I_+[g] + I_-[\overline{g}]. \tag{88}$$

Here,

$$I_\pm[g] = \frac{k}{2\pi}\left(-i\int_{\mathcal{M}_\infty} d^2x\, \mathrm{tr}\left((g^{-1})'\partial_\pm g\right) \pm \frac{1}{6}\int_{\mathcal{M}} \mathrm{tr}\left(g^{-1}dg \wedge g^{-1}dg \wedge g^{-1}dg\right)\right), \tag{89}$$

where $\partial_\pm = \frac{1}{2}(\partial_\tau \mp i\partial_x)$ with $x = l\phi$, we use the shorthand notation $f' = \partial_\tau f$ and the path integral measure is now $\mathcal{D}g$, the Haar-measure on $\mathrm{PSL}(2, \mathbb{R})$. The first term $I_\pm[g]$ is defined on the boundary, and the second is locally exact, meaning that it, too, affords a boundary representation. In this way the theory becomes one of gauge field fluctuations supported by the system boundary.

To make this reduction concrete, we employ the Gauss decomposition

$$g = \begin{pmatrix} 1 & 0 \\ h & 1 \end{pmatrix}\begin{pmatrix} \lambda & 0 \\ 0 & \lambda^{-1} \end{pmatrix}\begin{pmatrix} 1 & \Psi \\ 0 & 1 \end{pmatrix}, \qquad \overline{g} = \begin{pmatrix} 1 & \overline{h} \\ 0 & 1 \end{pmatrix}\begin{pmatrix} 1/\overline{\lambda} & 0 \\ 0 & \overline{\lambda} \end{pmatrix}\begin{pmatrix} 1 & 0 \\ \overline{\Psi} & 1 \end{pmatrix}, \tag{90}$$

with $h, \Psi \in \mathbb{R}$ and $\lambda > 0$, and likewise for the barred sector. Compatibility with the Brown-Henneaux boundary conditions (87) is established via the relations

$$\lambda = \sqrt{\frac{r}{l^2 h'}}, \quad \Psi = -\frac{l^2 h''}{2rh'}, \qquad \overline{\lambda} = \sqrt{\frac{r}{l^2 \overline{h}'}}, \quad \overline{\Psi} = -\frac{l^2 \overline{h}''}{2r\overline{h}'} \tag{91}$$

where $h \equiv h\big|_{\mathcal{M}_\infty}$ is independent of $r$, and $h' > 0$ for $g$ to be real. In this way, the theory has collapsed to a single real degree of freedom, $h = h(\tau, x)$. In this representation, the remaining bulk term in (89) becomes a boundary term

$$\frac{1}{3}\int_{\mathcal{M}} \mathrm{tr}\left((g^{-1}dg)^3\right) = \int_{\mathcal{M}} d\lambda^2 \wedge d\Psi \wedge dh = \int_{\mathcal{M}_\infty} \lambda^2 d\Psi \wedge dh$$

and with the constraints (91) in place the WZW-action assumes the form

$$I_\pm[h] = \frac{iC}{12\pi}\int_{\mathcal{M}_\infty} d^2x\left(\frac{3}{2}\frac{h''\partial_\pm h'}{h'^2} - \frac{\partial_\pm h''}{h'}\right) = \frac{iC}{12\pi}\int_{\mathcal{M}_\infty} d^2x\left(\frac{1}{2}\frac{h''\partial_\pm h'}{h'^2} - \left(\frac{\partial_\pm h'}{h'}\right)'\right) \tag{92}$$

Specifically, in the Gauss-parametrization of the BTZ stationary solution, up to a factor of $i$, we have $h_* = \tan(\pi(\tau + ix)/\beta)$. Generalizing to functions which are $i)$ continuously connected to the saddle and $ii)$ keep $g$ single-valued as a function of $\tau$, fluctuations are described by

$$h(\tau, x) = \tan\left(\frac{\pi f(\tau, x)}{\beta}\right) \tag{93}$$

in terms of a reparametrization of the imaginary time cycle, $f(\tau, x) \sim f(\tau, x) + \beta$ and $f'(\tau, x) > 0$. The fluctuation measure is obtained by evaluating the Gauss measure (which originally reads as $\mathcal{D}g = \mathcal{D}H\mathcal{D}\lambda\mathcal{D}\Psi$) at the boundary subject to the constraints (91) leading to $\mathcal{D}f\left(\prod_{\tau,x} f'\right)^{-1}$.

Substituting the trajectory (93) into the boundary WZW action (92), one recovers the AS action $S_\pm^\beta[f]$ by virtue of relation (47). We have thus reached our goal, a representation of the partition sum as

$$Z(\boldsymbol{\tau}) = \int_{\substack{f(\tau,x=0)=f(\tau,x=L) \\ \bar{f}(\tau,x=0)=\bar{f}(\tau,x=L)}} \frac{\mathcal{D}f(\tau,x)\mathcal{D}\bar{f}(\tau,x)}{\prod_{x,\tau} f'(\tau,x)\bar{f}'(\tau,x)} e^{-S_+^\beta[f]-S_-^\beta[\bar{f}]}, \tag{94}$$

with

$$S_\pm^\beta[f] = \frac{C}{24\pi} \int_0^\beta d\tau \int_0^{2\pi l} dx \left(\frac{f''\partial_\pm f'}{f'^2} - \frac{4\pi^2}{\beta^2} f'\partial_\pm f\right), \qquad f' \equiv \partial_\tau f, \tag{95}$$

and the coupling constant $C = \frac{3l}{2G} \gg 1$ given by the Brown-Henneaux central charge. This path integral is identical to that discussed in [27], but our interpretation and derivation is different: In our discussion, it is the fluctuation theory of the BTZ saddle-point. We note again that it describes real fluctuations around a complex saddle.

Before concluding this section, a few comments are in place. First, in the representation Eq. (94), the $\mathrm{PSL}(2,\mathbb{R})$ redundancy $g(x,\tau) \to \upsilon(x)g(x,\tau)$ that is hardwired into the theory acts as a fractional linear transformation

$$\tan\left(\frac{\pi}{\beta}f\right) \to \frac{a(x)\tan\left(\frac{\pi}{\beta}f\right) + b(x)}{c(x)\tan\left(\frac{\pi}{\beta}f\right) + d(x)}, \qquad ab - cd = 1. \tag{96}$$

From here on, we will exclusively work with $\mathrm{PSL}(2,\mathbb{R})$ quotients and ignore this transformation throughout. The action Eq. (95) is identical to the effective action (2) for the chiral SYK model, except that presently we are working with a velocity scale $u$ set to unity and with the identification $L \equiv 2\pi l$. Secondly, in Appendix E.1 we show that Gaussian integration around the stationary solutions leads to the partition sum

$$Z(\boldsymbol{\tau}) = |\chi_0(-1/\boldsymbol{\tau})|^2, \qquad \chi_0(\boldsymbol{\tau}) = q^{-\frac{c}{24}} \prod_{n=2}^{\infty} \frac{1}{1-q^n}, \qquad q = e^{2\pi i\boldsymbol{\tau}}, \tag{97}$$

which is the vacuum character of a CFT with central charge $c = C + 13$ in the dual channel. Finally, a Laplace transform of the partition function (97) to an energy representation recovers the expected Cardy growth of the density of states $\rho(E) \sim e^{2\pi\sqrt{\frac{c}{6}E}}$ at high energies $E \gg 1$, reproducing the microcanonical Bekenstein-Hawking entropy.

Let us finally comment on the matching of coupling constants between the AS actions derived from the bulk and the boundary. The bulk theory contains three different length

scales: $G, l$ and $\beta$, where time is measured in units of length. Consequently, the partition function depends only on two dimensionless ratios: $C = 3l/2G$ and $\tau = i\beta/2\pi l$, which must be equivalently identified in the boundary theory. There, the central charge $C$ is given by Eqs. (43), while the modular parameter can be extracted from the underlying geometry as $\tau = i\beta u/L$. Here, the system size is related to the $\mathrm{AdS}_3$ radius $l$ in the holographic setup by the simple relation $L = 2\pi l$, while the velocity $u$ can be determined from the same Eqs. (43) and (44). Also, given $C$, $L$ and $\beta u$ as boundary data, the effective Newton's constant $G$ in the bulk can then be reconstructed as $G = 3l/2C$.

## 4  Correlation functions from Liouville theory

The goal of this section is to evaluate the effect of the reparametrization averaged with the action (2) on various correlation functions of the model. In Subsection 4.1, we begin by introducing our method of choice for calculating observables, namely the Liouville map. This approach is then applied in Subsection 4.2 to the two-point function of Majorana operators at coinciding spatial points, and in Subsection 4.3 to 'clusters' of Majorana operators, which allow for more interesting effects in the correlation function as compared to single operators. Finally, in Subsection 4.5, we calculate the OTOC between a cluster of Majorana operators with a single operator, as a means to diagnose chaos.

The building blocks for all subsequent calculations are generalizations of the reparameterized two-point function (31), which we define as

$$\mathcal{O}^{2l}_{\tau_1\tau_2}(x) = \left( \frac{\partial_{\tau_1} f_1 \partial_{\tau_2} f_2}{\sin^2\left(\pi(f_1 - f_2)/\beta\right)} \right)^l, \qquad f_i = f(\tau_i, x) \tag{98}$$

and from which we recover $\widetilde{G}_{\tau_1\tau_2}$ for $l = 1/4$. These bilocal operators are interesting from the holographic point of view, since conformal correlation functions as e.g. (17) are known to describe particle propagation between two points anchored on the boundary (at equal spatial position) along a bulk geodesic. Dressing these objects with reparametrizations governed by the AS action is equivalent in the bulk to dressing the particles with gravitons [40, 68–70].

The calculations of observables simplify in the limit of large $N$, leading to a large coupling constant $C$ in front of the AS action via Eq. (43). The largeness of this coupling enables a classical approach, wherein correlation functions are computed in terms of on-shell solutions $f = f_0$. We also compare this approach to results obtained by more elaborate methods of conformal field theory, differing from ours only in anomalous corrections to the scaling dimensions of the operators appearing in the correlation functions. In executing this program, we will need to distinguish between two types of correlations functions: those containing operator insertions $\mathcal{O}$ which are 'light' in the sense that they do not affect the stationary phase solutions $f_0$ (up to corrections vanishing in the $C \to \infty$ limit), and others with 'heavy' operators which do modify the solutions. (In the holographic context, the difference between heavy and light particles is whether they back-react on the geometry or not [71].) We will show that the classical treatment of heavy operator insertions is a highly economical approach, alternative to previous strategies based on perturbation theory in the reparametrization mode formalism [26, 72] or the Liouville bootstrap [66, 73].

Within our approach, the complexity of the problem is shifted to the computation of equations of motion, in the presence of reparameterized two-point functions $\mathcal{O}^{2l}_{\tau_1\tau_2}(x)$. To simplify this step, we take preparative measures: We first map the AS theory to Liouville theory with boundaries, for which the equations of motion simplify considerably [71, 74]. Having found the solutions on the Liouville side, we map them back to obtain the saddle-point reparametrization $f_0(\tau, x)$. In the following, we explain this strategy in detail.

## 4.1 The Liouville map

The equivalence between AS theory and Liouville theory with boundaries we use builds on the work of [66, 73]. It posits the identity of path integrals [10]

$$\int \frac{\mathcal{D}f}{\prod_{x,\tau} f'(\tau,x)} (\cdots) e^{-S^\beta_\pm[f]} = \int \mathcal{D}\phi (\cdots) e^{-S_L[\phi]}, \tag{99}$$

where the ellipses symbolically represent matching operator insertions. This connection is established via a mapping of field variables, $\phi(\tau,x) = \phi(f(\tau,x))$, defined for positive $\tau \in [0, \beta/2]$ as

$$\phi(\tau,x) = \ln\left( \frac{-\partial_\tau f_+ \partial_\tau f_-}{\sin^2(\pi(f_+ - f_-)/\beta)} \right), \qquad f_\pm = f(\pm\tau, x). \tag{100}$$

The $\phi$-representation of the fluctuation action is given by [11]

$$S[\phi] = \frac{C}{24\pi u} \int_0^L \int_0^{\beta/2} dx\, d\tau \left( \frac{(\partial_\tau \phi)^2}{4} + \frac{(u\partial_x \phi)^2}{4} + \left(\frac{2\pi}{\beta}\right)^2 e^\phi - \partial_\tau^2 \phi \right), \tag{101}$$

which is a variant of the action of Liouville field theory in the semiclassical regime $C \gg 1$ [75], and is therefore denoted $S_L[\phi]$ in (99). Importantly, the constants appearing in Eq. (101) are direct imports from the microscopic model discussed in section 2, where we have reintroduced the velocity scale $u(1/\beta)$ given by Eq. (44). In Appendix E.1 we corroborate the equivalence to the $f$-representation by demonstrating that without operator insertions, integration over $\phi$ recovers the partition function (97).

Before proceeding, we need to take a closer look at the space-time interval underlying the $\phi$-theory, $[0, \beta/2] \times [0, L]$, see Fig. 5. Eq. (100) implies a divergence $\phi(\tau,x) \to \infty$ at $\tau = 0, \beta/2$ corresponding to Dirichlet, or ZZ-boundary conditions $e^{-\phi(0,x)} = e^{-\phi(\beta/2,x)} \to 0$ in the parlance of Liouville field theory. The rationale behind restricting $\phi$ to the positive imaginary time domain is likewise evident from Eq. (100): the $\phi$-field at $\tau > 0$ encodes information on reparametrization configurations both at $\tau$ and the mirror symmetric time $-\tau$. The same symmetry implies a constraint on temporally bilocal operator insertions $\mathcal{O}^{2l}_{\tau_1 \tau_2}(x)$ as in (98) in the theory: only configurations symmetric around the origin $\mathcal{O}^{2l}_{\tau_1, -\tau_1}(x)$ afford a consistent $\phi$-representation. Finally, the $f$-representation of our operators diverges at coinciding temporal arguments due to the presence of denominators $\sim 1/\sin(\pi(f_\tau - f_{-\tau})/\beta)^\alpha$. In the $\phi$-language, these divergences are reflected in the above-mentioned boundary conditions.

Within the set of temporally symmetric observables, it is convenient to switch to a holomorphic representation (83), i.e.

$$\mathcal{O}^{2l}_{\tau_1 \tau_2}(x) = e^{l\phi(\tau,x)} \equiv \mathcal{V}_l(w, \overline{w}) \qquad w = i\tau + x/u, \qquad \overline{w} = -i\tau + x/u. \tag{102}$$

This transformation conveniently eliminates the $SL(2,\mathbb{R})$ 'target space' redundancy (96) of the reparametrization mode [66], i.e. the Liouville representation describes the $f$-theory with $SL(2,\mathbb{R})$ appropriately modded out.

---

[10]The relation holds for $S_\pm$ independently. When following [66] and expressing $f(\tau,x)$ in terms of two functions $A(\tau,x)$ and $B(\tau,x)$ defined for $0 < \tau < \beta/2$ and $-\beta/2 < \tau < 0$ respectively, the roles of $A$ and $B$ are exchanged in the actions $S_+$ and $S_-$ and one hence has to also exchange their roles in the field transformation.

[11]To avoid confusion with the quantities on the disk which are defined with a prime, we denote temporal derivatives as $\partial_\tau$ in this section.

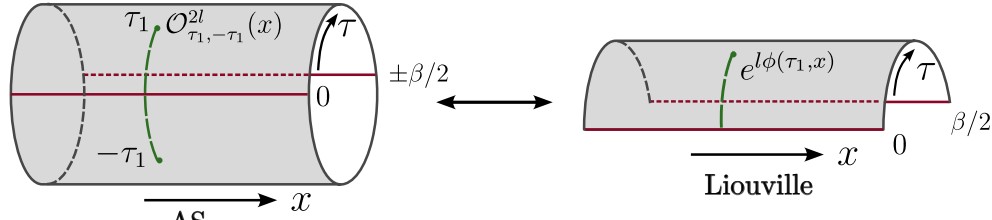

**Figure 5:** Left: Imaginary time $\tau$ runs in the interval $\tau \in (-\beta/2, \beta/2)$ along the cylinder. Bilocal operators are placed symmetrical around $\tau = 0$ with legs at the two points $(\tau_1, -\tau_1)$ with $\tau_1 > 0$. Right: After mapping the system to Liouville theory, the range of the imaginary time cycle is halved to $\tau \in (0, \beta/2)$. The bilocal operator becomes a local vertex operator with behavior for $\tau_1 \to 0, \beta/2$ prescribed by the ZZ-boundary conditions.

On any domain related to the strip by a holomorphic coordinate transformation defined by $(w, \overline{w}) \mapsto (\xi(w), \overline{\xi(w)}) \equiv (\xi, \overline{\xi})$ with $\xi, \overline{\xi}$ *dimensionless*, the field $\phi$ transforms as [76, 77]

$$\left(\frac{2\pi}{\beta}\right)^2 e^{\phi(w, \overline{w})} = e^{\phi'(\xi, \overline{\xi})} \left|\frac{\partial \xi(w)}{\partial w}\right|^2. \tag{103}$$

Here the bar denotes complex conjugation, and we multiplied by a factor of $(2\pi/\beta)^2$ to keep the field $\phi'$ dimensionless. Eq. (103) reveals the interpretation of $\phi$ as a Weyl factor of a metric with $e^{\phi} \, dw \, d\overline{w}$ invariant under conformal transformations. It also implies $\Delta_l = 2l$ as the engineering dimension of the operators Eq. (102).

Turning to operator insertions, depending on the value of $l$, one distinguishes between 'light' operators with $l = \mathcal{O}(1)$ and 'heavy' ones, $l = \mathcal{O}(C)$ [76]. To understand their difference, note that the insertion of $n$ vertex operators at coordinates $w_i$ with degree $l_i$ is described by a modified action

$$S[\phi] \mapsto S[\phi] - \sum_{i}^{n} l_i \phi(w_i). \tag{104}$$

Accordingly, the equations of motion $\delta_\phi S = 0$ generalize to

$$\frac{1}{2} \partial_a \partial^a \phi = \left(\frac{2\pi}{\beta}\right)^2 e^{\phi} - \frac{24\pi}{C} \sum_{i}^{n} l_i \delta^{(2)}(w - w_i). \tag{105}$$

Light operators affect the equations of motion only negligibly in terms of a $1/C$ correction, while the heavy ones need to be accounted for in the solution. As a consistency check, we note that the action in the absence of sources, $S_0 = -(\pi C/12u)(L/\beta)$, obtained by evaluating the AS action (95) on the trivial solution $f_0(\tau, x) = \tau$, follows equivalently by substituting the solution (114) of the sourceless Eq. (105) into (101).

The most general correlation function we will work with is between a light operator of conformal weight $l_1$ and a heavy one of conformal weight $l_2$. Classically, it reads

$$\left\langle \mathcal{V}_{l_1}(w_1) \mathcal{V}_{l_2}(w_2) \right\rangle = e^{l_1 \phi(w_1; l_2, w_2) - S(l_2, w_2) + S_0}. \tag{106}$$

Here, $\phi(w; l_2, w_2)$ is the solution to the Liouville equation (105) in the presence of a single heavy source of weight $l_2$ at $w_2$, $\exp(S_0)$ normalizes the correlation function, and we suppressed the $\overline{w}$-dependence for readability. The evaluation of its action,

$$S(l_2, w_2) \equiv S[\phi(\circ; l_2, w_2)] - l_2 \lim_{w \to w_2} \phi(w; l_2, w_2) \tag{107}$$

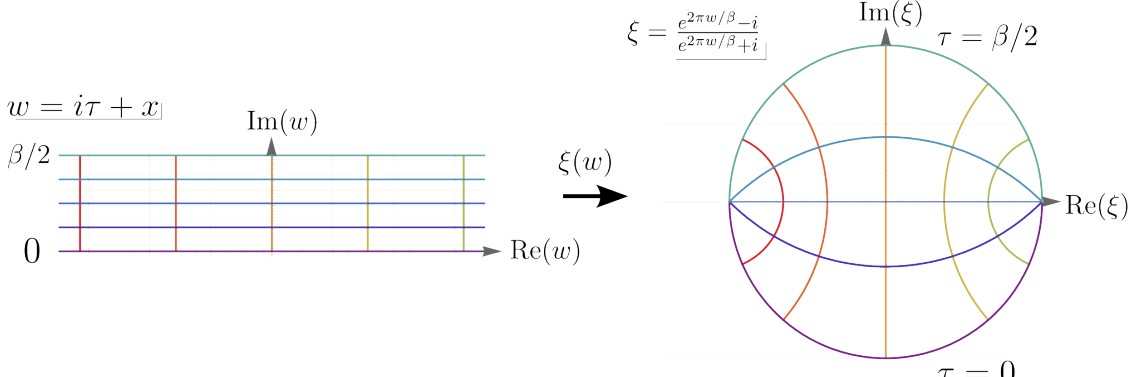

**Figure 6:** Behavior of constant $\tau$- and $x$-slices on the half-strip under the mapping (110).

requires a regularization procedure because of singularities in the limit $w \to w_2$[12]. Finally, we will describe the high-temperature regime $(\beta u)/L \ll 1$ of interest to us by taking the limit of a spatially infinite system, $L \to \infty$.

## 4.2 Single Majorana operator

We begin with an analysis of the correlation function (9) between two ('light') single Majorana operator insertions at the same point in space, $x = x'$, and times $\tau_1$ and $\tau_2$. In this case, the need to modify the action does not present itself, and the computation of the correlation function reduces to taking the average of (31) over the reparametrization $f$. Focusing on the leading $(0+1)d$ SYK contribution $G^0$, we have

$$G^{xx}_{\tau_1\tau_2} \equiv G(\tau_{12}) = \left\langle \widetilde{G}_{\tau_1\tau_2} \right\rangle_{\text{AS}}, \tag{108}$$

where we neglected the spatial argument due to translational invariance in $x$, and $\tau_{1,2} = \pm\tau_0/2$. Turning to the Liouville representation, we include the factor $(\pi/\beta)^{1/2}$ of the finite temperature theory and obtain

$$G(\tau_0) = n_0 \left(\frac{2\pi}{\beta J}\right)^{1/2} \left\langle \mathcal{V}_{1/4}(w_0) \right\rangle, \qquad n_0 = -\frac{\text{sgn}(\tau_{12})k_0}{\sqrt{4\pi}}, \tag{109}$$

where $w_0 = i\tau_0/2 + x/u$ and the exponent $1/4$ determines the engineering scaling dimension of the Green's function as $\Delta_G = 2 \times 1/4 = 1/2$.

We next analyze this functional average within the stationary phase approach. Inspection of Eq. (106) shows that in the absence of a heavy source, $\mathcal{V}_{l_2}$, the action contributions $-S(0,0) + S_0 = 0$ cancel, and we are left with the exponential $e^{\phi_0(w)/4}$, where $\phi_0$ is the solution to the Liouville equations of motion (105) without source insertion. The concrete computation of such solutions is best done in a disk-geometry, related to the strip by a conformal transformation [71, 78]. In the following, we review this map, and discuss how it enables the concrete evaluation of the mean field theory.

Our map between strip and disk is given by (cf. Figure 6)

$$\xi(w) = \frac{e^{\frac{2\pi}{\beta}w} - i}{e^{\frac{2\pi}{\beta}w} + i}. \tag{110}$$

---

[12]In order for $S(l_2, w_2)$ not to diverge close to the source, one needs to supplement the action with a counter term [74, 75, 78], which also modifies the behavior of $S(l_2, w_2)$ under conformal transformations. Details on the regularization are given in Appendix E.3.

Then the mapping $(w, \overline{w}) \mapsto (\xi(w), \overline{\xi(w)}) = (\xi, \overline{\xi})$ maps the full strip to the interior of the unit disk. Labeling functions $\psi'(\xi)$ defined on the disk by a prime, the translation back to the strip is defined as $\psi(w) = (\xi^{-1} \circ \psi' \circ \xi)(w)$, or implicitly through $\xi(\psi(w)) = \psi'(\xi(w))$.

One can show [71] that in disk coordinates the general solution $\phi'(\xi)$ of the Liouville equation coupled to sources, Eq. (105), is given by

$$e^{\phi'(\xi, \overline{\xi})} = \frac{4|\partial_\xi f'(\xi)|^2}{(1 - |f'(\xi)|^2)^2}, \tag{111}$$

where $f'(\xi)$ is a meromorphic function, whose specific profile is determined by the inhomogeneity. In addition, the asymptotic condition $|f'(\xi)|^2 \to 1$ for $|\xi| \to 1$ implements the divergence $e^{\phi'(\xi)} \to \infty$ required by the ZZ boundary conditions.

From this result, the solution on the strip is found by inverse transform under the $\xi$-map, via Eq. (103). Defining a function $f(w)$ through $\xi(f(w)) \equiv f'(\xi(w))$, the condition $\overline{f(w)} = f(\overline{w})$ holds. Using this feature, a straightforward calculation summarized in Appendix E.2 leads to

$$\phi(w, \overline{w}) = \ln\left(\frac{|\partial f(w)|^2}{\sin^2(i\pi(f(\overline{w}) - f(w))/\beta)}\right). \tag{112}$$

In this representation, $f(w)$ has the status of a conformal transformation of the strip. Comparing it with the definition of the Liouville field Eq. (100), we are led to the identification

$$f(\tau, x) = -if(w), \tag{113}$$

between the solutions of the Liouville equation and reparametrization transformations in the AS theory.

We next apply these structures to the computation of the light operator correlation function Eqs. (108) and (109). In this case, we just need the stationary solution in the absence of sources. On the disk, this is the identity map, $f'(\xi) = \xi$, implying that $f'(w) = w$ is the identity, too. (With Eq. (113) we find that, up to irrelevant $\mathrm{SL}(2, \mathbb{R})$ transformations, this result translates to the trivial reparametrization, $f(\tau, x) = \tau$.). Finally, Eq. (112) implies

$$\phi(w, \overline{w}) = -\ln\left(\sin^2(i\pi(\overline{w} - w)/\beta)\right), \tag{114}$$

leading to

$$G(\tau_0) = n_0 \left(\frac{2\pi}{\beta J}\right)^{1/2} \left|\sin\frac{\pi\tau_0}{\beta}\right|^{-1/2}. \tag{115}$$

We observe the absence of corrections to the scaling exponent within the variational framework. This finding, expected to become exact in the $C \to \infty$ limit, is consistent with boundary CFT results [79, 80]

$$\Delta_{l_1} = 2l_1 + \frac{6l_1}{C}(1 - 2l_1), \tag{116}$$

i.e. the mean field value modified by a correction vanishing in the semiclassical limit.

## 4.3 Majorana clusters

We next generalize to the insertion of a large number, $p$, of Majorana operators. Following Ref. [14], we consider the $p$ fields grouped in time into clusters of almost coinciding times

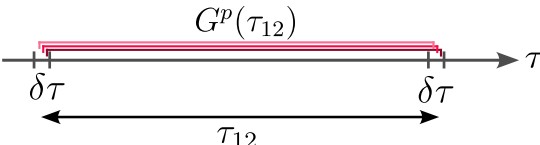

**Figure 7:** Construction of a cluster of Majorana fermions with $\delta\tau/\tau_{12} \ll 1$.

$-\tau_0/2 + \eta_j$ and $+\tau_0/2 + \eta_j'$, $j = 1, \ldots, p$, where $|\eta_j|, |\eta_p'| < \delta\tau$ are small, see Fig. 7. Taking the limit of vanishingly small separations, we obtain

$$G^{[p]}(\tau_0) \overset{\text{def.}}{=} N^{-p} \sum_{i_1 \cdots i_p} \langle \gamma^{i_1}(\tau_0/2 + \eta_1)\gamma^{i_1}(-\tau_0/2 + \eta_1', x) \cdots \gamma^{i_p}(\tau_0/2 + \eta_p)\gamma^{i_p}(-\tau_0/2 + \eta_p', x) \rangle$$

$$= \left\langle G^p_{\tau_0/2, -\tau_0/2}[f] \right\rangle_{\text{AS}} = n_0^p \left( \frac{2\pi}{\beta J} \right)^{p/2} \mathcal{V}_{p/4}(w_0, \overline{w}_0), \tag{117}$$

In this way, we define a cluster operator of engineering dimension $l_2 = p/4$, which in the case $p \sim C$ becomes heavy. Using again (106), but now at $l_1 = 0$, the correlation function becomes

$$G^{[p]}(\tau_0) = n_0^p \left( \frac{2\pi}{\beta J} \right)^{p/2} e^{-S(l_2, w_0) + S_0}. \tag{118}$$

In order to calculate this object, we need to evaluate the Liouville action in presence of a source at an arbitrary position.

To start, consider an operator insertion of weight $l_2/C = \text{const} \lesssim 1$[13] at strip coordinate $w_0 = i\beta/4$, or $\xi = 0$ in the disk picture. As shown in Appendix F, the solution is given by

$$f'(\xi) = \xi^{1-2\delta}, \qquad e^{\phi'(\xi; 0, l_2)} \overset{\text{Eq. (111)}}{=} \frac{4(1-2\delta)^2}{\left( |\xi|^{2\delta} - |\xi|^{2-2\delta} \right)^2}, \tag{119}$$

where $\delta = 6l_2/C$. We next shift the source from the origin to the disk coordinate $\xi_0 = \xi(w_0)$ corresponding to our strip coordinate $w_0$. This is achieved by application of the $SU(1,1)$-isometry on the Poincaré disk

$$h(\xi; \xi_0) = \frac{\xi_0 - \xi}{1 - \overline{\xi}_0 \xi}, \tag{120}$$

leading to [74, 78]

$$\phi'(\xi; \xi_0, l_2) = \phi'(h(\xi; \xi_0); 0, l_2) + \ln \left| \frac{dh(\xi; \xi_0)}{d\xi} \right|^2. \tag{121}$$

With the general disk-solution at hand and using the conformal mapping (110), one can relate the disk action $S'(l_2, \xi_0)$, evaluated on the configuration (121), to the desired action $S(l_2, w_0)$ on the strip. Referring to Appendix E.3 for somewhat intricate details of this procedure, we simply state the final result:

$$S(l_2, w_0) = S_0 - \ln U(l_2) - \frac{C\delta(1-\delta)}{6} \ln \left( \frac{\beta^2}{4\pi^2} \left| h_w'(\xi(w), \xi_0) \right|^2_{w=w_0} \right) + \frac{C\delta^2}{6} \ln \left| \frac{2\pi}{\beta J} \right|^2, \tag{122}$$

---

[13]Operator insertions with $pl > C/12$ create holes in the underlying manifold (in our case the unit disk) [71, 76]. Such system deformations are not included microscopic model, and we excluded it by imposing the bound.

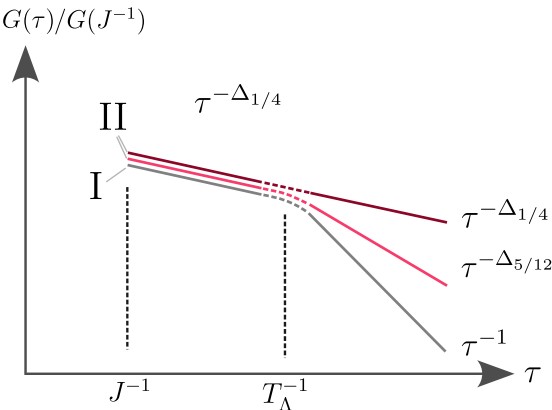

**Figure 8:** Log-log plot of the Majorana two-point functions for Model I (grey) and Model II (dark and light red) with $\gamma = 3/2$. While both models show the same behavior for times shorter than $T_\Lambda^{-1}$, the relevance of the kinetic term in Model I leads to a crossover to the Fermi-liquid regime at larger time scales. For Model II, beyond this time scale the Green's functions of localized and mobile Majoranas show algebraic decays with exponents $\Delta_{1/4} = \frac{1}{2}(1 + 3/2C)$ and $\Delta_{5/12} = \frac{5}{6}(1 + 2/C)$, respectively.

Here the prefactor $U(l_2)$ with $U(0) = 1$ stems from the $\xi$-independent part of the on-shell action and is given by Eq. (E.30). With the help of definitions for mappings $\xi(w)$ and $h(\xi)$, see Eqs. (110) and (120), a straightforward evaluation using a chain rules for derivatives yields

$$\frac{\beta^2}{4\pi^2}\left|h'_w(\xi(w), \xi_0)\right|^2_{w=w_0} = \frac{1}{4} \times \frac{1}{\sin^2(\pi\tau_0/\beta)}, \tag{123}$$

which is independent of the spatial position $x$ as expected. Plugging the result for the action $S(l_2, w_0)$ into Eq. (118), we obtain for the correlator

$$G^{[p]}(\tau_0) = n_0^p U(l_2)\left(\frac{\pi}{\beta J}\right)^{2l_2\left(1-\frac{6l_2}{C}\right)}\left|\sin\frac{\pi\tau_0}{\beta}\right|^{-2l_2\left(1-\frac{6l_2}{C}\right)}, \tag{124}$$

predicting a significant change of the scaling dimension $\Delta_{l_2} \to \Delta_{\text{s.cl.}} \equiv 2l_2(1 - 6l_2/C)$, of order $\mathcal{O}(C)$ in case of heavy operators. Comparing to the CFT framework, there the prefactor $U(l_2)$ is called the ZZ one-point coefficient [81], and the scaling dimension is given by [79,80]

$$\Delta_{l_2}^Q = \Delta_{\text{s.cl.}} + \frac{6l_2}{C} \equiv 2l_2 + \frac{6l_2}{C}(1 - 2l_2). \tag{125}$$

Again, we observe consistency with the semiclassical result up to subleading terms. Within the path integral approach employed here, one can also demonstrate [78] that a quantum correction $6l_2/C$ to the semiclassical scaling dimension $\Delta_{\text{s.cl.}}$ arises solely from the Gaussian fluctuations around the saddle-point trajectory (121). In other words, the result (125) is one-loop exact.

## 4.4 Majorana two-point functions

Having identified the relevant scaling dimensions, we are now in a position to discuss the two-point correlation function (9) of Majoranas operators. We begin with the linearly dispersive Model I whose correlation function at short times $J^{-1} \ll \tau \ll T_\Lambda^{-1}$ decays with

the mean-field SYK exponent $\Delta_G = 1/2$ (cf. section 2.1). Reparametrization fluctuations change this exponent to $\Delta^Q_{1/4}$, Eq. (116), leading to $\mathcal{O}(C^{-1})$ corrections relative to the mean-field result. For longer times, $\tau \gtrsim T_\Lambda^{-1}$, reparameterization fluctuations are gapped out, and Model I exhibits Fermi liquid behavior, with an exponent $\Delta_{\text{FL}} = 1$ (cf. the gray line in Fig. 8).

For Model II, we can analyze two correlators: the full two-point function, defined by Eq. (9), and the one associated only with mobile Majoranas, given by Eq. (22). The full correlator is again described by the mean-field SYK exponent $\Delta_G = 1/2$ for the entire time range, which is shifted to $\Delta^Q_{1/4}$ when quantum fluctuations are included (see the dark red line in Fig. 8, plotted on a log-log scale). By contrast, the second correlator, $g^{xx}_\tau$, which refers to delocalized Majoranas, decays with a mean-field exponent $\Delta = 1/2 + 1/2\gamma$ at long times $\tau \gtrsim T_\Lambda^{-1}$. For our case of interest, where $\gamma = 3/2$, reparametrization fluctuations change this value to $\Delta^Q_{5/12}$. As a result, the qualitative behavior of $g^{xx}_\tau$ (see the light red line in Fig. 8) closely resembles that of the Majorana two-point function of Model I.

It is now worth discussing the relation between the above results for the Majorana correlation function and the well-studied problem of electron tunneling in mesoscopic physics. As we have seen in sections 4.2 and 4.3, the net effect of reparametrization fluctuations is to change the mean-field scaling exponent of the two-point Majorana function, $\Delta_l = 2l$, to its quantum counterpart, as given in Eq. (125), where the induced correction scales as $1/C$ (and thus vanishes in the limit $N \to \infty$). The analogous phenomenon in condensed matter physics is broadly known as electron tunneling in the presence of an 'electromagnetic environment' [82]. This encompasses a large class of problems studying how electron-electron interactions suppress the single particle Green's function (or tunneling rate) in low-dimensional correlated electron systems. Examples include two-dimensional disordered films [83, 84], quasi-one-dimensional disordered wires [85], one-dimensional Luttinger liquids out of equilibrium [86], and compressible quantum Hall (QH) edge states [87]. In all these cases, the solution strategy closely follows the approach discussed above: one determines the optimal fluctuation of the electron density induced by a delta-source representing the injection of an extra electron charge into the system, and then evaluates the action cost of such an optimal fluctuation. This action typically scales as $S(\tau_0) \propto \ln^d(\tau_0)$, where $d = 1, 2$ is the system dimension and $\tau_0$ is the tunneling time. A similar scaling behavior appears in Eq. (122) in the limit $\tau_0 \ll \beta$. The precise nature of the 'electromagnetic environment' depends on the specific problem and may, for instance, include statistical Chern-Simons gauge fields when describing composite fermions [87]. In this context, reparametrization fluctuations in the chiral SYK model, which correspond to boundary gravitons in the dual holographic setting, can be viewed as a direct analog of such 'electromagnetic environment' in the Majorana system, with its strength controlled by the inverse central charge $1/C$ or, in other words, by the gravitational Newton's constant $G$ in the bulk.

## 4.5 OTOC in the Heavy-Heavy-Light-Light limit

We finally probe early-time chaos of our model by calculating the contribution of reparameterization fluctuations to the OTOC. This observable has been extensively studied in SYK chains [33, 88–91]. In particular, systems exhibiting an emergent reparameterization symmetry generically show growth of OTOCs with the maximal Lyapunov exponent $\lambda = \frac{2\pi}{\beta}$ [88–91]. While in the referenced works, the propagator of the soft modes in the Gaussian approximation has a diffusive pole, the one corresponding to the AS action is of ballistic nature [26, 72]. In both cases, one expects the same maximally chaotic behavior [92, 93], as we explicitly confirm now. To this end, we consider the OTOC between a cluster of Majorana operators and a single Majorana operator. In the low-energy theory, this quantity assumes the form of

a two-point function of a heavy and a light vertex operator, cf. Eq. (106), referred to as a Heavy-Heavy-Light-Light (HHLL) correlation function in the following. While this quantity can be computed within the reparametrization mode formalism [26], we use the Liouville approach for our analysis. Though these calculations operate within the general framework of Liouville theory as developed before, their detailed execution requires a sequence of multiple analytic continuations. Casual readers are invited to skip this construction and directly turn to our final result Eq. (135).

For a general field theory, the OTOC is defined as [94]

$$\tilde{F}(t,x) = \mathrm{Tr}[yVyW(t,x)yVyW(t,x)], \quad y^4 = \frac{1}{Z}e^{-\beta H}, \tag{126}$$

where $t = -i\tau$ is real time and $V \equiv V(0,0)$ and $W(\tau,x)$ are local operators. This function can be rewritten as [15]

$$\tilde{F}(t,x) = \mathrm{Tr}\big[y^4 T_\tau V(\tau_1,0)V(\tau_2,0)W(\tau_3,x)W(\tau_4,x)\big]\Big|_{\tau \to it}, \tag{127}$$

where we analytically continued to imaginary times

$$\tau_1 = \frac{3\beta}{4}, \quad \tau_3 = \frac{\beta}{2}+\tau, \quad \tau_2 = \frac{\beta}{4}, \quad \tau_4 = \tau, \quad \tau_1 > \tau_3 > \tau_2 > \tau_4, \tag{128}$$

and exchanged the order of the second and the third operator in the previous line. For our calculation, we take

$$V(\tau,0) = \gamma_{\mathbf{i}}^p(\tau,0) \equiv \prod_{k=1}^p \gamma_{i_k}(\tau,0), \qquad \mathbf{i} = \{i_1,..,i_p\}, \qquad W(\tau,x) = \gamma_j(\tau,x), \tag{129}$$

with $p = \mathcal{O}(C)$, where we sum over indices appearing twice in the correlation function and neglected the time increments in the definition of $V$ as compared to (117). Further, to simplify the calculation, one regularizes this object as

$$F(t,x) = \frac{\sum_{\mathbf{i},j} \mathrm{Tr}\big[y^4 T_\tau \gamma_{\mathbf{i}}^p(\tau_1,0)\gamma_{\mathbf{i}}^p(\tau_2,0)\gamma_j(\tau_3,x)\gamma_j(\tau_4,x)\big]}{\sum_{\mathbf{i}} \mathrm{Tr}\big[y^4 T_\tau \gamma_{\mathbf{i}}^p(\tau_1,0)\gamma_{\mathbf{i}}^p(\tau_2,0)\big] \sum_j \mathrm{Tr}\big[y^4 T_\tau \gamma_j(\tau_3,x)\gamma_j(\tau_4,x)\big]}\Bigg|_{\tau \to it}. \tag{130}$$

The question now is how to transfer this object to the Liouville framework. While the times $\tau_1$ and $\tau_2$ in the configuration (128) are mirrored on the upper half-plane and $V$ hence realizes a heavy operator as defined in Eq. (117), this is not true for the times $\tau_3$ and $\tau_4$ corresponding to $W$. Fortunately, as described in [95], this does not pose a problem since the non-mirrored configuration can always be obtained from the mirrored one by an analytical continuation. As illustrated in Figure 9 $a)$, on the plane with coordinates $z = e^{2\pi w \beta^{-1}}$ with $w = i\tau + x$, one starts from $w_3' = \overline{w}_4 = -i\tau + x$ and then continues $w_3' \mapsto w_3 = i(\tau + \beta/2) + x$. In a next step, one further continues to the mixed time OTOC configuration $\tau \mapsto it$ via the move $ii)$ shown in Figure 9 $b)$. Within the saddle-point approximation employed here, the possibility of such an analytical continuation is ensured by the analytic dependence of the Liouville field (112) on the complex coordinates $(w, \overline{w})$, as discussed in more details in Appendix E.2.

To implement these two steps in our context, we start from the two-point function on the strip

$$g_4(w_2, \overline{w}_2, w_4, \overline{w}_4) \equiv \frac{\big\langle \mathcal{V}_{l_1}(w_4, \overline{w}_4)\mathcal{V}_{l_2}(w_2, \overline{w}_2)\big\rangle}{\big\langle \mathcal{V}_{l_1}(w_4, \overline{w}_4)\big\rangle\big\langle \mathcal{V}_{l_2}(w_2, \overline{w}_2)\big\rangle}, \tag{131}$$

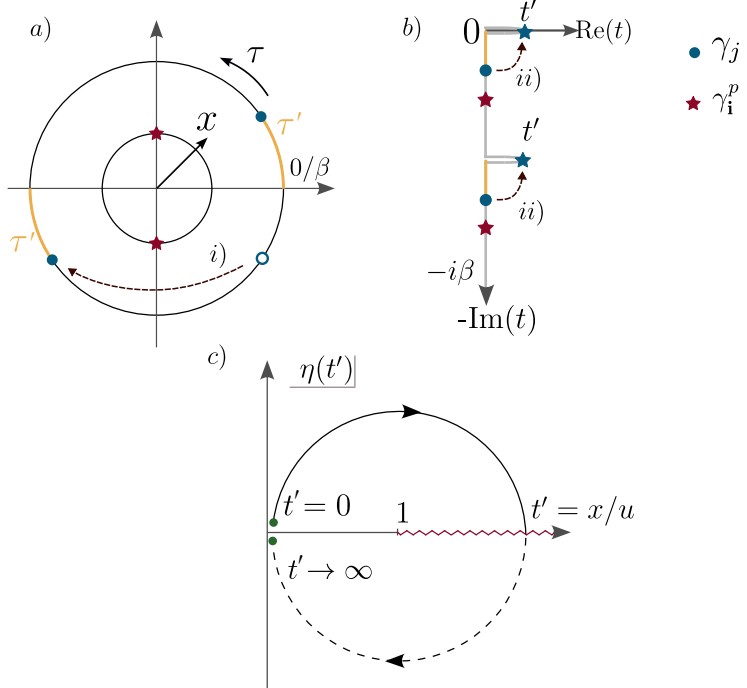

**Figure 9:** Configuration of operators in the OTOC calculation (time argument of the OTOC is denoted by $\tau'/t'$): In $a$) the original mirrored configuration is broken via the move $i$) to bring the operators into the configuration (128). In $b$) the complex time-contour of the correlation function after the analytic continuation $\tau' \to it'$ by the move $ii$) is shown. In $c$), the path of the cross-ratio $\eta(t')$ is shown, where the segments on the first and on the second Riemann sheet are indicated by full and dashed lines respectively.

where the conformal weights now scale like $l_1 = \mathcal{O}(1)$ and $l_2 = \mathcal{O}(C)$ as in (106) and we made the $\overline{w}$-dependence explicit. Also note that the normalization of the four-point function cancels against the conformal transformation factors (103) when moving between coordinate systems, i.e. we can switch freely between $\xi$- and $w$-coordinates. By (106), the logarithm of the full correlation function is then given by

$$\ln g_4 = l_1(\phi_1'(\xi_4; l_2, \xi_2) - \phi_0'(\xi_4)), \tag{132}$$

where we understand all disk-coordinates appearing as functions of the strip-coordinates $\xi = \xi(w)$. Using Eqs. (119) and (121) for the one-source solution and also taking the limit $\delta \to 0$ for $\phi_0'$, we obtain

$$l_1(\phi_1'(\xi_4; l_2, \xi_2) - \phi_0'(\xi_4)) = l_1 \ln\left(\frac{(1-2\delta)^2(1-|h|^2)^2|h|^{(-4\delta)}}{\left(1-|h|^{2-4\delta}\right)^2}\right)$$

$$= 2l_1 \ln\left(\frac{(1-2\delta)\eta(1-\eta)^{-\delta}}{1-(1-\eta)^{1-2\delta}}\right), \tag{133}$$

where we used again the definition $\delta = 6l_2/C$, $h = h(\xi_4; \xi_2)$ and introduced the conformal cross-ratio $\eta = 1 - |h(\xi_4; \xi_2)|^2$. This object is now exactly the logarithm of the HHLL vacuum conformal block of a CFT, encoding interactions due to pure gravity [69, 70]. We now perform the move $i$) $\overline{\xi}_4 \mapsto \xi_3$, which amounts to inserting the times (128) and the

corresponding positions into the conformal cross-ratio. From here, we perform move $ii$) $\tau \mapsto it$ and pick $x > 0$ to observe $\langle \mathcal{V}_{l_1}(w_4, \overline{w}_4) \rangle$ crossing the light-cone of $\langle \mathcal{V}_{l_2}(w_2, \overline{w}_2) \rangle$ at $t = x/u$. The cross-ratio can then be simplified to

$$\eta(t) = \frac{2i}{i - \sinh \frac{2\pi}{\beta}(t - x/u)} \, . \tag{134}$$

As a function of $t$, it traces out a circle in the complex plane centered around $\eta = 1$, as illustrated in Fig. 9 $c$): It starts at $\eta \approx 0$, moves in the first quadrant to $\eta = 2$ at $t = x/u$ and then traces out the lower half in the fourth quadrant on the way back to $\eta \approx 0$ for $t \gg x/u$. When hitting the light-cone at $t = x/u$, the cross-ratio crosses the branch cut of the root in the denominator of (133) and hence is defined on the second Riemann sheet of the function. There, the behavior of the correlation function changes qualitatively compared to that of a time-ordered correlation function. The OTOC is for all times well-approximated by the function [70, 96]

$$F(t, x) = g_4(w_2, \overline{w}_2, w_4, \overline{w}_4) \Big|_{\text{OTOC}} \simeq \frac{1}{\left(1 + \frac{6\pi l_2}{C} e^{\frac{2\pi}{\beta}(t - x/u)}\right)} \, . \tag{135}$$

From here we can read off the maximal Lyapunov exponent $\lambda = 2\pi/\beta$ and the butterfly velocity $v_B = u$. While the first result agrees with previous calculations of the OTOC in holographic CFTs in the same limit of parameters [19, 70] and reflects the universality mentioned before [92, 93], the second one is more non-trivial. The identification $v_B = u$, see Eqs. (43) and (44), implies that $v_B$ is approximately independent of $x$ and $t$, which needs to be contrasted to the more complex form of $v_B(x, t)$ found in the chiral variant of the SYK model proposed in Ref. [33]. Also, in our model $v_B$ shows a weak $J$-dependence, as compared, e.g., to the random hopping models of [89] and [91], where $\lim_{J/V \to \infty} v_B(J/V) = 0$ holds with $V$ being the average hopping, i.e. strong interactions suppress information scrambling[14]. In essence, the functional form of the OTOC says that in a strongly chaotic setting with large $\lambda \propto T$ operators separated by a distance $x > 0$ remain uncorrelated, i.e. $C(x, t) \equiv 1 - F(x, t) \approx 0$, for times $t < x/u$. At larger times, the OTOC quickly reaches its saturation value $C(t, x) \approx 1$, which in our normalization means that the two operators in question become correlated. A new insight gained from the present analysis is that we explicitly constructed operators of the microscopic theory, namely Majorana clusters, which map to vertex operators in the universal Liouville variables. These operators serve as probes into the chaotic behavior of the underlying chiral SYK model and can also be used as building blocks for more general correlation functions.

## 5  Summary and discussion

The quest for a boundary theory of pure $AdS_3$ gravity is a longstanding problem. A crucial step towards a better understanding was the identification of the AS action as a proposed theory of fluctuations around gravitational saddle points [26]. The structural similarity between the AS action and the Schwarzian action in one dimension lower provided strong evidence for the existence of a JT/SYK-like duality between gravity and a dimensionally extended variant of a model in the SYK universality class. In this work, we have constructed this theory and its holographic correspondence to the bulk at a level of explicitness previously

---

[14]Both models posses two independent hopping constants $V$ and $V'$, which we assume to be equal in this discussion.

realized in one dimension lower. Adapting the rationale of the two-dimensional holographic correspondence, our construction identified the AS action as the common low energy theory of a chain of SYK islands coupled to realize a particular type of chiral dispersion, and the BTZ black hole saddle of AdS$_3$ gravity. The construction of this theory from 'microscopically defined' parent theories allowed us to match coupling constants and to compute the concrete form of correlation functions, notably out-of-time-order correlation functions as witnesses of early time chaotic dynamics.

On the boundary, our starting point was a chain of SYK dots coupled in such a way that their Hamiltonian included a one body term with uniformly 'chiral' dispersion $\partial_k \epsilon(k) \geq 0$. Microscopically, such conditions cannot be realized in stand-alone quasi one-dimensional lattices, discrete symmetries excluding a uniformly increasing dispersion. However, systems exhibiting all signatures required by our construction — strong randomness, interactions, and chirality — are realizable at topological insulator boundaries, e.g., the edges of quantum Hall droplets. While our discussion has not been focusing on aspects of concrete experimental feasibility, a point to emphasize is that the microscopic one-dimensional boundary theory underlying our construction is realizable in principle within the framework of condensed matter physics. Similarly to the situation in one dimension lower, it exhibits an infinite dimensional symmetry under local reparameterizations of time, spontaneously broken at the mean level to a residual SL(2, $\mathbb{R}$), and explicitly broken by time derivatives and inter-site coupling. A lowest order gradient expansion in these symmetry breaking operators then produced the AS action, where the condition of exactly retained local SL(2, $\mathbb{R}$) symmetry implied stringent conditions on the allowed contents of that action.

Turning to the gravity side, our goal was to derive a matching AS action, now as the boundary fluctuation theory of BTZ black hole solutions. Indeed, earlier constructions in the literature [26, 27, 39] had identified the AS action as a fluctuation theory common to a variety of gravitational settings, using different methods. For our set-up, we opted to follow the well-established principal strategy of [26], but applied beyond the global AdS$_3$ case. We started from the Chern-Simons representation of three-dimensional gravity, considered it for field configurations representing fluctuations around a BTZ black hole, and from there projected to the boundary in Euclidean signature. The resulting AS action could then be compared to that following from the SYK construction, to obtain an identification matching microscopic system parameters (basically, $C = 3l/2G$). The net result of this construction is a bulk-boundary correspondence equaling that in one dimension lower in its level of explicitness (In fact, a straightforward dimensional reduction reduces to the familiar [12, 17] SYK/JT duality in terms of the Schwarzian action [66].)

As in the two-dimensional case, the present theory describes a holographic duality at time scales comparable to $N$, the number of local constituents of the SYK dots, or 'singly perturbative' time scales in the parlance of gravity. The two-dimensional holographic correspondence is also established via a second bridge, building on chaotic fluctuations on much later time scales, $\sim e^N$, comparable to the Hilbert space dimension of the SYK dots. This link has been established in terms of matrix theory series representations ('singly non-perturbative'), and elements of topological string theory ('doubly non-perturbative'). While elements of random matrix theory have been identified for three-dimensional gravity in Ref. [44], there remains the challenge to establish a corresponding link to a microscopic boundary theory.

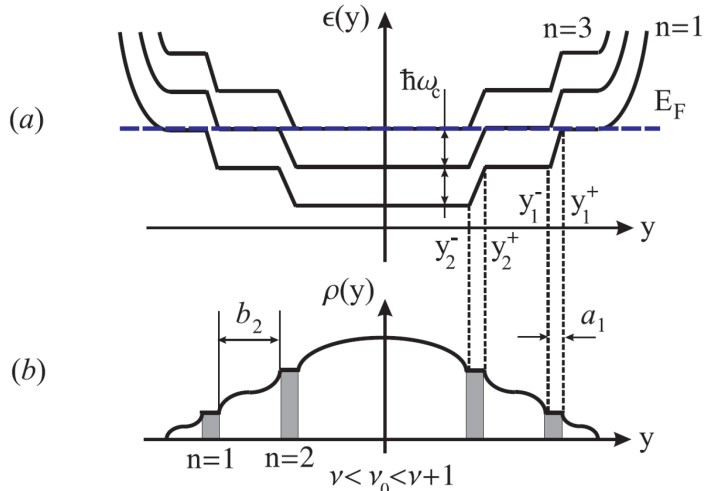

**Figure 10:** The structure of edge states in IQHE (taken from Ref. [86]). A cross-section through the center of a narrow 2DEG strip with a smooth gate confined electron density: (a) Landau level energies as the function of a transverse coordinate $y$ and (b) electron density $\rho(y)$. In addition to $\nu$ completely filled LLs, the central region can sustain a partially filled Landau level. We define $a_k = y_k^+ - y_k^-$ as the width of the $k$-th incompressible strip (shown in grey), which has the integer filling factor $k$. The compressible regions (shown in white) correspond to areas with non-integer filling fraction and have the width $b_k = y_{k-1}^- - y_k^+$. The illustration above corresponds to the case $\nu = 2$.

## Acknowledgments

We thank Thomas Mertens for valuable discussions. KW also thanks Suting Zhao for useful conversations. This work has been supported by the Deutsche Forschungsgemeinschaft (DFG, German Research Foundation) within CRC network TR 183 (Project Grant No. 277101999) as part of subproject B04.

## A  Shklovskii's model of IQHE egde modes

We summarize below the main results of Shklovskii's approach [49, 50] to the formation of edge states at the boundary of 2DEG in the IQHE regime in the situation of the so-called 'smooth' edge, i.e. when an electron density at the boundary gradually approaches to zero on a spatial scale which is much larger than the magnetic length $l_B$. Under such condition and when effects of Coulomb interaction are included on a level of Hartree-Fock approximation the spatial profile of Landau levels (LL) close the edge happens to be of the form shown in Fig. 2, which is characterized by a wide (almost) flat region and the power-law exponent $\gamma = 3/2$. In more concrete terms, the 2DEG in the QHE regime is divided into compressible and incompressible strips. The filling factor in the $n$-th incompressible strip is integer. These strips are separated by much wider regions of compressible Hall liquid with a non-integer filling factor (compressible strips). In Fig. 10 we show a corresponding sketch of a typical electron density profile $\rho(y)$ corresponding to the narrow in $y$-direction and infinite along $x$-direction strip of a gate confined 2DEG so that both left and right propagating edge modes emerge on respective boundaries. Let us denote by $y_k^\pm$ the boundaries between compressible

and incompressible regions. Then $a_k = y_k^+ - y_k^-$ is the width of the $k$-th incompressible strip, while $b_k = y_{k-1}^- - y_k^+$ is the width of the $k$-th compressible one. In the situation of gate-induced confinement of 2DEG in the QHE regime the widths $b_k \gg a_B$, with $a_B = \epsilon/(m_{\text{eff}} e^2)$ being the effective Bohr radius. At the same time $a_k$ scales as $a_k \sim (b_k a_B)^{1/2}$, so that in general the condition $b_k \gg a_k \gg a_B$ is satisfied. In this picture compressible regions play the role of edge channels — the self-consistent electrostatic potential is constant through the compressible strips and can be controlled by connecting them to external leads. Exact positions $y_k^\pm$ and widths of those strips were found in Refs. [49, 50].

It is also worth mentioning here another important spatial scale in QHE — the magnetic length $l_B = (c/eB)^{1/2}$. In the IQHE regime we have $l_B \gg a_B$. This condition can be also written as

$$\hbar\omega_c = \frac{eB}{mc} \ll \frac{e^2}{\epsilon l_B}, \tag{A.1}$$

i.e. the distance between the Landau levels is smaller that a typical Coulomb energy on the magnetic length scale. Thus the Bohr radius $a_B$ plays a role of the short distance cut-off.

One of the main results of the Shklovskii's theory are the following singularities in the spatial profiles of electron density $\rho(y)$ and electron energies close to the strip boundaries $y_k^\pm$. Namely, a density profile shows square-root-like behavior,

$$\rho(y) \sim \mp|y - y_k^\pm|^{1/2} + kn_L, \qquad y \to y_k^\pm \pm 0, \qquad n_L = 1/(2\pi l_B^2), \tag{A.2}$$

(here $n_L$ is an electron density on the 1st LL), while the corresponding energy profile of the $k$-th and $(k+1)$-th Landau levels close the Fermi energy reads as

$$\epsilon(y) \sim E_F \pm |y - y_k^\pm|^{3/2}, \qquad y \to y_k^\pm \mp 0, \tag{A.3}$$

and we have omitted dimensionful prefactors in front of singular terms for brevity (also, the sketch of LLs in Fig. 10 doesn't reflect the exponent 3/2 properly).

Now, it is well-known that for two-dimensional Landau problem one can relate the momentum and position dependencies of eigenstates. Namely, if we fix the gauge to be $A_x = -B_z y$ and $A_y = A_z = 0$, then in the absence of disorder a momentum $k_x$ along $x$-axis is a good quantum number, thus the wave function $\psi_{nk_x}(x, y)$ is characterized by the continuous momentum $k_x$ and a discrete Landau level index $n$. In this gauge and for large $k_x \gg 1/l_B$ the wave function forms a narrow thread extended along $x$-axis but localized on a scale $l_B$ in $y$-direction and centered around

$$y_{k_x} = k_x l_B^2. \tag{A.4}$$

When the self-consistent electrostatic potential changes adiabatically on a scale $l_B$, such that even an incompressible strip is sufficiently wide, $a_k \gg l_B$, the spectrum of Landau problem reads

$$\epsilon_{nk_x} = \epsilon_n(y_{k_x}) \equiv \epsilon_n(k_x l_B^2), \tag{A.5}$$

where energies $\epsilon_n(y)$ are those shown in Fig. 10. On taking into account the relation (A.3) one reproduces the model of the flat band in Fig. 2 with the exponent $\gamma = 3/2$. In particular, for a given compressible strip of width $b_k$ the corresponding momentum $k_0$ becomes $k_0 = b_k/(2l_B^2)$.

# B   Solution of mean-field equations in the IR limit

This Appendix provides technical details for the derivation of equations (20) and (21). They follow from the solution of the mean-field equation for the 'flat-band' model in the strong dispersive with the use of a perturbation theory w.r.t. a small parameter $k_1/k_0 \ll 1$.

## B.1 Perturbative scheme

We solve the system of mean-field equations (11) by neglecting the $i\epsilon$- term in the 1st equation. This approximation will be justified a posteriori. A simple evaluation of the integral (12) yields

$$
\begin{aligned}
G_\epsilon &\simeq -\frac{1}{2\pi} \int\limits_{-\pi/a}^{\pi/a} \frac{dk}{\epsilon_k + \Sigma_\epsilon} \simeq -\frac{k_0}{\pi\Sigma_\epsilon} + \frac{1}{\pi} \int_0^{+\infty} \frac{\Sigma_\epsilon\, dk}{\Lambda^2(k/k_1)^{2\gamma} + [i\Sigma_\epsilon]^2} \\
&= -\frac{k_0}{\pi\Sigma_\epsilon}\left(1 + b_1(\gamma)\frac{k_1}{k_0}\left(\frac{i\pi\Sigma_\epsilon}{\Lambda}\right)^{1/\gamma}\right),
\end{aligned}
\tag{B.1}
$$

where we've introduced a constant $b_1(\gamma)$ defined by an integral

$$
b_1(\gamma) = \frac{1}{\pi^\gamma} \int_0^{+\infty} \frac{dx}{x^{2\gamma} + 1} = \frac{\pi^{1-1/\gamma}}{2\gamma \sin\frac{\pi}{2\gamma}}.
\tag{B.2}
$$

It was used above that the integral in (B.1) is dominated by momenta satisfying $k \ll k_1$ as long as $|\epsilon| \ll T_\Lambda$, thus the upper limit of integration can be extended to infinity. Because of $\Sigma_\epsilon$ is yet unknown function of $k_1/k_0$, the result (B.1) needs to be understood as the starting point for the self-consistent calculation of both $G_\epsilon$ and $\Sigma_\epsilon$.

Taking the leading term in (B.1) we get $G_\epsilon \Sigma_\epsilon = -k_0/\pi$. Together with the relation $\Sigma_\tau = (J^2/k_0^3) G_\tau^3$ it yields the zeroth order SYK-type solution (16). To find the 1st order correction to the SYK solution one may use the ansatz (20) for the energy dependence of $\Sigma_\epsilon$ with yet undefined coefficient $\sigma_1(\gamma)$. Its self-consistence will be verified later. Expanding (B.1) up to linear order in $k_1/k_0$, the Green's function becomes

$$
G_\epsilon = G_\epsilon^0\left[1 + \left(b_1(\gamma) - \sigma_1(\gamma)\right)\left(\frac{k_1}{k_0}\right)\left|\frac{\epsilon J}{\Lambda^2}\right|^{1/2\gamma} + \ldots\right]
\tag{B.3}
$$

On defining the function $C(\Delta)$ as in (B.9), one can further translate this expression into the time domain using the Fourier transform (B.10),

$$
G_\tau = G_\tau^0\left[1 + \left(b_1(\gamma) - \sigma_1(\gamma)\right)b_2(\gamma)\left(\frac{k_1}{k_0}\right)\left|\frac{J}{\Lambda^2\tau}\right|^{1/2\gamma} + \ldots\right],
\tag{B.4}
$$

where $b_2(\gamma) = C(1 - \Delta_1)/\sqrt{2\pi}$ and $\Delta_1 = 1/2 + 1/(2\gamma)$. At this stage we can use the self-consistent equation, $\Sigma_\tau = (J^2/k_0^3) G_\tau^3$, which yields the self-energy

$$
\Sigma_\tau = \Sigma_\tau^0\left[1 + 3\left(b_1(\gamma) - \sigma_1(\gamma)\right)b_2(\gamma)\left(\frac{k_1}{k_0}\right)\left|\frac{J}{\Lambda^2\tau}\right|^{1/2\gamma} + \ldots\right].
\tag{B.5}
$$

The latter can be converted back to the energy domain using once again the relation (B.10). The final result reads

$$
\Sigma_\epsilon = \Sigma_\epsilon^0\left[1 + \left(b_1(\gamma) - \sigma_1(\gamma)\right)b_2(\gamma)b_3(\gamma)\left(\frac{k_1}{k_0}\right)\left|\frac{\epsilon J}{\Lambda^2}\right|^{1/2\gamma} + \ldots\right],
\tag{B.6}
$$

where $b_3(\gamma) = 3C(1 + \Delta_1)/(2\sqrt{2\pi})$. The latter expression indeed matches the initial ansatz for $\Sigma_\epsilon$ in the form of equation (20) and yields the linear equation for the unknown $\sigma_1(\gamma)$:

$$
\sigma_1(\gamma) = \left(b_1(\gamma) - \sigma_1(\gamma)\right)b_2(\gamma)b_3(\gamma),
\tag{B.7}
$$

and its solution is given by (B.14). After that the final expressions for coefficients $g_1(\gamma)$, $\tilde{g}_1(\gamma)$ and $\tilde{\sigma}_1(\gamma)$ follow from equations (B.3), (B.4) and (B.5), respectively. For example,

$$g_1(\gamma) = b_1(\gamma) - \sigma_1(\gamma) = \frac{b_1(\gamma)}{1 + b_2(\gamma)b_3(\gamma)}, \tag{B.8}$$

in agreement with Eq. (B.13). One also observes that dropping $i\epsilon$–term from the Dyson equation was a legitimate assumption, since the found correction to the self-energy scales as $\sim |\epsilon|^{\Delta_1}$ with $\Delta_1 < 1$.

## B.2 Table of integrals and coefficients

The coefficients in a perturbative solution of the Dyson's equation can be expressed via the function

$$C(\Delta) = 2\cos\left(\frac{\pi\Delta}{2}\right)\Gamma(1-\Delta), \qquad C(\Delta)C(1-\Delta) = 2\pi, \tag{B.9}$$

which specify the following direct Fourier transform,

$$\int_{-\infty}^{+\infty} \frac{\mathrm{sgn}(\tau)}{|\tau|^\Delta} e^{i\epsilon\tau}\, d\tau = iC(\Delta)\mathrm{sgn}(\epsilon)|\epsilon|^{\Delta-1}, \tag{B.10}$$

and its inverse,

$$\int_{-\infty}^{+\infty} i\mathrm{sgn}(\epsilon)|\epsilon|^{\Delta-1} e^{-i\epsilon\tau} \frac{d\epsilon}{2\pi} = \frac{C(1-\Delta)}{2\pi} \frac{\mathrm{sgn}(\tau)}{|\tau|^\Delta}. \tag{B.11}$$

Let us further define $\Delta_1 = 1/2 + 1/(2\gamma)$ to be the (twice) scaling dimension of the mobile Majoranas and introduce auxiliary functions

$$b_1(\gamma) = \frac{\pi^{1-1/\gamma}}{2\gamma \sin\frac{\pi}{2\gamma}}, \qquad\qquad b_2(\gamma) = \frac{C(1-\Delta_1)}{\sqrt{2\pi}},$$

$$b_3(\gamma) = \frac{3C(1+\Delta_1)}{2\sqrt{2\pi}}, \qquad b_{23}(\gamma) = b_2(\gamma)b_3(\gamma) \equiv \frac{3}{2\Delta_1} \tan\frac{\pi\Delta_1}{2}. \tag{B.12}$$

Then $g$-functions, defining the asymptotic behavior of correlators $G_\epsilon$ and $G_\tau$, read

$$g_1(\gamma) = \frac{b_1(\gamma)}{1 + b_{23}(\gamma)}, \qquad \tilde{g}_1(\gamma) = \frac{b_1(\gamma)b_2(\gamma)}{1 + b_{23}(\gamma)}, \tag{B.13}$$

while $\sigma$-functions, quantifying subleading corrections to $\Sigma_\epsilon$ and $\Sigma_\tau$, evaluate to

$$\sigma_1(\gamma) = \frac{b_1(\gamma)b_{23}(\gamma)}{1 + b_{23}(\gamma)}, \qquad \tilde{\sigma}_1(\gamma) = 3\tilde{g}_1(\gamma). \tag{B.14}$$

Here the monotonous functions $g_1(\gamma)$ and $\tilde{g}_1(\gamma)$ are equal to zero at $\gamma = 1$ and saturate to $1/4$ at $\gamma \to \infty$. The function $\sigma_1(\gamma)$ has limiting values $\sigma_1(1) = 1/2$ and $\sigma_1(+\infty) = 3/4$ and it displays a weak minimum around $\gamma = 5/4$.

## C Reparametrizations as similarity transformations

It is instructive to formulate the reparametrization of bilocal operators and, in particular, the transformation (33) in a language of matrix algebra. Consider the set of time dependent functions $g : S^1 \to S^1, \tau \to g(\tau) \equiv g_\tau$ as a linear space with scalar product

$\langle g|h \rangle = \int d\tau\, g_\tau h_{\tau'}$, and orthonormal basis $\{|t\rangle\}$, $\langle \tau|\tau \rangle = \delta_{\tau-\tau'}$. We may interpret the reparametrization, $\tau \mapsto t = f(\tau)$, and its inverse, $t \mapsto \tau = F(t)$, as the introduction of a new basis $\{|t\rangle\}$, which is obtained from the old one through a transformation matrix

$$\langle t|\tau \rangle \equiv S_{t,\tau} = \delta_{t-f(\tau)} f_\tau'^{1/2},$$
$$\langle \tau|t \rangle \equiv S_{\tau,t}^{-1} = \delta_{\tau-F(t)} F_t'^{1/2}, \tag{C.1}$$

where the scaling factors multiplying the $\delta$-function make the transformation unitary. Indeed, one can check it by evaluating

$$[S^\dagger]_{\tau,t} = S_{t,\tau}^* = \delta_{t-f(\tau)} f_\tau'^{1/2} = \delta_{\tau-F(t)} F_t'^{1/2} = S_{\tau,t}^{-1}, \tag{C.2}$$

as required, where it was used that $f_\tau' F_{f(\tau)}' = 1$. With this definition, the orthonormality of the new basis is established as

$$\langle t|t' \rangle = \int d\tau\, \langle t|\tau \rangle \langle \tau|t' \rangle = \int d\tau S_{t,\tau} S_{\tau,t'}^{-1} = \int d\tau \delta_{t-f(\tau)} f_\tau'^{1/2} \delta_{\tau-F(t')} F_{t'}'^{1/2} = \delta_{t-t'}.$$

Also note that the $f$-scaling factors commute with the transformation in the sense that $F_t' S_{t,\tau} = S_{t,\tau} f_\tau'$. With these structures in place, the time reparametrization can be understood as a change of basis. Specifically, one can represent the mapping $\mathcal{M}_\Delta$ as a superposition of the unitary $S$ and a diagonal (in time) congruent transformations. To see it explicitly, one may evaluate

$$\langle \tau_1|\overline{O}|\tau_2 \rangle = \left(S^{-1}\overline{O}S\right)_{\tau_1\tau_2} = f_{\tau_1}'^{1/2} \overline{O}_{t_1 t_2} f_{\tau_2}'^{1/2}, \qquad t_i = f(\tau_i), \tag{C.3}$$

where the 'matrix products' are defined as $(S^{-1}\overline{O}S)_{\tau\tau'} = \int dt dt' S_{\tau,t}^{-1} \overline{O}_{tt'} S_{t',\tau'}$. Then from definition of the mapping $\mathcal{M}_\Delta$ it follows that

$$O_{\tau_1\tau_2} = f_{\tau_1}'^{\frac{\Delta-1}{2}} (S^{-1}\overline{O}S)_{\tau_1\tau_2} f_{\tau_2}'^{\frac{\Delta-1}{2}} \equiv \left(f'^{\frac{\Delta-1}{2}} S^{-1}\overline{O}S f'^{\frac{\Delta-1}{2}}\right)_{\tau_1\tau_2}. \tag{C.4}$$

Consequently, the inverse relation to above reads

$$\overline{O}_{t_1 t_2} = \left(F'^{\frac{1-\Delta}{2}} SOS^{-1} F'^{\frac{1-\Delta}{2}}\right)_{t_1 t_2}, \tag{C.5}$$

which shows that in the language of matrix algebra $\mathcal{M}_\Delta = F'^{\frac{1-\Delta}{2}} S$ so that one can write $\overline{O} = \mathcal{M}_\Delta O \mathcal{M}_\Delta^T$, and thereby our proposition is proved.

# D  Gradient expansion of the $G\Sigma$–action

In this Appendix we collect the technical details of our derivation of the AS action from the $G\Sigma$–functional, which was outlined in Sec. 2.4.

## D.1  Vertex operators

We begin by deriving the vertex operator $\rho$, which is the image of the time derivative operator with matrix elements $[\partial_\tau]_{\tau_1\tau_2} = \delta'(\tau_1 - \tau_2)$. From the definition (33), it follows that one can express $\rho_{t_1 t_2} = F_1'^{3/4} \delta'(F_1 - F_2) F_2'^{3/4}$, which requires understanding of the kernel $\delta'(F_1 - F_2)$. By acting with this kernel on an arbitrary function of time $f(t)$, one can verify that

$$\delta'(F_1 - F_2) = \frac{1}{F_1'} \delta'(t_1 - t_2) \frac{1}{F_2'} \tag{D.1}$$

holds. Hence, the latter gives us the matrix elements $\rho_{t_1 t_2} = F_1'^{-1/4} \delta'(t_1 - t_2) F_2'^{-1/4}$. One can further associate $\delta'(t_1 - t_2)$ with the kernel of a symmetrized differential operator $\frac{1}{2}(\overrightarrow{\partial_t} - \overleftarrow{\partial_t})$, whose action is defined similarly as in (56), but with $b = 1$. By setting $a \equiv F'^{-1/4} = \sqrt{b}$, we finally find

$$\rho = \frac{1}{2} a_t (\overrightarrow{\partial_t} a_t) - \frac{1}{2} (a_t \overleftarrow{\partial_t}) a_t = \frac{1}{2} (b_t \overrightarrow{\partial_t} - \overleftarrow{\partial_t} b_t), \tag{D.2}$$

as was stated in Eq. (56).

To derive the second vertex operator, the heat current density $j$, one needs to analyze how the time-local kinetic energy operator $\varepsilon_{\tau_1 \tau_2}^{x_1 x_2} = \varepsilon_{x_1 x_2} \delta_{\tau_1 \tau_2}$ changes under a substitution $\tau_i \to F(t_i, x)$. Here

$$\varepsilon_{xx'} = \int_k e^{ik(x-x')} \epsilon_k \tag{D.3}$$

is the position representation of the chiral dispersion relation. Further calculations below now become somewhat more involved. To start, we define the kinetic energy in the new time frame,

$$\mathcal{E}_{t_1 t_2}^{x_1 x_2} = \varepsilon_{x_1 x_2} \delta(F_{t_1, x_1} - F_{t_2, x_2}), \tag{D.4}$$

and evaluate its Wigner symbol with respect to spatial coordinates in a first-order gradient expansion. Using the integral representation for the $\delta$-function,

$$\mathcal{E}_{t_1 t_2}^{x_1 x_2} = \frac{1}{2\pi} \int e^{i\alpha F_1} \varepsilon_{x_1 x_2} e^{-i\alpha F_2} \, d\alpha, \qquad F_i \equiv F(t_i, x_i), \tag{D.5}$$

which properly orders the product of three operators, one can further apply the Moyal expansion to find,

$$
\begin{aligned}
\mathcal{E}_{t_1 t_2}(x, k) &= \epsilon_k \delta(F_1 - F_2) - \frac{1}{4\pi} \int (\partial_x F_1 + \partial_x F_2) \partial_k \epsilon_k \, e^{i\alpha(F_1 - F_2)} \alpha d\alpha + \mathcal{O}(\hbar^2) \\
&= \frac{\epsilon_k}{F_1'} \delta_{t_1 t_2} + \frac{i}{2} \partial_k \epsilon_k (\partial_x F_1 + \partial_x F_2) \delta'(F_1 - F_2) + \mathcal{O}(\hbar^2),
\end{aligned}
\tag{D.6}
$$

where we have now redefined $F_i$ to be $F_i = F(t_i, x)$ with $x$ being a coordinate variable of the Wigner transform. At this stage one can simplify $\delta'(F_1 - F_2)$ according to (D.1). Further, using the definition (53) with $\Delta = 3/2$, one finds for the heat current operator

$$
\begin{aligned}
j_{t_1 t_2}(x, k) &= F_1'^{3/4} \mathcal{E}_{t_1 t_2}(x, k) F_2'^{3/4} = \epsilon_k F_1'^{1/2} \delta_{t_1 t_2} \\
&+ \frac{i}{2} \partial_k \epsilon_k \times F_1'^{-1/4} (\partial_x F_1 \, \delta'(t_1 - t_2) + \delta'(t_1 - t_2) \partial_x F_2) F_2'^{-1/4}.
\end{aligned}
\tag{D.7}
$$

This expression can be further simplified. For that, we consider the kernel of an auxiliary operator,

$$J_{t_1 t_2} = g_{t_1} h_{t_1} \delta'(t_1 - t_2) g_{t_2} + g_{t_1} \delta'(t_1 - t_2) h_{t_2} g_{t_2}, \tag{D.8}$$

where, as before, $g_t$ and $h_t$ are two arbitrary functions with periodic boundary conditions in time. On substituting $\delta'(t_1 - t_2) \to \frac{1}{2}(\overrightarrow{\partial_t} - \overleftarrow{\partial_t})$, one finds (after some simplification) that $J_{t_1 t_2}$ is, in fact, the kernel of a simpler differential operator

$$J = \frac{1}{2}(g^2 h) \overrightarrow{\partial_t} - \frac{1}{2} \overleftarrow{\partial_t} (g^2 h). \tag{D.9}$$

In order to decode (D.7), one further sets $g \to F_t'^{-1/4} = \sqrt{b_t}$ and $h \to \partial_x F$, which leads to

$$j(x, k) = \epsilon_k / b + \frac{i}{2} \partial_k \epsilon_k \times \left( b \partial_x F \overrightarrow{\partial_t} - \overleftarrow{\partial_t} b \partial_x F \right) \equiv j_k^0(x) + j_k^1(x). \tag{D.10}$$

In this way, we reproduce the vertex operator $j_k(x)$, defined by Eq. (58) in the main text.

## D.2 Gradient correction to the Wigner symbol of the Green's function

In this subsection, we identify temporal gradient corrections to the Wigner symbol $\overline{\mathcal{G}}_{k,\epsilon}$ of the exact propagator $\overline{\mathcal{G}}$, which is defined by the relation $\overline{\mathcal{G}} = -(j^0 + \overline{\Sigma})^{-1}$, where $j^0_{k,\epsilon}(s,x) = \epsilon_k / b_{s,x}$ denotes the corresponding Wigner symbol of the current operator and $\overline{\Sigma}_\epsilon(s) = \Sigma(\epsilon; \Lambda / b_s)$ refers to the one of the self-energy.

We first note that if $A_\epsilon(s)$ and $B_\epsilon(s)$ are the Wigner symbols of two operators $\hat{A}_{t_1 t_2}$ and $\hat{B}_{t_1 t_2}$, then the Weyl product $A_\epsilon(s) \star B_\epsilon(s)$ corresponds to the Wigner symbol of $(AB)_{t_1 t_2}$. The same definition applies to the spatial domain. Using this framework, the exact propagator can be defined via the relation $\overline{\mathcal{G}}_{\epsilon,k}(s) \star h_{\epsilon,k}(s) = -1$, where $h_{\epsilon,k}(s)$ is the effective Hamiltonian (38). To resolve this equation, note that the Weyl product admits the following formal representation,

$$(A \star B)(\epsilon, s) = A_\epsilon(s) e^{-\frac{i\hbar}{2}(\overleftarrow{\partial_s}\overrightarrow{\partial_\epsilon} - \overleftarrow{\partial_\epsilon}\overrightarrow{\partial_s})} B_\epsilon(s), \tag{D.11}$$

which can be employed to construct its semiclassical expansion in powers of $\hbar$. Our immediate goal is to determine gradient corrections in time, as the inclusion of spatial gradients will be addressed separately (see Appendix D.4). Note that in the leading approximation, used consistently throughout the derivation of the AS action, the self-energy $\overline{\Sigma}_\epsilon(s)$ can be approximated by its SYK limit (16) and is thus independent of the 'slow' time $s$. Accordingly, it is instructive to evaluate the following restricted Weyl product:

$$\begin{aligned} A_\epsilon \star \phi(s) \star B_\epsilon &= A_\epsilon e^{\frac{i\hbar}{2}\overleftarrow{\partial_\epsilon}\overrightarrow{\partial_s}} \phi(s) e^{-\frac{i\hbar}{2}\overleftarrow{\partial_s}\overrightarrow{\partial_\epsilon}} B_\epsilon \\ &= A_\epsilon \phi(s) B_\epsilon - i\phi'(s) A_\epsilon \overleftrightarrow{\partial_\epsilon} B_\epsilon - \frac{1}{2}\phi''(s) A_\epsilon \overleftrightarrow{\square}_\epsilon B_\epsilon + \mathcal{O}(\hbar^3), \end{aligned} \tag{D.12}$$

where the first order derivative is properly anti-symmetrized, $\overleftrightarrow{\partial_\epsilon} = \frac{1}{2}(\overrightarrow{\partial_\epsilon} - \overleftarrow{\partial_\epsilon})$, while the second order differential operator is defined by the relation

$$A_\epsilon \overleftrightarrow{\square}_\epsilon B_\epsilon := \frac{d^2}{d\omega^2}\Big(A_{\epsilon+\omega/2} B_{\epsilon-\omega/2}\Big)\Big|_{\omega=0} = \frac{1}{4}A''_\epsilon B_\epsilon + \frac{1}{4}A_\epsilon B''_\epsilon - \frac{1}{2}A'_\epsilon B'_\epsilon. \tag{D.13}$$

The above considerations lead to the following simple rule: one may replace the Wigner symbol $j^0_{k,\epsilon}(s,x)$ by the corresponding differential operator,

$$j^0_{k,\epsilon}(s,x) \to \frac{\epsilon_k}{b} + \delta\hat{j}^0_k, \qquad \delta\hat{j}^0_k = -i\epsilon_k\left(\frac{1}{b}\right)' \overleftrightarrow{\partial_\epsilon} - \frac{\epsilon_k}{2}\left(\frac{1}{b}\right)'' \overleftrightarrow{\square}_\epsilon + \mathcal{O}(\hbar^3), \tag{D.14}$$

and approximately evaluate the Wigner symbol $\overline{\mathcal{G}}_{\epsilon,k}$ using a perturbation series over the gradient correction $\hat{\delta}j^0_k$ as follows,

$$\overline{\mathcal{G}}_{\epsilon,k} = \overline{G}_{\epsilon,k} + \overline{G}_{\epsilon,k}\delta\hat{j}^0_k\overline{G}_{\epsilon,k} + \overline{G}_{\epsilon,k}\delta\hat{j}^0_k\overline{G}_{\epsilon,k}\delta\hat{j}^0_k\overline{G}_{\epsilon,k} + \ldots, \tag{D.15}$$

where $\overline{G}_{\epsilon,k} \simeq -(\epsilon_k/b + \Sigma^0_\epsilon)^{-1}$ is the zeroth order Green's function. After straightforward simplifications, one finds:

$$\begin{aligned} \overline{\mathcal{G}}_{\epsilon,k} &= \overline{G}_{\epsilon,k} + \delta\overline{G}_{\epsilon,k} + \mathcal{O}(\hbar^3), \\ \delta\overline{G}_{\epsilon,k} &= -\frac{\epsilon_k}{2}\left(\frac{1}{b}\right)'' \overline{G}_{\epsilon,k}\overleftrightarrow{\square}_\epsilon\overline{G}_{\epsilon,k} + \frac{\epsilon_k^2}{2}\left(\frac{1}{b}\right)'^2\left(\partial_\epsilon\overline{G}_{\epsilon,k}\overleftrightarrow{\partial_\epsilon}\overline{G}^2_{\epsilon,k}\right). \end{aligned} \tag{D.16}$$

The correction of order $\mathcal{O}(\hbar)$ is absent here, as previously noticed in Eq. (39).

## D.3 Reparametrization invariant part of the $G\Sigma$-action

In this Appendix, we demonstrate that within the temperature range $T_\Lambda < T < J$, the action $S_0[\overline{G},\overline{\Sigma}]$, defined by Eq. (62), is reparametrization invariant up to gradient terms of order $\mathcal{O}(b'^2, bb'')$ and therefore does not contribute to the low-energy AS action. On the other hand, the explicit symmetry breaking introduced by the kinetic energy term $(\epsilon_k)$ in the microscopic Hamiltonian leads to the emergence of a non-local-in-time and relevant perturbation to the AS action in the infrared limit, i.e. at $T < T_\Lambda$. This observation is significant only in the case of Model II. We then proceed to determine the conditions under which such a relevant perturbation can be considered small, ensuring that the AS action remains a valid description of the chiral SYK model over a broad range of temperatures $T_* < T < T_\Lambda$, with the infrared temperature cutoff $T_*$ identified below.

For future reference, we represent Eq. (62) in the equivalent form:

$$S_*[\overline{G},\overline{\Sigma}] = \frac{N}{2}\operatorname{tr}\ln\overline{\mathcal{G}} - \frac{N}{2}I[\overline{G},\overline{\Sigma}], \qquad (D.17)$$

where $\overline{\mathcal{G}} = -(j_0 + \overline{\Sigma})^{-1}$ is the exact Green's function of the effective Hamiltonian (60). For the convenience of the reader, we repeat the transformed Green's function and self-energy expressed in terms of their mean-field solutions (25), which read

$$\overline{G}^{xx}_{t_1 t_2} = G(t_1 - t_2; \Lambda^x_{t_1 t_2}), \qquad \overline{\Sigma}^{xx'}_{t_1 t_2} = \Sigma(t_1 - t_2; \Lambda^x_{t_1 t_2})\delta_{xx'}, \qquad (D.18)$$

where the rescaled kinetic energy scale reads $\Lambda^x_{t_1 t_2} = \Lambda(b_1 b_2)^{-1/2}$, and we abbreviated $b_i = b(t_i, x)$. We also introduce a momentum-resolved Green's function of the mean-field solution,

$$G_{\epsilon,k}(\Lambda) = -[\epsilon_k + \Sigma(\epsilon; \Lambda)]^{-1}. \qquad (D.19)$$

It can be used to construct the approximate Wigner symbol of the propagator $\overline{\mathcal{G}}$ through the relation

$$\overline{\mathcal{G}}_{\epsilon,k}(\Lambda) = G_{\epsilon,k}(\Lambda/b) + \mathcal{O}(\hbar^2), \qquad b = b(x,s) \qquad (D.20)$$

which is valid up to second-order gradient corrections. The latter were identified in the previous section and are explicitly given by Eq. (D.16).

### D.3.1 Local approximation

With the above definitions at hand, one can now check that the action (D.17) is $b$-independent within a *local approximation* (possible corrections due to temporal gradients will be accounted for later). To this end, we set $\Lambda^x_{t_1 t_2} \to \Lambda/b_{s,x}$, where $s = (t_1 + t_2)/2$ is the center of mass time and approximate $\ln\det\overline{\mathcal{G}}$ using the Moyal symbol (D.20). These two steps lead to the action,

$$S_*(\Lambda) = \frac{N}{2}\int_{s,x}\int_{k,\epsilon}\ln G_{\epsilon,k}(\Lambda/b) - \frac{NJ^2}{8k_0^3}\int_{x,t_{1,2}}[\overline{G}^{xx}_{t_1 t_2}]^4 - \frac{N}{2}\int_{x,t_{1,2}}\overline{G}^{xx}_{t_1 t_2}\overline{\Sigma}^{xx}_{t_2 t_1}, \qquad (D.21)$$

which should be regarded as a function of the kinetic energy scale $\Lambda$ as indicated above. For $\Lambda \to 0$, the action equals the $f$-independent constant,

$$S_*(0) = -(N/2)\operatorname{tr}\ln\Sigma^0 - (N/2)I[\Sigma^0, G^0], \qquad (D.22)$$

with $(\Sigma^0, G^0)$ being the SYK saddle-point (7). One can then find the flow equation when $\Lambda$ changes:

$$\Lambda\partial_\Lambda S_*(\Lambda) = \frac{N}{2}\int_{\substack{|\epsilon|<J/b^2}}\frac{d\epsilon}{2\pi}\int_{s,x,k}(\epsilon_k/b)G_{\epsilon,k}(\Lambda/b). \qquad (D.23)$$

where it was used that $\epsilon_k \propto \Lambda k^\gamma$. During the derivation of the above relation, one can disregard partial derivatives $\partial_\Lambda \overline{G}$ and $\partial_\Lambda \overline{\Sigma}$, as within the local approximation, the pair $(\overline{G}, \overline{\Sigma})$ becomes the exact saddle-point of the action (D.21). We have also restricted the energy integral using a reparametrization dependent UV cut-off. This regularization scheme was previously used to derive the Schwarzian action from the SYK model [14], and we will provide a few comments on it below. To proceed, note that the mean-field self-energy exhibits the following scaling property in the energy representation,

$$\Sigma(\epsilon; \Lambda/b) = b^{-1}\Sigma(\epsilon b^2; \Lambda), \tag{D.24}$$

which follows from Eq. (26). Hence, for the Green's function one can write the relation

$$G_{\epsilon,k}(\Lambda/b) = -b\left[\epsilon_k + \Sigma(\epsilon b^2; \Lambda)\right]^{-1} = b G_{\epsilon b^2, k}(\Lambda). \tag{D.25}$$

On further introducing a new energy integration variable, $\widetilde{\epsilon} = \epsilon b^2$, one arrives at the flow Eq. (D.23) in the $b$-independent form,

$$\Lambda \partial_\Lambda S_*(\Lambda) = \frac{N}{2} \int d\tau dx \int_{|\widetilde{\epsilon}|<J} \frac{d\widetilde{\epsilon} dk}{(2\pi)^2} \epsilon_k G_{\widetilde{\epsilon}, k}(\Lambda), \tag{D.26}$$

where it was used that $d\tau = F'ds = ds/b^2$. Hence, one concludes that $S_*(\Lambda)$ is reparametrization invariant as well.

Let us now comment on the origin of the reparametrization-dependent cut-off. To this end, consider a typical operator bilocal in time with scaling dimension $\Delta$:

$$O^\Delta_{\tau_1\tau_2}(x) = \left[\frac{f_1'f_2'}{(f_1-f_2)^2}\right]^{\Delta/2} \rightarrow \quad \mathrm{Re}\left[\frac{f_1'f_2'}{(f_1-f_2+i\delta_{12})^2}\right]^{\Delta/2}. \tag{D.27}$$

The singular kernel here requires a regularization at short times, which can be achieved by shifting the time argument into the complex plane, as it is reflected in the expression above, with some cut-off scale $\delta_{12} \sim J^{-1}$. Further on, one should require that in the UV limit, when $t_1 \rightarrow t_2$, the $f$-dependence drops out from $O_{\tau_1\tau_2}(x)$ in the leading order (the next-to-leading term produces the Schwarzian), since reparametrizations, by definition, are an IR effect. The latter suggests to define $\delta_{12} = (f_1'f_2')^{1/2}/J$. In the local approximation, it becomes $\delta = b^2/J$, which also translates into the UV cut-off $J' = J/b^2$ in the energy domain.

### D.3.2 Inclusion of gradient corrections

We now address the fate of gradient corrections, appearing when one goes beyond the local approximation used above. These corrections can be categorized into two types. The first type is related to the substitution $\Lambda^x_{t_1t_2} \rightarrow \Lambda/b_{s,x}$, employed in the previous subsection. With $t_{1,2} = s \pm t/2$, this substitution introduces an error:

$$\delta\Lambda = \Lambda^x_{t_1t_2} - \Lambda/b_{s,x} = (\Lambda t^2) \times \frac{b'^2 - bb''}{8b^3} + \mathcal{O}(t^4), \qquad b' = \partial_s b, \tag{D.28}$$

which in turn results in the following variation to the action (D.17),

$$\delta S_*[\overline{G}, \overline{\Sigma}] = \mathrm{Tr}\left(\frac{\delta S_*}{\delta\Sigma}\bigg|_{X_*} \partial_\Lambda \overline{\Sigma} + \frac{\delta S_*}{\delta G}\bigg|_{X_*} \partial_\Lambda \overline{G}\right)\delta\Lambda \rightarrow 0, \tag{D.29}$$

where $X_* = (\overline{G}, \overline{\Sigma})$ is the saddle point solution within the local approximation, namely $\overline{G}^x_{t_1 t_2} = G(t, \Lambda/b_{s,x})$, and the same applies to $\overline{\Sigma}^x_{t_1 t_2}$. However, this variation vanishes because the mean-field solution $X_*$ is an exact saddle point of the action (D.17) for an arbitrary reparametrization $b$; see also the discussion in Sec. 2.3. It is worth noting that this observation parallels the remark made during the derivation of Eq. (D.23). Consequently, up to terms of order $\mathcal{O}(b'^2, bb'')$, there is no contribution to the low-energy action in this case.

Another type of gradient corrections, not yet included into our analysis, arises when the action $S_L[b] = (N/2)\,\mathrm{tr}\ln\mathcal{G}$ in Eq. (D.17) is expanded in terms of the variation $\delta\mathcal{G}$, which defines the exact Wigner symbol (D.16) of the Green's function. A straightforward implementation of this program encounters infrared divergences at temperature scales well below $T_\Lambda$, as the direct gradient expansion in terms of slow and fast degrees of freedom becomes unjustified in this limit. Since our analysis of Model I — with a linear dispersion $\epsilon_k \sim k$ — is restricted to the high temperature range $T_\Lambda < T < J$, the discussion that follows is only relevant for Model II, which features flat bands.

When properly regularized, the logarithmic action $S_L[b]$ expanded over the gradient correction $\delta\overline{\mathcal{G}}[b]$ generates at small temperatures $T \ll T_\Lambda$ an infinite series of $SL(2,\mathbb{R})$-invariant and nonlocal-in-time operators. The simplest among them takes the following form:

$$\mathcal{S}[f] \sim -N k_1 T_\Lambda^{-1/2\gamma} \int_{x,\tau_{1,2}} \beta^\Delta \mathcal{O}^\Delta_{\tau_1 \tau_2}(x), \qquad \Delta = 2 + 1/2\gamma, \qquad (\text{D}.30)$$

with the time-bilocal operator defined by Eq. (98). The tilde symbol ($\sim$) indicates that here and in what follows we omit all numerical constants of order unity. Note that the above action stems entirely from itinerant chiral Majorana modes, which explains its prefactor that includes a small momentum $k_1$, see Fig. 2. The overall coupling constant here has a positive scaling dimension, $\mathcal{D} = 1 - 1/2\gamma \geq 1/2$ (we set $[x] = [\tau] = -1$), indicating the relevance of this perturbation to the AS action in the infra-red limit. We relegate a derivation of the result (D.30) and its holographic interpretation to our subsequent communication and proceed below with formulating the conditions under which such a relevant perturbation can be considered small compared to the AS action.

Treating the bilocal operator in Eq. (D.30) as a perturbation, we can average it over quantum fluctuations governed by the AS action, $\langle \mathcal{O}^\Delta_{\tau_1 \tau_2}(x) \rangle_{\mathrm{AS}}$, to obtain a small correction to the free energy. The technical procedure for calculating such averages was outlined in Sec. 4.3. We know that the engineering scaling dimension $\Delta$ undergoes transmutation to the quantum dimension $\Delta_Q = \Delta + 3\Delta(1-\Delta)/C$. As a result, the free energy per length acquires the non-analytic dependence on temperature,

$$\frac{\mathcal{F}(T)}{L} = -\frac{\pi C}{12} \times \left[ \frac{T^2}{u_0} \ln\frac{J}{T} + \frac{T_\Lambda^2}{u_1}\left(\frac{T}{T_\Lambda}\right)^\delta \right], \qquad \delta = \Delta_Q - 1. \qquad (\text{D}.31)$$

Here, the first term arises from the Schwarzian action with a temperature dependent velocity (44), and we have introduced a second velocity scale,

$$u_1 \sim \frac{C T_\Lambda}{N k_1} \sim \frac{\Lambda^3}{J^2 k_1} \ln\frac{J}{T_\Lambda}, \qquad (\text{D}.32)$$

to characterize the magnitude of the perturbative correction. Initially, the latter can be considered small, provided the condition $u_0/u_1 \sim (J/T_\Lambda)(k_1/k_0) \ll 1$ is satisfied. This ensures that at the crossover temperature scale $T_\Lambda$, the Schwarzian contribution to the free energy dominates. Furthermore, based on physical grounds, the exponent $\delta$ must be positive, which imposes a lower bound on the acceptable central charge,

$$C > C_* = 6 + 3/(2\gamma). \qquad (\text{D}.33)$$

On the other hand, the upper bound for the exponent, $\delta < 3/2$, holds by construction. As one can see from the derived relation (D.31), the perturbation theory breaks down around the infra-red temperature scale $T_* \ll T_\Lambda$, which can be estimated from the following implicit relation,

$$\frac{T_*}{T_\Lambda} = \left( \frac{u_0}{u_1} \ln^{-1} \frac{J}{T_*} \right)^{\frac{1}{2-\delta}} . \tag{D.34}$$

which is found by approximately equalizing two different contributions to $\mathcal{F}(T)$. At smaller temperatures, $T < T_*$, the AS action deformed by the relevant operator (D.30) calls for a non-perturbative treatment, possibly within the RG scheme similar to the one developed earlier for the model of granular SYK arrays [48].

To summarize this subsection, we have verified that if a velocity ratio satisfies the condition $u_0/u_1 \ll 1$, then a holographic duality between the chiral SYK model comprising flat bands and the AS action may hold in the parametrically wide range of temperatures $T_* \ll T \ll J$.

## D.4   First-order gradient terms

In this subsection, we are aiming to analyze linear in gradient terms,

$$S_\rho[f] + S_j[f] = \frac{N}{2} \mathrm{tr}(\rho \overline{\mathcal{G}}) + \frac{N}{2} \mathrm{tr}(j^1 \overline{\mathcal{G}}), \tag{D.35}$$

which are obtained from the first-order expansion of the action (61). We start our calculations from the action $S_\rho[f]$, which involves the energy density operator (56). Following Pruisken, we double the power of the Green function in this action. To this end, one introduces the propagator $\overline{\mathcal{G}}(\mu)$, which depends on an auxiliary energy $\mu$, so that one can write

$$\overline{\mathcal{G}} = -\int_{-\infty}^0 d\mu \overline{\mathcal{G}}^2(\mu), \qquad \overline{\mathcal{G}}(\mu) = (\mu - j_0 - \overline{\Sigma})^{-1}. \tag{D.36}$$

Based on the above identity, the action $S_\rho[f]$ can be equivalently rewritten as a sum of two (right and left) contributions:

$$S_\rho[f] = S_\rho^R[f] + S_\rho^L[f] = -\frac{N}{4} \int_{-\infty}^0 d\mu \, \mathrm{tr}\left[ \overline{\mathcal{G}}(\mu)(b\overrightarrow{\partial_t} - \overleftarrow{\partial_t} b)\overline{\mathcal{G}}(\mu) \right], \tag{D.37}$$

where the trace operation implies an integration over temporal- and spatial indices. From here we switch to the Wigner representation by writing

$$S_\rho^R[f] = -\frac{N}{4} \int_{-\infty}^0 d\mu \int_{s,x} \int_{k,\epsilon} \overline{\mathcal{G}}_{\epsilon,k}(\mu) \star b_s \star (-i\epsilon + \tfrac{1}{2}\overrightarrow{\partial_s})\overline{\mathcal{G}}_{\epsilon,k}(\mu), \tag{D.38}$$

$$S_\rho^L[f] = -\frac{N}{4} \int_{-\infty}^0 d\mu \int_{s,x} \int_{k,\epsilon} \overline{\mathcal{G}}_{\epsilon,k}(\mu)(-i\epsilon - \tfrac{1}{2}\overleftarrow{\partial_s}) \star b_s \star \overline{\mathcal{G}}_{\epsilon,k}(\mu), \tag{D.39}$$

where it was used that time derivative operators, when acting in the Wigner representation, become

$$\partial_{t_1} = -i\epsilon + \tfrac{1}{2}\partial_s, \qquad \partial_{t_2} = i\epsilon + \tfrac{1}{2}\partial_s. \tag{D.40}$$

In what follows we are aiming to employ the 1st order Moyal expansion to the convolutions $\overline{\mathcal{G}}(\mu) \star b_s \star \partial_s \overline{\mathcal{G}}(\mu)$ and $\partial_s \overline{\mathcal{G}} \star b_s \star \overline{\mathcal{G}}(\mu)$ stemming, respectively, from the right- and left actions. When doing so, one may substitute the exact Wigner symbol $\overline{\mathcal{G}}_{\epsilon,k}(\mu)$ by its local approximation,

$$\overline{\mathcal{G}}_{\epsilon,k}(\mu; \Lambda) \equiv \mathcal{G}_{\epsilon,k}(\mu; \Lambda/b) = [\mu - \epsilon_k/b - \Sigma(\epsilon; \Lambda/b)]^{-1}, \tag{D.41}$$

cf. Eqs. (D.19) and (D.20). Similarly to Eq. (D.19), we keep the spatio-temporal dependence of $G_{\epsilon,k}(\mu;\Lambda)$ implicit. To proceed, we introduce the time derivative

$$F(\mu) = \partial_s \overline{G}(\mu) = \overline{G}^2(\mu)\epsilon_k(1/b_s)', \tag{D.42}$$

where further on all dependencies on $(k,\epsilon)$ and $(x,s)$ are omitted for brevity. Then, the first-order Moyal expansion takes the form

$$\overline{\mathcal{G}}(\mu) \star b_s \star \partial_s \overline{G}(\mu) = \overline{\mathcal{G}}(\mu)b_s\partial_s\overline{G}(\mu) + \frac{i}{2}\left[\overline{G}(\mu)\partial_k F(\mu) - F(\mu)\partial_k \overline{G}(\mu)\right]\partial_x b_s + \mathcal{O}(\hbar^2), \tag{D.43}$$

with a similar expression being valid for the other convolution, where $\overline{\mathcal{G}}$ and its partial derivative are swapped. Taking into account that the momentum derivative simplifies to

$$\partial_k \overline{G}(\mu) = \overline{G}^2(\mu)\partial_k\epsilon_k/b, \tag{D.44}$$

we find the following gradient expansion for the action $S_\rho[f]$:

$$S_\rho[f] = S_\rho^{\mathrm{UV}}[f] + \frac{iN}{8}\int_{-\infty}^0 d\mu \int_{s,x}\int_{k,\epsilon}\left[\overline{G}^3(\mu)\partial_k\epsilon_k \times \frac{b'\partial_x b}{b^2} + \overline{G}^4(\mu)\epsilon_k\partial_k\epsilon_k \times \frac{b'\partial_x b}{b^3}\right]. \tag{D.45}$$

Here, the first contribution $S_\rho^{\mathrm{UV}}[f]$ arises from the leading-order Moyal expansion. We will analyze it at the end of this subsection. For now, we focus on the second contribution, which involves a spatial gradient $\partial_x b$. After integration over the auxiliary energy $\mu$, it becomes

$$\Delta S_\rho[f] = -\frac{iN}{8}\int_{k,\epsilon}\int_{s,x}\left[G_{\epsilon,k}^2(\Lambda/b)\partial_k\epsilon_k \times \frac{b'\partial_x b}{2b^2} + G_{\epsilon,k}^3(\Lambda/b)\epsilon_k\partial_k\epsilon_k \times \frac{b'\partial_x b}{3b^3}\right], \tag{D.46}$$

where the mean-field Green's function $G_{\epsilon,k}(\Lambda)$ was defined in Eq. (D.19). At this stage, it is important to rescale the energy integration variable by setting $\epsilon = \widetilde{\epsilon}/b^2$. Using the transformation law (D.25) for the Green's function, one finds that energy-momentum and space-time integrations are factorized, yielding

$$\Delta S_\rho[f] = -\frac{iN}{8}\int_{k,\widetilde{\epsilon}}\partial_k\epsilon_k\left[\frac{1}{2}G_{\widetilde{\epsilon},k}^2(\Lambda) + \frac{1}{3}G_{\widetilde{\epsilon},k}^3(\Lambda)\,\epsilon_k\right] \times \int_{s,x}\frac{b'\partial_x b}{b^2}. \tag{D.47}$$

To extract the leading logarithmic expression for the coupling constant from here, one observes that at large energy, the Green's function scales as $G_{\epsilon,k}(\Lambda) \sim |\epsilon|^{-1/2}$. Hence, the first term in Eq. (D.47) is a logarithmic integral, while the second one gives a subleading contribution. We further note that in the relevant range of energies, $T_\Lambda \ll |\epsilon| \ll J$, the kinetic energy is negligible as compared to the self-energy, i.e. $|\epsilon_k| < \Lambda \ll \Sigma_\epsilon^0$, where $\Sigma_\epsilon^0 \propto \sqrt{|\epsilon|}$ is the SYK self-energy (16). Approximating the Green's function as $G_{\epsilon,k}(\Lambda) \simeq -(\Sigma_\epsilon^0)^{-1}$, one then arrives at

$$\Delta S_\rho[f] \simeq -\frac{iN}{16}\int\frac{d\epsilon(k)}{(2\pi)^2}\int_{T_\Lambda<|\widetilde{\epsilon}|<J}\frac{d\widetilde{\epsilon}}{(\Sigma_{\widetilde{\epsilon}}^0)^2} \times \int_{s,x}\frac{b'\partial_x b}{b^2} = \frac{iC}{12\pi}\int_{s,x}\frac{b'\partial_x b}{b^2}, \tag{D.48}$$

where $C$ is the central charge given by Eq. (43). In what follows (see Subsection D.5), we will interpret the above contribution as a counter-term, which contributes to the imaginary part of the action $S_\pm^{\mathrm{AS}}[b]$.

We now return to the first contribution in Eq. (D.45). After the integration over $\mu$ it reduces to the following simple expression,

$$S_\rho^{\mathrm{UV}}[f] = -\frac{iN}{2}\int_{s,x}\int_{k,\epsilon}(b\epsilon)\mathcal{G}_{\epsilon,k}, \tag{D.49}$$

which can be shown to be a $f$-independent constant. The way to demonstrate it follows the analysis of Sec. D.3. We start by substituting the exact Wigner symbol $\mathcal{G}_{\epsilon,k}$ by its leading approximation $G_{\epsilon,k}(\Lambda/b)$. For this choice, by restricting the energy integration to $|\epsilon| < J/b^2$ and introducing $\epsilon = \widetilde{\epsilon}/b^2$ as before, one employs the scaling property (D.41) of the Green's function to find:

$$S_\rho^{\mathrm{UV}}[f] = -\frac{iN}{2} \int \frac{dsdx}{b^2} \times \int_k \int_{|\widetilde{\epsilon}|<J} \widetilde{\epsilon}\, G_{\widetilde{\epsilon},k}(\Lambda) \propto -\frac{iN}{2} \int d\tau dx, \qquad (D.50)$$

where the relation $d\tau = F'ds = ds/b^2$ was used. Therefore, the spatio-temporal integral produces a $b$-independent result as it was claimed above.

We next show that a variation of the above result due to gradient corrections (D.16) vanishes, i.e.

$$\int_{s,x} \int_{k,\epsilon} (b\epsilon)\, \delta\mathcal{G}_{\epsilon,k} = 0, \qquad (D.51)$$

which follows from the analytic structure of the energy integral. For the sake of illustration, consider the first term contributing to the gradient correction of the propagator,

$$\delta\overline{\mathcal{G}}_{\epsilon,k}^{(1)} = -\frac{\epsilon_k}{2}\left(\frac{1}{b}\right)'' \overline{G}_{\epsilon,k} \overset{\longleftrightarrow}{\Box}_\epsilon \overline{G}_{\epsilon,k}. \qquad (D.52)$$

On changing the energy integration variable to $\epsilon = \widetilde{\epsilon}/b^2$ and integrating by parts a few times, we can transform (D.51) to the following integral,

$$\mathcal{J}^{(1)} \propto \int_{s,x} b'^2 \int_{\widetilde{\epsilon},k} \epsilon_k \widetilde{\epsilon}\,[\partial_{\widetilde{\epsilon}} G_{\widetilde{\epsilon},k}(\Lambda)]^2, \qquad (D.53)$$

which nominally looks like a contribution to the real part of the AS action (50). We further employ our standard approximation by using the SYK result $\Sigma_\epsilon^0$ for the self-energy, and change the integration variable to $\widetilde{\epsilon} = \mathrm{sgn}(\lambda)\lambda^2/J$, which yeilds

$$\mathcal{J}^{(1)} \propto \int_{s,x} b'^2 \int_k \epsilon_k \int_{-\infty}^{+\infty} G_{\lambda,k}^4 \lambda d\lambda \to 0, \qquad -G_{\lambda,k} = \frac{1}{\epsilon_k - i(\lambda/\pi)}. \qquad (D.54)$$

For a given momentum $k$, the integral over $\lambda$ now evaluates to zero due to simple analytic properties of the Green's function $G_{\lambda,k}$. The analysis of the second gradient correction to the Wigner symbol $\overline{\mathcal{G}}_{\epsilon,k}$,

$$\delta\overline{\mathcal{G}}_{\epsilon,k}^{(2)} = \frac{\epsilon_k^2}{2}\left(\frac{1}{b}\right)'^2 \left(\partial_\epsilon \overline{G}_{\epsilon,k} \overset{\longleftrightarrow}{\partial}_\epsilon \overline{G}_{\epsilon,k}^2\right), \qquad (D.55)$$

can be accomplished along the same lines as above, thereby substantiating the validity of Eq. (D.51).

We further explore the action $S_j[f]$ following the same principles. Since the current operator $j^1$ already contains one spatial derivative, it is sufficient to use the local Moyal approximation. Taking into account the relation between the two Wigner symbols, $j_{\epsilon,k}^1 = i(\partial_k \epsilon_k \partial_x F)\rho_{\epsilon,k}$, see Eq. (64), one arrives at the action

$$S_j^{\mathrm{UV}}[f] = \frac{N}{2} \int_{s,x} \int_{k,\epsilon} j_{\epsilon,k}^1 G_{\epsilon,k}(\Lambda/b) = \frac{N}{2} \int_{s,x} \partial_x F F' \times \int_k \int_{|\widetilde{\epsilon}|<J} \widetilde{\epsilon}\partial_k \epsilon_k\, G_{\widetilde{\epsilon},k}(\Lambda), \qquad (D.56)$$

which is a counterpart of Eqs. (D.49) and (D.50). However, this result corresponds neither to any of the contributions to the desired AS action (50), nor is a full derivative. We will demonstrate how it cancels in combination with higher-order terms in the gradient expansion in the following subsection.

## D.5 Second-order gradient terms

In this Appendix, we continue with the second-order expansion over gradients, $\rho$ and $j^1$, in the fluctuation action $S_{\mathrm{fl}}[f]$, see Eq. (61). First, we discuss how this expansion can be expressed in the Wigner representation and consider a typical second-order term $S_{\rho\rho}[f]$. Using the definition of the operator $\rho$, see Eq. (56), one can rewrite this piece of the action as

$$S_{\rho\rho}[f] = \int b_{t_1} \Pi_{t_1 t_2} b_{t_2} d^2 t dx, \qquad (D.57)$$

with $\Pi_{t_1 t_2}$ being a generalized polarization operator. Its most general expression takes the following form,

$$\Pi_{t_1 t_2} = -\frac{N}{4} \int_k \overline{\mathcal{G}}_{t_1 t_2}(k) \overleftrightarrow{\partial}_{t_1} \overleftrightarrow{\partial}_{t_2} \overline{\mathcal{G}}_{t_2 t_1}(k), \qquad \overleftrightarrow{\partial}_t := \frac{1}{2}(\overrightarrow{\partial}_t - \overleftarrow{\partial}_t). \qquad (D.58)$$

Here $\overline{\mathcal{G}}_{t_1 t_2}(k)$ refers to the Wigner symbol of the propagator $\overline{\mathcal{G}}$ w.r.t. position coordinates, and the expression (D.57) is written in the leading order Moyal expansion, assuming $b$ to change adiabatically in space. It is a reasonable approximation since the polarization operator already contains two temporal gradients. We can simplify (D.58) further by using the full Wigner symbol of the Green's function, $\overline{\mathcal{G}}_{\epsilon,k}(s,x)$, evaluated with respect to both temporal- and spatial indices. Then, after some simple algebra, one can check that the polarization operator $\Pi_{t_1 t_2}$ in the Wigner representation acquires the following form,

$$\Pi_\omega = -\frac{N}{4} \int_{k,\epsilon} \overline{\mathcal{G}}_{\epsilon_+,k}\left(\epsilon^2 + \tfrac{1}{4}\overleftrightarrow{\square}_s\right)\overline{\mathcal{G}}_{\epsilon_-,k} = \Pi_\omega^{\mathrm{I}} + \Pi_\omega^{\mathrm{II}}, \quad \epsilon_\pm = \epsilon \pm \tfrac{1}{2}\omega, \qquad (D.59)$$

where the 1st and 2nd terms above refer, respectively, to $\epsilon^2$ and $s$-derivative parts. Here, all dependencies on the slow time $s$ and the position $x$ were left implicit for brevity. We have also introduced the second-order differential operator $\overleftrightarrow{\square}_s$ which needs both left and right functions to act on and is defined as

$$g(s)\overleftrightarrow{\square}_s h(s) := \frac{d^2}{dr^2}\left(g(s+\tfrac{1}{2}r)h(s-\tfrac{1}{2}r)\right)\Big|_{r=0} = \frac{1}{4}g''_{s^2}h + \frac{1}{4}gh''_{s^2} - \frac{1}{2}g'_s h'_s. \qquad (D.60)$$

Similarly, one can analyze other terms in the gradient expansion, i.e. $S_{j\rho}$ and $S_{jj}$.

### D.5.1 Evaluation of the Schwarzian action

In the following two subsections, we demonstrate how to derive the AS action within the leading logarithmic approximation. Specifically, we show that in order to determine coupling constants of the AS action with logarithmic accuracy, it suffices to use the local approximation for the exact Wigner symbol of the Green's functions defining the polarization operator (D.59) [see also the comment around Eq. (D.41)]. That is, one can substitute $\overline{\mathcal{G}}_{\epsilon,k} \to G_{\epsilon,k}(\Lambda/b)$, as suggested in Eq. (D.20).

We start from the Schwarzian piece, which constitutes the real part in the action $S_\pm^{\mathrm{AS}}[b]$ and originates from $S_{\rho\rho}[f]$, see Eq. (D.57). It is more instructive to discuss Model II first. Here the main contribution comes from the flat band region, $|k| < k_0$, where the Green's function $\overline{\mathcal{G}}_{\epsilon,k}(\Lambda/b)$ simplifies to the SYK solution $G_\epsilon^0$, see Eq. (16), and becomes $k$ (momentum) and $s$ (slow time) independent. In Fourier space one obtains,

$$S_{\rho\rho}[f] = \int dx \sum_\omega b_\omega \Pi_\omega b_{-\omega}, \quad \Pi_\omega = -\frac{N}{4}\int_{k,\epsilon} G_{\epsilon_+}^0 G_{\epsilon_-}^0 \epsilon^2 = \Pi_0 + \frac{\omega^2}{2}\Pi_0'' + \dots \qquad (D.61)$$

The $\Pi_0$-contribution here is UV-divergent and we use the parametrization dependent cut-off scheme discussed in Appendix D.3 to handle it, which gives

$$\Pi_0 \propto \int_0^{J/b^2} \epsilon \, d\epsilon \stackrel{\widetilde{\epsilon}=\epsilon b^2}{=} \frac{1}{b^4} \int_0^J \widetilde{\epsilon} \, d\widetilde{\epsilon} \propto \frac{J^2}{b^4}. \tag{D.62}$$

We then see that it contributes a constant to the action

$$S_{\rho\rho}^0 \propto \int \frac{dx\,ds}{b^2} = \int dx\,ds\,F' = \int dx\,d\tau = \beta L. \tag{D.63}$$

On other hand, the second-order order frequency expansion of the polarization operator evaluates to

$$\frac{1}{2}\Pi_0'' = -\frac{N}{8} \int_{k,\epsilon} (G_{\epsilon,k}^0 \overleftrightarrow{\square}_\omega G_{\epsilon,k}^0) \epsilon^2 \stackrel{|k|\le k_0}{=} -\frac{N}{32} \int_{k,\epsilon} (\Sigma_\epsilon^0)^{-2} = \frac{Nk_0}{32J} \int_{\sim T}^J \frac{d\epsilon}{\epsilon} = \frac{Nk_0}{32J} \ln \frac{J}{T}, \tag{D.64}$$

where the logarithmic infra-red singularity was cut by the temperature $T$. The above frequency expansion brings the Schwarzian action,

$$S_{\rho\rho}[f] = \frac{1}{2}\Pi_0'' \int dx \sum_\omega b_\omega \omega^2 b_{-\omega} = \frac{1}{2}\Pi_0'' \int_{s,x} b'^2 = -\frac{Nk_0}{64J} \ln \frac{J}{T} \int_{x,\tau} \{f,\tau\}. \tag{D.65}$$

In the last transformation results of Sec. 2.4 were used, see Eq. (50). Comparing $S_{\rho\rho}[f]$ to the real part of the AS action, one derives the relation

$$\frac{C}{u(T)} = \left(\frac{3\pi}{8}\right) \frac{Nk_0}{J} \ln \frac{J}{T}, \tag{D.66}$$

which fixes the ratio of a central charge $C$ to the temperature dependent velocity $u(T)$. One can also easily analyze the contributions stemming from the small range of momenta $k_0 < |k| < \pi/a$, where $\epsilon_k$ is nonvanishing. Here, the term proportional to $\epsilon^2$ in the polarization operator (D.59) produces a logarithmic correction to $S_{\rho\rho}[f]$, which scales as $(Nk_1/J)\ln(J/T_\Lambda)$. It is negligible compared to the main contribution (D.65). The contributions originating from the $\overleftrightarrow{\square}_s$ - term in Eq. (D.59) are even smaller. It can be shown that their functional dependence on the reparametrizations reduces to the Schwarzian form, as discussed above, and at the same time corresponding diagrams defining the prefactor are all IR convergent and free from UV logarithmic divergencies (see also additional comments in the end of this subsection).

We now turn to the analysis of Model I. Similar to the discussion above, the primary contribution to the Schwarzian action arises from the first part of the polarization operator. Here, we begin with the local approximation for the propagators, employing Eq. (D.20), which yields

$$\Pi_\omega^I = -\frac{N}{4} \int_{k,\epsilon} \epsilon^2 G_{\epsilon_+,k}(\Lambda/b) G_{\epsilon_-,k}(\Lambda/b) = \Pi_0^I + \frac{\omega^2}{2}(\Pi^I)_0'' + \ldots \tag{D.67}$$

The zero frequency piece, after rescaling $\epsilon = \widetilde{\epsilon}/b^2$ and using the transformation law (D.25), evaluates to

$$\Pi_0^I \propto -\frac{N}{b^4} \int_k \int_{|\widetilde{\epsilon}|<J} \widetilde{\epsilon}^2 G_{\widetilde{\epsilon},k}^2(\Lambda) \propto \frac{\text{Const}}{b^4}. \tag{D.68}$$

It contributes only an inessential constant to the overall action, see Eq. (D.63). In turn, for the frequency-dependent piece, we obtain with the logarithmic accuracy

$$\frac{1}{2}(\Pi^{\mathrm{I}})_0'' = -\frac{N}{8}\int_{k,\epsilon}\Big[G_{\epsilon,k}(\Lambda/b)\overset{\leftrightarrow}{\Box}_\epsilon G_{\epsilon,k}(\Lambda/b)\Big]\epsilon^2\overset{\widetilde{\epsilon}=\epsilon b^2}{=} \tag{D.69}$$

$$-\frac{N}{8}\int_{k,\widetilde{\epsilon}}\Big[G_{\widetilde{\epsilon},k}(\Lambda)\overset{\leftrightarrow}{\Box}_{\widetilde{\epsilon}} G_{\widetilde{\epsilon},k}(\Lambda)\Big]\widetilde{\epsilon}^2 \simeq -\frac{Nk_0}{32\pi^2}\int_{\sim T_\Lambda}^{J}\frac{d\widetilde{\epsilon}}{(\Sigma_{\widetilde{\epsilon}}^0)^2}=\frac{Nk_0}{32J}\ln\frac{J}{T_\Lambda}.$$

To derive this result, we employed (D.25) to eliminate the $b$-dependence from the integral and substituted $G_{\epsilon,k}^{-1}\to-\Sigma_\epsilon^0$, which is valid in the high-energy limit $T_\Lambda\ll\epsilon\ll J$. We then relied on the fact that the resulting infra-red logarithmic divergence in this case is regularized by the finite kinetic energy $\epsilon_k$ and thus is effectively cut-off at the scale $T_\Lambda$. The latter implies the relation

$$\frac{C}{u}=\left(\frac{3\pi}{8}\right)\frac{Nk_0}{J}\ln\frac{J}{T_\Lambda}, \tag{D.70}$$

where the velocity $u$ turns out to be temperature independent at variance with our previous result (D.66) for Model II.

To proceed, we demonstrate that the gradient corrections (D.16) to the Wigner symbol $\mathcal{G}_{\epsilon,k}$ of the Green's function do not alter the derived result. For concreteness, we illustrate this by considering the second correction (D.55). Since the latter already includes temporal gradients in terms of $b'^2$, it suffies to take the zero frequency limit of the polarization operator, $\Pi_{\omega\to0}^{\mathrm{I}}$, see (D.59), to obtain a correction to the action $S_{\rho\rho}[f]$. With the integration variable $\epsilon$ rescaled as usual, we obtain an intermediate result in the following form:

$$\delta S_{\rho\rho}^{(2)}[f] = -\frac{N}{4}\int_{s,x}\left(\frac{b'}{b}\right)^2\int_{\epsilon,k}\epsilon^2\epsilon_k^2\Big(\partial_\epsilon\overline{G}_{\epsilon,k}\overset{\leftrightarrow}{\partial}_\epsilon\overline{G}_{\epsilon,k}^2\Big)\overline{G}_{\epsilon,k}$$

$$\overset{\widetilde{\epsilon}=\epsilon b^2}{=} -\frac{N}{4}\int_{s,x}b'^2\times\int_{\widetilde{\epsilon},k}\widetilde{\epsilon}^2\epsilon_k^2\Big[\partial_{\widetilde{\epsilon}} G_{\widetilde{\epsilon},k}(\Lambda)\overset{\leftrightarrow}{\partial}_{\widetilde{\epsilon}} G_{\widetilde{\epsilon},k}^2(\Lambda)\Big]G_{\widetilde{\epsilon},k}(\Lambda), \tag{D.71}$$

where the relation (D.25) was used to transition from the first to the second line. Referring to Eq. (50), it becomes evident that the above expression is that of the Schwarzian. To further estimate the energy-momentum integral, we use the UV behavior of the Green's function: $G_{\epsilon,k}\sim|\epsilon J|^{-1/2}$ in the range $T_\Lambda\ll\epsilon\ll J$, and find

$$\delta S_{\rho\rho}^{(2)}[f]\sim N\int_{s,x}b'^2\int_{|k|<k_0}\epsilon_k^2\int_{\epsilon_k^2/J}^{+\infty}\frac{d\epsilon}{(J\epsilon)^2}\sim\frac{Nk_0}{J}\int_{x,\tau}\{f,\tau\}. \tag{D.72}$$

Note that the resulting contribution to the Schwarzian action is subleading compared to the logarithmic one (D.70), as stated earlier. An exact evaluation of the integral (D.71), which we do not reproduce here, confirms this estimate. Additionally, considering another gradient correction to the Wigner symbol, as given by Eq. (D.51), yields a similar result.

The evaluation of the second part of the polarization operator, $\Pi_\omega^{\mathrm{II}}$, see Eq. (D.59), which contributes to the Schwarzian action, proceeds along the same lines as above and does not provide any new insights. Due to the presence of the second-order differential operator $\overset{\leftrightarrow}{\Box}_s$, acting on slow times, one can, as before, take a zero frequency limit, $\omega\to0$, in the above polarization operator. The analytical structure of resulting one-loop momentum-energy integrals happens to be analogous to that in Eq. (D.70). They are free from both UV and IR divergencies and provide an additional contribution of order $Nk_0/J$ to the coupling constant in front of the Schwarzian, cf. Eq. (D.72). Consequently, the relation (D.70) remains unchanged when understood with logarithmic accuracy.

### D.5.2 Central charge and the kinetic term in the AS action

We next proceed with a derivation of the central charge $C$ by analyzing the action $S_{j\rho}[f]$, which brings about the imaginary contribution to the AS action. Following the steps around Eqs. (D.57–D.59), one can rewrite this action in terms of a corresponding polarization operator,

$$S_{j\rho}[f] = \int \partial_x F_1 b_{t_1} P_{t_1 t_2} b_{t_2} \, d^2 t \, dx, \qquad F_1 = F(t_1, x), \qquad (\text{D.73})$$

where the Wigner symbol of $P_{t_1 t_2}$ takes the following form:

$$P_\omega = -\frac{iN}{2} \int_{\epsilon,k} \partial_k \epsilon_k \, \overline{\mathcal{G}}_{\epsilon_+,k} \left( \epsilon^2 + \tfrac{1}{4} \overleftrightarrow{\Box}_s \right) \overline{\mathcal{G}}_{\epsilon_+,k} = P_\omega^{\mathrm{I}} + P_\omega^{\mathrm{II}}, \qquad \epsilon_\pm = \epsilon \pm \omega/2, \qquad (\text{D.74})$$

with the first and the second terms referring to $\epsilon^2$- and $s$-derivative parts of the above polarization operator, respectively. Following our detailed discussion in the previous subsection D.5.1, we may disregard $P_\omega^{\mathrm{II}}$ and use the local approximation to the Wigner symbols, $\overline{\mathcal{G}}_{\epsilon,k} \to G_{\epsilon,k}(\Lambda/b)$, when deriving the central charge $C$ with logarithmic accuracy.

The first part of the polarization operator can be expanded as $P_\omega^{\mathrm{I}} = P_0^{\mathrm{I}} + \frac{\omega^2}{2} P_0'' + \ldots$, with the leading zero-frequency term being UV dominated. As before, we evaluate it by changing the energy integration variable to $\widetilde{\epsilon} = \epsilon b^2$, which yields

$$P_0^{\mathrm{I}} = -\frac{i}{b^4} \times \frac{N}{2} \int_k \int_{|\widetilde{\epsilon}| < J} \widetilde{\epsilon}^2 \, \partial_k \epsilon_k \, G_{\widetilde{\epsilon},k}^2(\Lambda). \qquad (\text{D.75})$$

On making use of the general relation (D.73), the UV contribution to the action $S_{j\rho}[f]$ then reads

$$S_{j\rho}^{\mathrm{UV}}[f] = -\frac{iN}{2} \int_{s,x} \partial_x F F' \times \int_k \int_{|\widetilde{\epsilon}| < J} \widetilde{\epsilon}^2 \, \partial_k \epsilon_k \, G_{\widetilde{\epsilon},k}^2(\Lambda). \qquad (\text{D.76})$$

It needs to be analyzed further in combination with the analogous contribution (D.56) arising from the first-order gradient expansion, which was derived previously. At the end of this subsection, we will demonstrate how a mutual cancellation of these terms, and similar ones, can be achieved.

For now, we proceed by discussing the $P_0''$-term in the frequency expansion, which contributes to the AS action, as will become clear shortly. Following the derivation leading to Eq. (D.70), we evaluate the constant $P_0''$ as

$$\frac{1}{2}(P^{\mathrm{I}})_0'' = -\frac{iN}{4} \int_{k,\epsilon} \epsilon^2 \partial_k \epsilon_k \left[ G_{\epsilon,k}(\Lambda/b) \overleftrightarrow{\Box}_\epsilon G_{\epsilon,k}(\Lambda/b) \right]_{\phantom{.}}^{\widetilde{\epsilon} = \epsilon b^2} \qquad (\text{D.77})$$

$$- \frac{iN}{32\pi^2} \int_k \partial_k \epsilon_k \int_{\sim T_\Lambda}^J \frac{d\widetilde{\epsilon}}{(\Sigma_{\widetilde{\epsilon}}^0)^2} = \frac{iN\Lambda}{16J} \ln \frac{J}{T_\Lambda}.$$

To obtain the above result, which is found with logarithmic accuracy, it is sufficient to use the high-energy asymptotics for the Green's function $G_{\epsilon,k}^{-1} \to -\Sigma_\epsilon^0$, obtained by neglecting its exact momentum dependence, and to cut off the energy integral by the scale $T_\Lambda$ in the IR limit. This second-order frequency expansion produces the following contribution to the action,

$$S_{j\rho}[f] = \frac{1}{2} P_0'' \int dx \sum_\omega (\partial_x F b_s)_{-\omega} \, \omega^2 b_\omega = \frac{iC}{12\pi} \int dx \sum_\omega (\partial_x F b)_{-\omega} \, \omega^2 b_\omega, \qquad (\text{D.78})$$

where we have introduced the central charge,

$$C = \left(\frac{3\pi}{4}\right)\frac{N\Lambda}{J}\ln\frac{J}{T_\Lambda}. \tag{D.79}$$

By transforming the result (D.78) to the time domain and using integration by parts, the action $S_{j\rho}[f]$ reduces to

$$S_{j\rho}[f] = -\frac{iC}{12\pi}\int_{s,x}\partial_x F\, b b'' = -\frac{iC}{24\pi}\int_{s,x}\partial_x F(b b'' - b'^2) - \frac{iC}{12\pi}\int_{s,x}\frac{(\partial_x b)b'}{b^2}, \tag{D.80}$$

where, to obtain the final form of the action, the relation $\partial_x F' = -2\partial_x b/b^3$ has been employed (it follows from $F' = 1/b^2$). Using Eq. (50) for the imaginary part of the AS action, we see that Eq. (D.80) indeed reproduces its expected form up to a residual contribution, i.e.

$$S_{j\rho}[f] = -\frac{iC}{48\pi}\int_{x,\tau}\frac{f''\partial_x f'}{f'^2} - \frac{iC}{12\pi}\int_{s,x}\frac{(\partial_x b)b'}{b^2} \equiv i\,\mathrm{Im}\,S_-^{\mathrm{AS}}[b] + \Delta S_{j\rho}[b]. \tag{D.81}$$

(Note that the negative index in the action $S_-^{\mathrm{AS}}[b]$ corresponds to right moving Majoranas). On comparing the result with the counter term $\Delta S_\rho[f]$ (D.48), we see that the latter exactly cancels against $\Delta S_{j\rho}[b]$. In this way we have achieved our goal: To show that the sum of two actions, $S_\rho[f] + S_{j\rho}[f]$, reproduces the kinematic (or imaginary) part of the AS action. By using the previously found ratios of the coupling constants $C/u$ for both models, see Eqs. (D.66) and (D.70), together with the value of the central charge (D.79), the velocity $u(T)$ can now be easily recovered as listed in the beginning of Sec. 2.4.

### D.5.3 Analysis of the UV terms

Here we discuss why the UV contributions (D.56) and (D.76), which were previously identified, can be disregarded. We start by noting that similar UV contributions arise when one analyses higher-order terms of the type $\mathrm{tr}(j^1\overline{\mathcal{G}}[\rho\overline{\mathcal{G}}]^n)$ in the action of fluctuations (63). It is not hard to verify that the UV part of such $n$-th order term reads

$$S_{j\rho^n}^{\mathrm{UV}}[f] = \frac{N}{2}\int_{s,x}\partial_x F F' \times \int_k\int_{|\tilde\epsilon|<J}\tilde\epsilon\,\partial_k\epsilon_k\,G_{\tilde\epsilon,k}(\Lambda)[-i\tilde\epsilon G_{\tilde\epsilon,k}(\Lambda)]^n. \tag{D.82}$$

As a result, the following geometric series can be summed up,

$$\mathcal{S}^{\mathrm{UV}}[f] = \sum_{n=0}^{+\infty}S_{j\rho^n}^{\mathrm{UV}}[f] = \frac{N}{2}\int_{s,x}\partial_x F F' \times \int_k\int_{|\tilde\epsilon|<J}\tilde\epsilon\,\partial_k\epsilon_k\,\mathbf{G}_{\tilde\epsilon,k}(\Lambda), \tag{D.83}$$

where the full Green's function is given by the standard relation

$$\mathbf{G}_{\tilde\epsilon,k}^{-1}(\Lambda) = i\tilde\epsilon + G_{\tilde\epsilon,k}^{-1}(\Lambda) = i\tilde\epsilon - \epsilon_k - \Sigma(\tilde\epsilon,\Lambda), \tag{D.84}$$

with $\Sigma(\tilde\epsilon,\Lambda)$ being the self-energy found in the mean-field approximation. At this stage, we observe that, up to terms of order $\mathcal{O}(a_x^2)$, the result (D.83) can be equivalently represented as a difference of two actions,

$$\mathcal{I}_{a_x}^{\mathrm{UV}}[f] - \mathcal{I}_{a_x=0}^{\mathrm{UV}}[f] = \mathcal{S}^{\mathrm{UV}}[f] + \mathcal{O}(a_x^2), \qquad a_x = \partial_x F. \tag{D.85}$$

Here, the newly introduced action $\mathcal{I}_{a_x}^{\mathrm{UV}}[f]$ is defined as

$$\mathcal{I}_{a_x}^{\mathrm{UV}}[f] = -\frac{N}{2}\int_{s,x}F'\int_k\int_{|\tilde\epsilon|<J}\ln\left[i\tilde\epsilon - \epsilon_{k+\tilde\epsilon a_x} - \Sigma(\tilde\epsilon,\Lambda)\right], \tag{D.86}$$

with the momentum $k$ in the dispersion relation being shifted by the 'vector potential' $a_x$ and the energy $\widetilde{\epsilon}$ playing the role of an effective charge.

At this stage, the Green's function $\mathbf{G}_{\widetilde{\epsilon},k}(\Lambda)$ can be regarded as an operator acting in momentum space and parametrically depending on the energy $\widetilde{\epsilon}$. Following this logic, the UV action can be expressed via the determinant,

$$\mathcal{I}_{a_x}^{\mathrm{UV}}[f] = \frac{N}{2} \int_{s,x} F' \int_{|\widetilde{\epsilon}|<J} \ln\det \mathbf{G}_{\widetilde{\epsilon},\hat{k}+\widetilde{\epsilon}a_x}(\Lambda). \tag{D.87}$$

Taking into account the commutation relation $[\hat{x},\hat{k}] = i$, one can further write:

$$\mathbf{G}_{\widetilde{\epsilon},\hat{k}+\widetilde{\epsilon}a_x}(\Lambda) = e^{-i\widetilde{\epsilon}F(s,x)}\, \mathbf{G}_{\widetilde{\epsilon},\hat{k}}(\Lambda)\, e^{i\widetilde{\epsilon}F(s,x)}, \tag{D.88}$$

which shows that the Green's functions above are related by a local gauge transformation in space. If the dispersion relation $\epsilon_k$ did not describe chiral edge modes, it would be possible to identify the determinants of the two Green's functions — one with $a_x = \partial_x F$ and the other with $a_x = 0$ — and the action $\mathcal{S}^{\mathrm{UV}}[f]$ would vanish. Consequently, the non-zero UV contribution (D.83) should be interpreted as the chiral anomaly. We attribute the appearance of this anomaly to the transformation $\mathcal{M}_{3/2}$, which was employed to obtain the regularized action (54) from its defining expression (51). Recall that the determinant of the differential operator $\mathcal{D}$ in that action was regularized by dividing it by $\det\widetilde{\Sigma}^0$, which contains no kinetic part. As a result, any possible chiral anomaly remained uncompensated.

Bearing these considerations in mind, we conclude that the UV contributions (D.56) and (D.76) are absent in the starting $G\Sigma$-action (54) and therefore must be disregarded.

# E    Aspects of (semi)-classical Liouville field theory

This appendix deals with three aspects of (semi)-classical Liouville field theory. Firstly, we calculate the one-loop partition function of the Liouville action (101) and demonstrate its agreement with the result (97). Secondly, we detail on how solutions to the disk equations of motion (111) can be transformed to the strip. At last, we extend the Liouville action to the presence of sources and show that we obtain a finite on-shell action, which allows us to calculate correlation functions.

## E.1    One-loop partition function

The one-loop partition function is obtained by considering Gaussian fluctuations around the field $\phi_0$ given in (114) evaluated on the action (101). We expand $\phi(\tau,x) = \phi_0(\tau) + \varepsilon(\tau,x)$ and impose $\varepsilon(0,x) = \varepsilon(\beta/2,x) = 0$ since the ZZ-boundary conditions are fulfilled by the saddle-point solution. We then obtain

$$S_2[\epsilon] = S_0 + \frac{C}{96\pi u} \int_0^L \int_0^{\beta/2} \mathrm{d}x\, \mathrm{d}\tau\, \varepsilon(\tau,x)\left(-u^2\partial_x^2 - \partial_\tau^2 + \left(\frac{2\pi}{\beta}\right)^2 2\csc\left(\frac{2\pi\tau}{\beta}\right)^2\right)\varepsilon(\tau,x), \tag{E.1}$$

where all source terms integrate to boundary terms. Note that the on-shell action coincides with $S_0$ calculated below (105) and is finite since all divergencies of $\phi_0$ close to the boundary of the strip at $\tau = 0, \beta/2$ cancel. The fluctuation determinant is evaluated as

$$Z_2 = \int_{\varepsilon(0,x)=\varepsilon(\beta/2,x)=0} \mathcal{D}[\varepsilon]e^{-S_2[\varepsilon]} = \frac{\tilde{N}}{\det(D)^{\frac{1}{2}}}e^{-S_0}, \tag{E.2}$$

where $\det(D)$ is the determinant of the differential operator in the quadratic action (E.1) acting on functions with prescribed boundary conditions. To obtain its eigenvalues, we first factorize the part acting in the $x$-direction by a Fourier expansion

$$\varepsilon(\tau, x) = \frac{1}{L} \sum_m e^{2\pi i m x / L} \varepsilon_m(\tau). \tag{E.3}$$

Ignoring the spatial dependence for now, we are left with the Schroedinger problem

$$\left( -\partial_\tau^2 + 2 \left( \frac{2\pi}{\beta} \right)^2 \csc \left( \frac{2\pi\tau}{\beta} \right)^2 \right) \varepsilon(\tau) = \lambda^2 \varepsilon(\tau). \tag{E.4}$$

Using the substitution

$$u = \cot \left( \frac{2\pi\tau}{\beta} \right), \qquad (u^2 + 1) = \csc \left( \frac{2\pi\tau}{\beta} \right)^2, \tag{E.5}$$

with $u(0) \to \infty$ and $u(\beta/2) \to -\infty$, the problem is brought into the form

$$\left( \frac{2\pi}{\beta} \right)^2 (u^2 + 1) \left[ -(u^2 + 1)\partial_u^2 - 2u\partial_u + 2 \right] \varepsilon(\tau) = \lambda^2 \varepsilon(\tau). \tag{E.6}$$

This is the Legendre differential equation with two independent solutions [97]

$$\varepsilon(u) = c_1 P_1^n(iu) + c_2 Q_1^n(iu), \qquad n = \frac{\beta\lambda}{2\pi}, \tag{E.7}$$

where $Q_m^n(u)$ and $P_m^n(u)$ are the associated Legendre functions. Our goal is now to find a restriction on the allowed values of $\lambda$. We start by noting that $P_1^n(iu)$ grows as $|u|$ for $|u| \to \infty$ for all $n$ and hence is not normalizable, i.e. it does not fulfill the given boundary conditions. More interesting is the solution

$$Q_1^n(u) = \frac{\pi}{2} \left( -\sin \left( \frac{(1+n)\pi}{2} \right) w_1(1, n, u) + \cos \left( \frac{(1+n)\pi}{2} \right) w_2(1, n, x) \right), \tag{E.8}$$

with

$$w_1(1, n, u) = \frac{2^n \Gamma(1 + \frac{n}{2})}{\Gamma(\frac{3}{2} - \frac{n}{2})} (1 - u^2)^{-n/2} {}_1F_2 \left[ \begin{matrix} -\frac{1+n}{2}, \frac{2-n}{2}, \frac{1}{2} \\ u^2 \end{matrix} \right],$$

$$w_2(1, n, u) = \frac{2^n \Gamma(1 + \frac{n}{2})}{\Gamma(1 - \frac{n}{2})} u(1 - u^2)^{-n/2} {}_1F_2 \left[ \begin{matrix} -\frac{n}{2}, \frac{3-n}{2}, \frac{3}{2} \\ u^2 \end{matrix} \right], \tag{E.9}$$

where ${}_1F_2 \left[ \begin{smallmatrix} a,b,c \\ d \end{smallmatrix} \right]$ denote the hypergeometric functions and the representation of $Q_1^n(u)$ can be continued to $u \in \mathbb{C} \backslash ((-\infty, -1] \cup [1, \infty))$ for $n \in \mathbb{C}$ and $1 + n \notin -\mathbb{N}$, c.f. [97] Theorem 4.3. Importantly, all branch cuts of the function lie in the excluded region of the complex plane. Then, using the expansion of the hypergeometric function for large arguments[15], we obtain

$$Q_1^n(iu) \approx \frac{\alpha u}{\Gamma(2 - n)} + \frac{\beta}{\Gamma(2 - n)u} + \frac{\gamma}{u^2} + O\left( \frac{1}{u^3} \right), \qquad |u| \to \infty \tag{E.10}$$

with $\alpha, \beta, \gamma \in \mathbb{C}$. In order for the diverging term to vanish, we need to demand $2 - n$ to lie on the poles of the $\Gamma$-function. This allows us to conclude that $n$ must take values in $\mathbb{N}_{>1}$, which is the desired quantization condition on $n$.

---

[15]Denoting the first two arguments of the hypergeometric function by $a$ and $b$, the expansion is valid if $a - b \notin \mathbb{N}$, which is the case for us.

Including the spatial modes, we are now in a position to evaluate the one-loop determinant as

$$
\text{Det}\left[\left(-\partial_\tau^2 - u^2\partial_x^2 + 2\left(\frac{2\pi}{\beta}\right)^2 \csc\left(\frac{2\pi\tau}{\beta}\right)^2\right)\right]
$$

$$
= \prod_{m=-\infty}^{\infty}\prod_{n>1}^{\infty}\left(\left(\frac{2\pi m u}{L}\right)^2 + \left(\frac{2\pi n}{\beta}\right)^2\right) = N\prod_{m=-\infty}^{\infty}\prod_{n>1}^{\infty}\left(m^2 + \left(\frac{Ln}{u\beta}\right)^2\right)
$$

$$
= N\prod_{m=-\infty}^{\infty}\prod_{n>1}^{\infty}\left(m^2 - \frac{n^2}{(-u\tau)^2}\right) = N\prod_{m=-\infty}^{\infty}\prod_{n>1}^{\infty}\left(m - \frac{n}{u\tau}\right)\left(m + \frac{n}{u\tau}\right) = N\prod_{m=-\infty}^{\infty}\prod_{|n|>1}^{\infty}\left(m + \frac{n}{u\tau}\right).
$$
(E.11)

Up to the global prefactor $N$ and the transformation $\tau \mapsto -1/\tau$, this determinant is equal to the one obtained in (5.19) of [26]. Following the $\zeta$-function regularization outlined in their work, one obtains exactly the AS partition function $Z_2 = Z(\tau u)$ given in (97) with renormalized central charge $c = C + 13$.

## E.2   Liouville field on the strip

The purpose of this subsection is to derive the general solution for the Liouville field $\phi(w, \overline{w})$ on the strip, see Eq. (112), from the corresponding solution (111) on the disk. To simplify calculations, we introduce an additional pair of coordinates $(z, \overline{z})$ that parametrize the full complex plane and consider two conformal transformations:

$$
z(w) = e^{\frac{2\pi}{\beta}w}, \qquad \xi(z) = \frac{z-i}{z+i}.
$$
(E.12)

The first transformation here maps the half-strip $S_+$ on the upper-half complex plane $C_+$, i.e. $z : S_+ \mapsto C_+$, while the second one maps $C_+$ onto the unit disc $D$, meaning $\xi : C_+ \mapsto D$. The combination of these two mappings, $(\xi \circ z)(w)$, gives the Cayley map (110).

Our approach will be first to obtain the general solution $\widetilde{\phi}(z, \overline{z})$ on the upper-half complex plane by relating it to $\phi'(\xi, \overline{\xi})$, and then to transform $\widetilde{\phi}(z, \overline{z})$ into the desired solution $\phi(w, \overline{w})$ on the strip. To this end, given a 'conformal reparametrization' $f'(\xi)$ of the disk, we consider a related reparametrization $h(z)$ on the upper-half plane, which is defined by the relation

$$
f'(\xi(z)) \equiv \xi(h(z)).
$$
(E.13)

The logic behind this construction can be gained from the following commutative diagram:

$$
\begin{array}{ccc}
f' : D & \xrightarrow{f'} & D \\
\xi \uparrow & & \uparrow \xi \\
h : C_+ & \xrightarrow{h} & C_+
\end{array}
$$

In turn, the Liouville fields in domains $C_+$ and $D$ are related by the transformation law (103), which for the case under consideration becomes

$$
e^{\widetilde{\phi}(z, \overline{z})} = e^{\phi'(\xi, \overline{\xi})}\left|\frac{\partial \xi(z)}{\partial z}\right|^2.
$$
(E.14)

Taking into account the analytical form of the solution (111) for the field $\phi'(\xi, \overline{\xi})$, it is instructive to consider the chain of relations:

$$
\partial_\xi f'(\xi)\Big|_{\xi(z)}\frac{\partial \xi(z)}{\partial z} \overset{(\text{E.13})}{=} \frac{\partial \xi(z)}{\partial z}\Big|_{h(z)}\partial_z h(z) \equiv \partial_z \xi(h(z)).
$$
(E.15)

The latter enables us to express the field $\widetilde{\phi}(z,\overline{z})$ solely in terms of the conformal reparametrization $h(z)$ on the upper-half plane:

$$e^{\widetilde{\phi}(z,\overline{z})} = \frac{4|\partial_z \xi(h(z))|^2}{(1-|\xi(h(z))|^2)^2} \overset{(E.12)}{=} -\frac{4\partial_z h\,\partial_{\overline{z}}\overline{h}}{\left(h(z)-\overline{h(z)}\right)^2}, \qquad (E.16)$$

where the explicit form of the conformal transformation $\xi(z)$, see Eq. (E.12), was used to derive the very last relation.

At this stage, one can rely on the Schwarz reflection principle to state that $\overline{h(z)} = h(\overline{z})$ and thereby to simplify the denominator in Eq. (E.16). Indeed, the reparametrization on the disk satisfies the asymptotic relation $|f'(\xi)| \to 1$ for $|\xi| \to 1$. On the other hand, the (invertible) conformal transformation $\xi(z)$ maps the real axis onto the unit circle, i.e. $\xi : \mathbb{R} \mapsto \partial D$, where $\partial D$ denotes a boundary of the disk. Then from the defining relation (E.13) it follows that the function $h(z)$ is real valued on the real axis. This latter property justifies the usage of the Schwarz principle, as it was suggested above.

In the second step, following the same procedure as above, one can transform $\widetilde{\phi}$ to the desired solution $\phi$ on the half-strip. These two Liouville fields are related as

$$\left(\frac{2\pi}{\beta}\right)^2 e^{\phi(w,\overline{w})} = e^{\widetilde{\phi}(z,\overline{z})}\left|\frac{\partial z(w)}{\partial w}\right|^2. \qquad (E.17)$$

With the help of the commutative diagram,

$$
\begin{array}{ccc}
h : C_+ & \overset{h}{\longrightarrow} & C_+ \\
{\scriptstyle z}\uparrow & & {\scriptstyle z}\uparrow \\
f : S_+ & \overset{f}{\longrightarrow} & S_+
\end{array}
$$

we define the classical reparametrization $f(w)$ on the half-strip $S_+$ by the relation

$$h(z(w)) \equiv z(f(w)), \qquad (E.18)$$

which in somewhat more explicit terms reads as

$$f(w) = \frac{\beta}{2\pi}\ln\left[h(e^{\frac{2\pi}{\beta}w})\right]. \qquad (E.19)$$

In particular, one can see from the above analytical formula that the Schwarz reflection principle for the reparametrization $h(z)$ on the disk guarantees that the same relation, $\overline{f(w)} = f(\overline{w})$, holds on the strip.

To further resolve the Liouville field $\phi(w,\overline{w})$ in terms of $f(z)$, one again manipulates with derivatives,

$$\partial_z h(z)\Big|_{z(w)}\frac{\partial z(w)}{\partial w} \overset{(E.18)}{=} \partial_w z(f(w)). \qquad (E.20)$$

Then from relation (E.17) we finally find

$$e^{\phi(w,\overline{w})} = -\left(\frac{\beta}{\pi}\right)^2\frac{|\partial_w z(f(w))|^2}{\left(z(f(w))-z(f(\overline{w}))\right)^2} \overset{(E.12)}{=} \frac{|\partial_w f(w)|^2}{\sin^2(i\pi(f(\overline{w})-f(w))/\beta)}, \qquad (E.21)$$

thereby recovering the result (112) from Sec. 4.2.

## E.3 Action in presence of a source

In this subsection, we generalize the action (101) to the presence of a source of weight $l$ at position $w_0$ and also calculate its on-shell value. Inspired by the geometric construction of Ref. [78], we set

$$S(l, w_0) = \widetilde{S}(l, w_0) + 2l\delta \ln \frac{2\pi\tau_{\mathrm{uv}}}{\beta}, \qquad (E.22)$$

where $\tau_{\mathrm{uv}} \sim 1/J$ is the short-time cutoff and

$$\widetilde{S}(l, w_0) = \lim_{\tau_{\mathrm{uv}} \to 0} \left[ \frac{C}{24\pi u} \left( \int_{S \backslash \Delta_{\tau_{\mathrm{uv}}}} \mathrm{d}x\,\mathrm{d}\tau \left( \frac{(\partial_\tau \phi)^2}{4} + \frac{(u\partial_x \phi)^2}{4} + \left( \frac{2\pi}{\beta} \right)^2 e^\phi \right) - \oint_{\partial S} \mathrm{d}\vec{l} \cdot \nabla_{\vec{n}} \phi \right) \right.$$
$$\left. - \frac{l}{4\pi i} \oint_{\partial \Delta_{\tau_{\mathrm{uv}}}} \phi \left( \frac{\mathrm{d}w}{w - w_0} - \frac{\mathrm{d}\overline{w}}{\overline{w} - \overline{w}_0} \right) - 2l\delta \ln \frac{2\pi\tau_{\mathrm{uv}}}{\beta} \right], \qquad (E.23)$$

is the regularized action, constructed to remain well-defined in the limit $\tau_{\mathrm{uv}} \to 0$. In the above definition, $\Delta_{\tau_{\mathrm{uv}}}$ denotes a disk of radius $\tau_{\mathrm{uv}}$ around the source, which is omitted in the bulk term, $\vec{n}$ represents the outward pointing normal vector at the boundaries of the strip $S$, and $\delta = 6l/C$.

We now comment on the terms in the second line of (E.23). To this end, we inspect potential singularities in the integrand and show that the subtracted counter-term, $2l\delta \ln(\tau_{\mathrm{uv}}/\beta)$, ensures that the regularized action $\widetilde{S}(l, w_0)$ is finite. Starting with the behavior near the boundary as $\tau \to 0, \beta/2$, only the first line of (E.23) contributes. Notably, in this limit, for the zero-source solution $\phi_0$, the integrand remains finite even though the field itself diverges, as discussed in the previous subsection. By inspection, this result can be generalized to solutions $\phi$ with insertions at arbitrary positions: In the Liouville equation with ZZ boundary conditions (105), the source term becomes negligible close to the boundary compared to the diverging exponential. As a result, it does not alter the leading divergence but only introduces a correction linear in the distance to the boundary, ensuring that the action remains finite.

Moving to the the second line of Eq. (E.23), the first term is the source included in the action expressed as a contour integral via Cauchy's theorem (it is a regularized form of the $\delta$-function in two dimensions). The second one, which we add to (E.22) and subtract here, is a counter-term making the expression in square brackets regular and allowing to take the $\tau_{\mathrm{uv}} \to 0$ limit there. This can be seen the following way: By (103) and (119), the one-source solution close to the source in any geometry behaves as $\phi = -4\delta \ln(r/\beta) + O(1)$, where $r$ is the distance to the source in $w$-coordinates. The only divergence in the first line of (E.23) comes from the kinetic term, whose leading divergence is given by

$$\frac{C}{24\pi} \int_{S \backslash \Delta_{\tau_{\mathrm{uv}}}} \mathrm{d}^2 x \frac{1}{4} \partial_\mu \phi \partial^\mu \phi \sim 2l\delta \int_{\tau_{\mathrm{uv}}}^{\beta} \mathrm{d}r\, r (\partial_r \ln(r/\beta))^2 = -2l\delta \ln(\tau_{\mathrm{uv}}/\beta) + O(1). \quad (E.24)$$

On other hand, the source term diverges as $(-l\phi) = 4l\delta \ln(r/\beta) + O(1)$ and both terms together cancel the divergence of the counter term, which is also logarithmic in $\tau_{\mathrm{uv}}$. Again, due to conformal covariance of $\phi$, this result holds for any source insertion on any geometry. By construction, the full cut-off dependence is shifted onto the last term of (E.23), yielding a normalization constant in the correlation function (124).

In a next step, we want to find the on-shell value of $S(l, w_0)$ which is required to calculate the heavy one-point function (124), and on the way also confirm the conformal transformation behavior used in its derivation. To this end, we define the Liouville action on the $\xi$-disk

$$\tilde{S}'(l,\xi_0) = \lim_{\tau_{uv}\to 0}\left[\frac{C}{24\pi}\left(\int_{D\backslash\Delta_{\tau'_{uv}}}\mathrm{d}\xi\,\mathrm{d}\overline{\xi}\left(\partial_\xi\phi'\partial_{\overline{\xi}}\phi'+e^{\phi'}\right)-\oint_{\partial D}\mathrm{d}\vec{l}\cdot\nabla_{\vec{n}}\phi'\right)\right.$$
$$\left.-\frac{l}{4\pi i}\oint_{\partial\Delta_{\tau'_{uv}}}\phi'\left(\frac{\mathrm{d}\xi}{\xi-\xi_0}-\frac{\mathrm{d}\overline{\xi}}{\overline{\xi}-\overline{\xi}_0}\right)-2l\delta\ln\tau'_{uv}\right], \tag{E.25}$$

where the solution $\phi'$ is related to the strip solution $\phi$ via the transformation law (103), $D$ denotes the $\xi$-disk, and $\Delta_{\tau'_{uv}}$ is again a small disk of radius

$$\tau'_{uv} = \left|\xi(w_0+\tau_{uv}e^{i\theta})-\xi(w_0)\right| \simeq \tau_{uv}\left|\xi'(w_0)\right| \tag{E.26}$$

around the source. The difference of the actions on the strip and on the disk then reads

$$(\tilde{S}(l,w_0)-S(0,w_0))-(\tilde{S}'(l,\xi_0)-S'(0,\xi_0)) = -l(1-\delta)\ln\left|\frac{\beta}{2\pi}\xi'(w_0)\right|^2, \tag{E.27}$$

where we also subtracted the vacuum-actions $S_0 = \tilde{S}_0$ and $S'_0 = \tilde{S}'_0$. Now, the right-hand side of (E.27) stems purely from the second line of (E.23) and (E.25): The contribution $\sim l$ comes from sending $\phi \mapsto \phi'$ using (103) in the source term and the term $\sim l\delta$ is obtained by substituting $\tau_{uv} \mapsto \tau'_{uv}$ in the counter term. Together, both terms give the classical scaling dimension of the vertex operator. To conclude the argument, one must show that there are no other contributions, or equivalently, the first lines of the vacuum-normalized actions (E.23) and (E.25) are equivalent. Since on both geometries the boundary term cancels between the one-source- and the vacuum action and the potential $e^{\phi'}$ is manifestly covariant, the only non-trivial terms are the kinetic ones. For these one can invoke Green's theorem to show that

$$\int \mathrm{d}^2\xi\left(\partial_\xi\phi'\partial_{\overline{\xi}}\phi'-\partial_\xi\phi'_0\partial_{\overline{\xi}}\phi'_0\right) = \int \mathrm{d}^2\xi\left(\partial_\xi\phi\partial_{\overline{\xi}}\phi-\partial_\xi\phi_0\partial_{\overline{\xi}}\phi_0\right), \tag{E.28}$$

i.e. the logarithms appearing in the transformation law from $\phi$ to $\phi'$ can be discarded. From here, one applies a change of variables to recover exactly $\tilde{S}(l,w_0)-S(0,w_0)$, confirming the assertion.

To further evaluate the expression (E.27), we focus on the action difference on the $\xi$-disk. It can be calculated to be finite as [78]

$$\tilde{S}'(l,\xi_0)-S(0,\xi_0) = (\tilde{S}'(l,0)-S'(0,\xi_0))+(\tilde{S}'(l,\xi_0)-\tilde{S}'(l,0))$$
$$= -\ln U(l)-l(1-\delta)\ln\left(\left|h'_\xi(\xi,\xi_0)\right|^2_{\xi=\xi_0}\right). \tag{E.29}$$

Here the second term in the final result follows from the same arguments as above and the first term

$$\ln U(l) = -\frac{C}{6}(2\delta(1-\ln 2)+(1-2\delta)\ln(1-2\delta)) \tag{E.30}$$

contains all the information independent of the position. To find it, we introduce a cut-off $\eta \ll 1$ close to the boundary $|\xi| = 1$ and evaluate the on-shell action $\tilde{S}[l,0]$ without the boundary term on the solution $\phi'(r) \equiv \phi'(r;0,l)$ with $|\xi| = r$ (c.f. (119) for its definition). Expanding this solution close to the origin, one finds that the source term at $\tau'_{uv} \ll 1$ evaluates to

$$S_*(\delta) = -\frac{l}{4\pi i}\oint_{|\xi|=\tau'_{uv}}\phi'\left(\frac{\mathrm{d}\xi}{\xi-\xi_0}-\text{c.c.}\right) = -l\phi'(\tau'_{uv}) = \frac{C\delta}{3}(2\delta\ln\tau'_{uv}-\ln(2-4\delta)). \tag{E.31}$$

Then invoking polar coordinates, the on-shell action becomes (we use $\delta = 6l/C$)

$$I(\delta) = \frac{C}{12} \int_{\tau'_{\text{uv}}}^{1-\eta} \left( \frac{1}{4} \left[ \partial_r \phi'(r) \right]^2 + e^{\phi'(r)} \right) r \, dr + S_*(\delta) - \frac{C\delta^2}{3} \ln \tau'_{\text{uv}}$$

$$\overset{\tau'_{\text{uv}} \to 0}{=} \frac{C}{6} \left( 2\delta(1 - \ln 2) + (1 - 2\delta) \ln(1 - 2\delta) + \eta^{-1} + \ln(2\eta) - 3/2 \right) + \mathcal{O}(\eta), \quad \text{(E.32)}$$

and we obtain $I(0) - I(\delta) \overset{\eta \to 0}{=} \ln U(l)$, as given by Eq. (E.30). The constant $U(l)$ can be identified with the semi-classical limit of the ZZ one-point coefficient [98], modifying the prefactor of the correlation function, while the second term depending on (121) contains the coordinate-dependence of the correlation function. Putting everything together, using $l = C\delta/6$, and fixing the UV-scale as $\tau_{\text{uv}} = 1/J$ we finally obtain the one-source action

$$S(l, w_0) = S_0 - \ln U(l) - \frac{C\delta(1-\delta)}{6} \ln \left( \frac{\beta^2}{4\pi^2} \left| \xi'(w_0) \right|^2 \left| h'_\xi(\xi, \xi_0) \right|^2 \Big|_{\xi=\xi_0} \right) + \frac{C\delta^2}{6} \ln \left| \frac{2\pi}{\beta J} \right|^2, \quad \text{(E.33)}$$

which leads to Eq. (122) from the main text after application of the chain rule for derivatives.

# F  One-source solution to Liouville equation

In this appendix, we derive the one-source solution (119) to the Liouville equation with heavy insertion at $\xi_0 = 0$, which can be derived either from (105) by applying the transformation law (103), or directly from the action (E.25). The strategy is to solve the equation without source insertion and to later implement the corresponding boundary conditions close to the source and close to the boundary. Since the solution will be radially symmetric around the origin, we move to polar coordinates $\xi = re^{i\theta}, \overline{\xi} = re^{-i\theta}$ and neglect the $\theta$-dependence of the equation. We also write the equations in terms of $\Phi = e^\phi$, leading to

$$-2\Phi^2 + \frac{\Phi'}{r} - \frac{\Phi'^2}{\Phi} + \Phi'' = 0, \quad \text{(F.1)}$$

where $\Phi = \Phi(r)$. The general solution reads

$$\Phi(r) = \frac{4c_1^2}{r^2 (e^{c_2} r^{-c_1} - e^{-c_2} r^{c_1})^2}, \quad \text{(F.2)}$$

in terms of two real constants $c_1$ and $c_2$. The behavior close to the source at $r \to 0$ can be determined as $\Phi \sim |r|^{-4\delta}$ by neglecting the exponential term and using $\delta^{(2)}(\xi) = \pi^{-1} \overline{\partial} \xi^{-1}$, with $\delta = 6l_1/C < 1/2$. Comparing this result to the solution (F.2), we obtain $c_1 = \pm(1-2\delta)$, both choices leading to the same result. For the behavior close to the boundary, as proven in slightly different settings in Appendix A of [78] and Appendix B of [71], for $r \to 1$, the Liouville field should diverge as $\Phi \sim 4(1-r^2)^{-2} + O(1)$, i.e. no $O((1-r^2)^{-1})$ term is present. The function $\Phi$ fulfilling this condition is obtained by setting $c_2 = 0$, uniquely recovering the solution (119).

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
