# Peer review of "The chiral SYK model in three-dimensional holography"

_SciPost Physics_

## Round 1 · Referee Report · Anonymous (Referee 1) · 2025-5-21

Report

Dear Editor,

In this work, the authors propose a 1+1d SYK-chain model that has a large N long wavelength limit to an effective description of a dynamical reparametrization governed by the Alekseev-Shatashvili (AS) geometric actions.
Their paper consists of three main sections.
Section 2 performs this limit in the $G\Sigma$ formulation of the model, in a similar way to how Kitaev argued for the Schwarzian model describing the SYK model at large N and late times.
In section 3, the authors derive or review how the AS actions arise from the 3d gravitational path integral, passing through the Chern-Simons and chiral WZW formulations as argued in the past literature.
In section 4, they compute various correlation functions of local observables in the AS model using a mapping between the AS model and the 2d Liouville CFT between ZZ-branes. They study this in the large C limit where a saddle approximation is appropriate. They consider various combinations of light and heavy operators in the large C regime, and in particular also consider the OTOC.
The paper itself is beautifully written, with many pedagogical discussions and derivations. The results themselves are intriguing and are of considerable interest to the wider community (both high-energy theory and condensed matter theory).

I have some small questions:

(1) To what extent can these derivations be trusted in the case of small C? This would be the quantum regime according to both the AS actions, and the 2+1d gravity calculations. It would be good if the authors added some comment on the precise regime of validity of the AS action to describe their chiral SYK model.
(2) In earlier work by some of the authors in the Schwarzian model, they considered the transformation $f'=e^\phi$ to explicitly derive expressions for the partition function and correlators. If one performs the same transformation here directly in the AS action (at zero temperature), this seems to lead to the U(1) chiral boson Floreanini-Jackiw action. Do the authors think this could be a useful perspective in this case as well?
(3) There is a typo in eq (96): ad-bc=1

Beyond these small comments, I strongly recommend the work for publication.

Recommendation

Publish (surpasses expectations and criteria for this Journal; among top 10%)

  • validity: -
  • significance: -
  • originality: -
  • clarity: -
  • formatting: -
  • grammar: -

Author:  Konstantin Weisenberger  on 2025-06-16  [id 5575]

(in reply to Report 1 on 2025-05-21)
Category:
answer to question

Dear Editor,

We would like to thank the Referees for their careful reading of the manuscript and positive recommendations for the publication of our paper at SciPost. Below we provide the answers to the questions raised by the first Referee.

With best regards,
Authors

(1) "To what extent can these derivations be trusted in the case of small C? This would be the quantum regime according to both the AS actions, and the 2+1d gravity calculations. It would be good if the authors added some comment on the precise regime of validity of the AS action to describe their chiral SYK model".

We require the central charge to be large, $C\gg 1$. At the level of the chiral SYK model, this condition justifies the gradient expansion of the $G\Sigma$-action when only reparametrization modes are kept in the path integral. Regarding the derivation of the AS action from the 3d gravity, the semiclassical condition on $C$ is necessary to justify the expansion around the BTZ saddle point and ensures that quantum fluctuations remain weak. It is known that quantizing the Chern-Simons (CS) gauge theory defined on the $SL(2,R) \times SL(2,R)$ group is not equivalent to the quantization of pure 3d gravity, see e.g. Sci-Post Phys. 15, 151 (2023) by Collier, Eberhardt and Zhang. In the quantum regime, when $C \sim 1$, the Chern-Simons description includes quantum fluctuations that violate the physically meaningful choice of metric, since the requirement that the metric be non-degenerate is not naturally built into the CS framework.

We have included this discussion in an added paragraph at the end of Section 3.4.

(2) "In earlier work by some of the authors in the Schwarzian model, they considered the transformation $f’ = e^\phi$ to explicitly derive expressions for the partition function and correlators. If one performs the same transformation here directly in the AS action (at zero temperature), this seems to lead to the U(1) chiral boson Floreanini-Jackiw action. Do the authors think this could be a useful perspective in this case as well?"

It is indeed true that such a transformation in the zero-temperature limit leads to the Gaussian action of the $U(1)$ chiral boson. However, the focus of our paper is primarily on the finite-temperature regime of 3D gravity, which admits BTZ black hole solutions. At finite T, the analogous transformation defined as $e^\phi = \partial_\tau \tan( \pi f/\beta)$ necessarily exhibits a singularity at the point $(\tau_*, x_*)$, where $f(\tau_*, x_*) = \pm \tfrac 12 \beta$. It is not immediately clear how to incorporate this subtlety into the path integral over the $\phi$ variable. In our earlier work [Nuclear Physics B 921, 727 (2017)], where the original SYK model in D=1+0 was treated using the Schwarzian action, only one additional integration over a singular point in time $\tau_*$ was required. In the current context, however, this would amount to an extra integration over all non-contractible closed loops arbitrarily positioned on the 2D torus, along with an as-yet undetermined integration measure, which requires further investigation. In this regard, a mapping onto boundary 2D Liouville field theory, following the approach of T. Mertens [JHEP 2018, 36], as employed in our study, circumvents this problem. In such a mapping, the phase $\phi$ diverges at the fixed boundaries of the time interval $[0,\beta/2]$, independently of the spatial position.

(3) The typo is corrected.

---

## Round 1 · Referee Report · Anonymous (Referee 2) · 2025-5-23

Report

The paper reports progress in finding a holographic description of pure gravity in three dimensions. In Ref. [26,27] the 1+1 dimensional Alekseev-Shatashvili (AS) action has been obtained by rewriting gravity on AdS3. In the presented paper the AS action is derived as an effective theory for an underlying multi particle model with random couplings akin to the SYK model in one dimension less. In order to
achieve validity of the effective description at large time scales the authors propose a particular non-relativistic dispersion relation. The AS action appears as an effective theory for space dependent time reparametrization in the multi particle model. Its parameters can be expressed in terms of particle number, variance of the random couplings, the temperature and depend also on parameters in the dispersion relation. Another part of the paper considers 2+1 dimensional gravity. Studying fluctuations around a BTZ black hole the authors
recast the action into the AS action. Now, the AS parameters are related to Newton's constant, the AdS3 radius and the temperature of the Hawking radiation. The results are applied to compute several correlation functions
including OTOCs exhibiting a maximal Lyapunov coefficient. The paper is very interesting and reports results obtained by explicit computations. I recommend publication without further changes.

Recommendation

Publish (surpasses expectations and criteria for this Journal; among top 10%)

---

## Editorial Decision

accepted_in_target_journal